# Uncertainty-quantified Pulse Signal Recovery from Facial Video using Regularized Stochastic Interpolants

## Abstract

Imaging Photoplethysmography (iPPG), an optical procedure which recovers a human's blood volume pulse (BVP) waveform using pixel readout from a camera, is an exciting research field with many researchers performing clinical studies of iPPG algorithms. While current algorithms to solve the iPPG task have shown outstanding performance on benchmark datasets, no state-of-the art algorithms, to the best of our knowledge, performs test-time sampling of solution space, precluding an uncertainty analysis that is critical for clinical applications. We address this deficiency though a new paradigm named *Regularized Interpolants with Stochasticity for iPPG (RIS-iPPG)*. Modeling iPPG recovery as an inverse problem, we build probability paths that evolve the camera pixel distribution to the ground-truth signal distribution by predicting the instantaneous flow and score vectors of a time-dependent stochastic process; and at test-time, we sample the posterior distribution of the correct BVP waveform given the camera pixel intensity measurements by solving a stochastic differential equation. Given that physiological changes are slowly varying, we show that iPPG recovery can be improved through regularization that maximizes the correlation between the residual flow vector predictions of two adjacent time windows. Experimental results on three datasets show that RIS-iPPG provides superior reconstruction quality and uncertainty estimates of the reconstruction, a critical tool for the widespread adoption of iPPG algorithms in clinical and consumer settings.

## 1 Introduction

Vital signs estimation using cameras has recently received strong interest in the research community. Extending photoplethysmography (PPG)—the technique in which a light is shined transdermally through the skin, the reflections of which capture volumetric changes due to blood flow—imaging Photoplethysmography (iPPG) seeks to observe the same volumetric changes using a non-contact imager of the skin, typically an RGB camera. The pixel intensity of the camera, under mild assumptions and noise, captures the skin pigmentation changes due to blood flow. Current iPPG algorithms that denoise the camera signal to estimate the BVP signal are based on traditional signal processing or deep learning methods. Traditional signal processing methods De Haan & Jeanne (2013); De Haan & Van Leest (2014); Poh et al. (2010); Lewandowska & Nowak (2012); Nowara et al. (2021) recover signals in training-free paradigms by assuming inherent signal structure and priors—whether it be statistical independence Poh et al. (2010), uncorrelated signals Lewandowska & Nowak (2012), color demixing De Haan & Jeanne (2013); De Haan & Van Leest (2014), or Fourier-based sparsity Nowara et al. (2020). Deep learning methods generally perform better, but require training: supervised learning methods use synchronized facial video and contact PPG, and assume that the pulse signal structure is best learned though a model that maps from video to PPG data. Advances in self-supervised Gideon & Stent (2021); Sun & Li (2024); Yue et al. (2023b); Speth et al. (2023); Liu et al. (2024) algorithms learn using facial video data only, obviating contact PPG, and achieve competitive performance with fully supervised methods. These algorithmic advances on lab data have led to clinical validation studies of iPPG algorithms Huang et al. (2024) in neonatal care units, or in emergency departments for acute trauma injuries Shenoy et al. (2025).

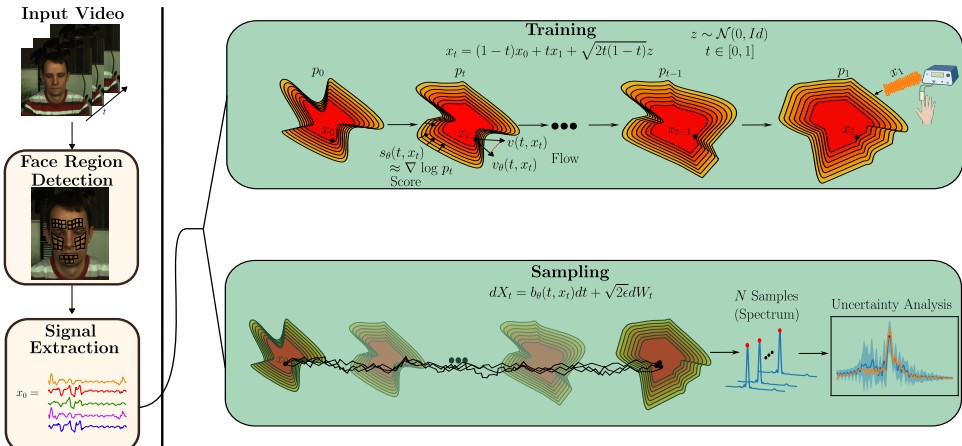

Figure 1: We first preprocess the video to extract a signal estimate from various facial regions. During training time, we learn the score and flow between the measurement distribution and the ground-truth distribution, regularized based on the temporal characteristics of the signal. During test time, we solve an SDE with the measurement as the initial condition, and sample solution space. We then perform an uncertainty analysis.

However, previous iPPG algorithms are deficient at rationalizing the results at test time for end users like clinicians. A previous study Tonekaboni et al. (2019) interviewed clinicians regarding their *trust* of machine learning models, which noted that "metrics such as reliability, specificity, and sensitivity were important for the initial uptake of an AI tool, [but] a critical factor for continued usage was whether the tool was repeatedly successful in prognosticating patient's condition in [the doctor's] personal experience". Each of the aforementioned algorithms achieved state-of-the-art performance on population level metrics, but to the best of our knowledge, no previous iPPG algorithm samples the solution space of potential iPPG signals at test time *for each test sample.* Sampling the solution space would allow for *uncertainty quantification* for each individual test-time sample, a crucial tool for eventual clinical adoption of such algorithms Begoli et al. (2019); Tonekaboni et al. (2019). This type of analysis is especially important on protected attributes like skin tone and gender.

To address uncertainty quantification, we propose *Regularized Interpolants with Stochasticity for imaging Photoplethysmography (RIS-iPPG)*, a flow-based diffusion model framework that learns a probability path from the distribution of camera pixel intensity signals to blood volume pulse signals, and allows for posterior sampling and *uncertainty estimation* of the recovered pulse signals. We achieve such advances by formulating pulse signal recovery as an inverse problem with coupled camera measurements and ground-truth signals. We then learn the drift coefficient of a Stochastic Differential Equation (SDE) Albergo et al. (2023) that maps the distribution of camera measurements to the distribution of ground-truth signals, implemented in practice by learning flow and score vectors for the training data. However, unregularized flow models do not typically yield best predictions. Given that physiological changes in blood volume are usually slowly varying Nichols et al. (2022), we propose to regularize the predicted flow by maximizing the correlation between the residual flow vectors (i.e. ground-truth flow minus predicted flow) from two adjacent time windows. After training our regularized model, at inference time we extract a pulse signal estimate from facial video and repeat it $N$ times as our initial condition, where $N$ is user specified, and solve for the pulse signal via the aforementioned SDE. Given the $N$ signal estimates, we perform an uncertainty analysis of pulse signal recovery from a facial video.

In summary, our contributions are as follows:

- We formulate pulse signal recovery from video as a posterior sampling method using flow-based diffusion models. Using the framework of stochastic interpolants, we learn the flow and score vectors

that, when integrated into the drift coefficient of an SDE, transform the camera pixel signal to the blood volume pulse signal.

- We propose a Residual Correlation Loss (RCL) that maximizes the correlation between the residuals of predicted flow vectors from two overlapping, adjacent time windows. We show that this regularization can lead to better recovery results.

- We evaluate our algorithm using three datasets and perform test-time sampling of solution space. We show that even when our final prediction is incorrect, our sampling procedure highlights other possible solutions. We are able to capture the modes of the distribution more effectively, while also minimizing the uncertainty around frequency bins that are not of interest. Finally, we evaluate the quality of our uncertainty estimates on protected attributes of the data, establishing the first baselines for uncertainty calibration for the iPPG task.

## 2 Related Works

### 2.1 Imaging Photoplethysmography

After preliminary investigations Wu et al. (2000) showed that peripheral blood volume could be measured using an RGB imager, signal processing algorithms and later deep learning algorithms have been proposed to recover the blood volume signal in noise. Early signal processing methods, modeling the camera signal as a mixture of signals one of which was the pulse, demixed the signals by assuming the pulse signal to be statistically independent of the mixture (Independent Component Analysis) Lewandowska & Nowak (2012) or assumed signals could be demixed along the directions of maximum variance (Principal Component Analysis) Poh et al. (2010). Recognizing skin reflection properties were critical for pulse signal recovery, both De Haan & Jeanne (2013) and Wang et al. (2016) performed skin tone corrections before projecting the signal onto an optimal plane for recovery. Other researchers viewed iPPG recovery as an optimization problem and imposed explicit sparsity Nowara et al. (2020) and low-rank Tulyakov et al. (2016) constraints on the recovered signal.

Recent iPPG research is dominated by deep learning methods, which have shown improved performance compared to model-based methods. Underlying all these methods is the assumption that the non-linear noise and imaging processes that modulate the pulse signal from the body to the camera can be learned through sophisticated neural networks. Using paired video and ground-truth data, previous works developed techniques such as frame differencing Chen & McDuff (2018), temporal shift modules Liu et al. (2020), spatiotemporal CNN Yu et al. (2019), and transformers Yu et al. (2022) to extract the rPPG signals; these architectures were further adapted by others to learn noise profiles Nowara et al. (2021); Liu & Yuen (2023) for better signal denoising. Recent algorithmic advances have demonstrated that deep learning-based iPPG extractors can be learned without a supervisory PPG signal Gideon & Stent (2021); Yue et al. (2023b); Sun & Li (2024); Liu et al. (2024) and have achieved competitive performance with fully supervised methods. Researchers have also explored newer problem domains for iPPG such as federated learning Liu et al. (2022b), few-shot learning Liu et al. (2021), and on-device iPPG recovery Liu et al. (2023).

### 2.2 Flow-based Diffusion Models

The inspiration for density estimation and transport-based sampling was founded on Gaussianizing data through some transformation, and undoing that transformation to recover the distribution Tabak & Vanden-Eijnden (2010); Chen & Gopinath (2000). A few works obtained the transformation as the solution of an ODE Chen et al. (2018); Grathwohl et al. (2019), of which the drift coefficient could be learned through neural networks. However, learning the drift is intractable at large scale due to the simulation of the ODE for learning. While some other works proposed to regularize the path Finlay et al. (2020); Onken et al. (2021); Tong et al. (2020), many problems persisted.

Others took a stochastic view of the problem, modeling the transformation of a data distribution to a Gaussian as the evolution of of an Ornstein-Ulhenbeck (OU) process. Traversing this process forward in time simply involves adding Gaussian noise, while reversing this process can be done if the gradient of the

log of the time-dependent data density is available Hyvärinen & Dayan (2005); Vincent (2011): this quantity, called the score function, could be estimated via least-squares regression Song & Ermon (2019). The major drawback of this method was its reliance on the OU process and Gaussians; while some works used bridges to map between arbitrary distributions, these formulations were complex and inexact.

Recent work has introduced simulation-free methods for mapping between two arbitrary probability distributions. The key idea is that this mapping can happen gradually over time as one distribution transforms to another Lipman et al. (2022); Tong et al. (2023). This can be generalized to stochastic dynamics as well Albergo et al. (2023), which defines a stochastic process that maps from one point to another. In either the deterministic or stochastic versions, the goal is to learn the instantaneous change of the time-dependent distribution towards the target, which can be predicted via a neural network and is known as the "flow". In the stochastic case, the score is learned as well. After both are learned, the drift coefficient of the corresponding ODE/SDE can be learned, after which off-the-shelf solvers can solve the differential equation given an initial condition, mapping the point from one distribution to another. These techniques can learn the time-dependent vital sign trajectories of patients in the ICU Zhang et al. (2025), generate new types of inorganic crystalline materials Hoellmer et al. (2025), and learn the manifold of cellular dynamics Tong et al. (2023).

## 3 Background: Stochastic Interpolants

Our goal is to link two related probability distributions, the camera pixel distribution and the ground-truth pulse distribution, and build a time-dependent probability path between them. The recently proposed work of Stochastic Interpolants Albergo et al. (2023) achieves this in finite time, and exactly, by defining a stochastic process that smoothly interpolates from a sample in one distribution to a sample in another distribution. The key goal is to learn, through neural networks, the instantaneous flow and score at all points interpolated between our two distributions. If the flow and score can be learned effectively, then we can take a series of steps (i.e. solving an SDE) that maps an initial condition (i.e. camera pixel signal) from one distribution to a point in the other distribution (i.e. blood volume pulse signal). Let our two arbitrary distributions be $p_0$ and $p_1$, let $\mathbf{x_0} \sim p_0$ and $\mathbf{x_1} \sim p_1$. Consider the following stochastic interpolant:

$$\mathbf{x}_t = I(t, \mathbf{x}_0, \mathbf{x}_1) + \gamma(t)\mathbf{z}, \text{ where } \mathbf{z} \sim \mathcal{N}(0, Id), t \in [0, 1] \tag{1}$$

The function $I(t, \mathbf{x}_0, \mathbf{x}_1)$ smoothly interpolates between the points $\mathbf{x}_0$ and $\mathbf{x}_1$ as a function of $t$ such that $\mathbf{x}_{t=0} = \mathbf{x}_0$ and $\mathbf{x}_{t=1} = \mathbf{x}_1$, while $\gamma(t)\mathbf{z}$ is an added noise term. To bridge between these points, we would like to know the instantaneous change of $\mathbf{x}_t$, which if learned for all $t$, would enable us to transport $\mathbf{x}_1$ to $\mathbf{x}_0$ or vice versa via an SDE. The instantaneous change of the interpolant $\mathbf{x}_t$ with respect to time, $\mathbf{b}(t, \mathbf{x})$, is known as velocity, and the score of the distribution at a time $t$, $\mathbf{s}(t, \mathbf{x})$, is

$$\mathbf{b}(t, \mathbf{x}) = \mathbb{E}[\frac{\partial}{\partial t}\mathbf{x_t}|\mathbf{x_t} = \mathbf{x}], \mathbf{s}(t, \mathbf{x}) = \nabla \log p_t(\mathbf{x}) = -\gamma^{-1}(t)\mathbb{E}(\mathbf{z}|\mathbf{x_t} = \mathbf{x}) \tag{2}$$

It is still to be shown, however, that these quantities can be used to form a valid drift coefficient of an SDE that maps probability mass from one distribution to another. We repeat below the theorem of Albergo which showed that this interpolant, and associated flow and score, satisfy both the continuity equation and a family of Fokker-Planck equations, allowing us to build the drift coefficient of an SDE that solves the mapping between the two distributions.

**Theorem 3.1.** (Theorem 2.6 of Albergo et al. (2023)) The probability distribution of the interpolant $\mathbf{x}_t$ is absolutely continuous with respect to the Lebesgue measure at times $t \in [0, 1]$ and solves the transport equation

$$\frac{\partial}{\partial t}p_t + \nabla \cdot (\mathbf{b}p) = 0 \tag{3}$$

In addition, the forward and backward Fokker-Planck equations are satisfied

$$\frac{\partial}{\partial t}p_t + \nabla \cdot (\mathbf{b}_F p) = \epsilon(t)\Delta p, \mathbf{b}_F = \mathbf{b}(t,\mathbf{x}) + \epsilon(t)\mathbf{s}(t,\mathbf{x}) \tag{4}$$

$$\frac{\partial}{\partial t}p_t + \nabla \cdot (\mathbf{b}_B p) = -\epsilon(t)\Delta p, \mathbf{b}_B = \mathbf{b}(t,\mathbf{x}) - \epsilon(t)\mathbf{s}(t,\mathbf{x}) \tag{5}$$

where $\epsilon(t)$ is some noise schedule.

$\square$

As a consequence of this theorem, if we can learn the flow and score of the interpolant bridging our two data distributions, we can build a drift coefficient and solve a SDE that smoothly transforms a data point from one distribution to another.

## 4 Problem Formulation and Approach

### 4.1 The iPPG signal model

The arterial tree can be modeled as a branching system of elastic tubes that carry blood to the body Nichols et al. (2022). Pressure differences at the ends of the tubes, induced by pump a called the heart, generate the flow of the liquid through the tubes Nichols et al. (2022). The flow can be measured using optical sensors which shine light transdermally and record reflection of the light corresponding to the blood volume; this technique is called photoplethsmography Alian & Shelley (2014) and is common in consumer smartwatches. Imaging Photoplethysmography or remote Photoplethysmography aims to replicate PPG but replaces the contact-based optical sensor with a non-contact camera McDuff (2023).

This imaging setup can be modeled as two processes. Given the volumetric flow signal $\mathbf{x}_0$, the first process generates an analog signal on the skin via reflections of incident illumination on $\mathbf{x}_0$ modulated by physical structures in the dermis/epidermis (known as the physiological forward process) and physiological noise. The second process converts the analog skin signal to a digital signal; in non-contact iPPG, the analog signal is modulated by digital camera hardware (known as the forward imaging process) and imaging noise. To simplify the model, we unify both processes to represent a digital camera signal $\mathbf{x}_1$ as:

$$\mathbf{x}_1 = A(\mathbf{x}_0) + \mathbf{n} \tag{6}$$

where the unknown $A(\cdot)$ models the unified forward processes composed of the imaging forward processes acting on the result of the physiological forward process and noise, and $\mathbf{n}$ is the sum of the physiological and imaging noise. Previous work models the recovery of $\mathbf{x}_0$ as a regularized optimization problem with an approximate forward operator. In the next section, we show the deficiency of this approach, and the need for a new paradigm.

### 4.2 Preliminary Investigation

We begin by generating $(\mathbf{x}_0, \mathbf{x}_1)$ pairs from the data. Given a video signal of time $T$, we extract time-domain pixel intensity signals from five regions of the face to generate $\mathbf{x}_1 \in \mathbb{R}^{T \times 5}$ as in Figure 1; the corresponding ground-truth BVP signal measured from the finger is repeated five times to generate $\mathbf{x}_0 \in \mathbb{R}^{T \times 5}$, resulting in paired data $(\mathbf{x}_0, \mathbf{x}_1)$. Our goal, given the camera pixel intensity signal $\mathbf{x}_1$, is to recover the signal $\mathbf{x}_0$ *and* sample the space of possible solutions. Starting from Equation 6, we decided to follow a regularized optimization scheme. Using $A = \mathbf{F}^{-1}$ Shenoy et al. (2023); Nowara et al. (2020), where $\mathbf{F}^{-1}$ is the inverse Fourier Transform, our initial investigation modeled signal recovery as a regularized optimization problem, $\min_{\mathbf{x}} \frac{1}{2}\|\mathbf{x}_1 - A(\mathbf{x})\|_2^2 + \lambda \cdot h(\mathbf{x})$, where $h(\mathbf{x}) = -\log p(\mathbf{x})$ is the log of the data distribution. This is commonly referred to as the score function Song & Ermon (2019), and is learned through neural networks. The optimization problem can then be solved with posterior sampling via the Plug-and-Play Monte-Carlo Approach of Sun et al. (2024). Our preliminary results, labeled as *PMC-iPPG*, are in Table 1.

While this approach allows for test-time sampling, the performance is unsatisfactory. The key drawback is that the score only learns the signal prior i.e. the distribution of ground-truth volumetric signals, not

Table 1: Applying PMC Sun et al. (2024) to the iPPG task, and comparing against regularized optimization methods with sparse priors Nowara et al. (2020) and learned priors Shenoy et al. (2023). Formulating the iPPG task as regularized optimization problem with a plugged-in prior is ineffective.

| Method | Sampling? | MAE (bpm) ↓ | RMSE (bpm) ↓ |
|---|---|---|---|
| AutoSparsePPG Nowara et al. (2020) | ✗ | 4.55 | 14.42 |
| Unrolled-iPPG Shenoy et al. (2023) | ✗ | 1.11 | 2.97 |
| PMC-iPPG | ✓ | 12.42 | 23.98 |

the mapping $A(\cdot)$ from pulse to camera signals. The key failing is the approximation $A(\cdot) \approx \mathbf{F}^{-1}$, which is equivalent to a signal denoising problem in additive gaussian noise; this was a valid assumption in Nowara et al. (2020) because they assumed Gaussian denoising and used orthogonal projection to reduce noise *before* implementing their signal recovery algorithm. While the approximation of the unknown $A(\cdot) \approx \mathbf{F}^{-1}$ was sufficient for Shenoy et al. (2023), their unrolling method *implicitly* corrected the approximate forward model through end-to-end training. Fundamentally, however, PMC-iPPG reveals that $A = \mathbf{F}^{-1}$ is inadequate without extra noise reduction and implicit model correction.

Naturally, the next step would be to build a more accurate forward model. Initial experiments to learn the forward model followed the approach of previous literature Lunz et al. (2020); Arridge et al. (2023); however, we noticed that models would not converge because the forward model is a time-dependent and facial-region-specific function of the pulse signal, specular and diffuse reflections, skin tone, and motion. See Appendix A.3.1 for a full discussion. Instead of learning networks to model both the forward model (which did not converge) as well as the signal prior, a single posterior sampling framework that encapsulates both of the aforementioned models and the many-to-one mapping between face regions and pulse signals may be more effective. Stochastic interpolants provides such a framework that allows for test-time sampling.

### 4.3 Unregularized Stochastic Interpolants for iPPG

Without an explicit forward model, we seek to learn an implicit mapping of BVP signals to camera pixel signals. We assume that there exists a distinct BVP and camera pixel distribution, and that there exists a mapping *in distribution* between them.

Given these two paired data distributions, we seek to define a a stochastic process that smoothly interpolates between samples from two distributions, namely the camera intensity signals $\mathbf{x}_1 \sim p_1$ and its ground-truth pulse signal $\mathbf{x}_0 \sim p_0$.

$$\mathbf{x}_t = (1-t)\mathbf{x}_0 + t\mathbf{x}_1 + \sqrt{2t(1-t)}\mathbf{z} \tag{7}$$

This choice of $I(t, \mathbf{x}_0, \mathbf{x}_1) = (1-t)\mathbf{x}_0 + t\mathbf{x}_1$ and $\gamma(t) = \sqrt{2t(1-t)}$ as in Equation 1 ensures that at $\mathbf{x}_{t=0} = \mathbf{x}_0$ and $\mathbf{x}_{t=1} = \mathbf{x}_1$. Using this interpolant, our goal is to build a drift coefficient that satisfies Theorem 3.1. We are looking for the instantaneous change of Equation 7: therefore, we are seeking $\mathbf{b}(t, \mathbf{x}) = \mathbb{E}[\frac{\partial}{\partial t}((1-t)\mathbf{x}_0 + t\mathbf{x}_1 + \sqrt{2t(1-t)}\mathbf{z})]$, learned through neural networks, as described in Section 3. In practice, however, we can decompose $\mathbf{b}(t, \mathbf{x}) = \mathbf{v}(t, \mathbf{x}) - \gamma(t) \cdot \frac{\partial}{\partial t}(\sqrt{2t(1-t)}) \cdot \mathbf{s}(t, \mathbf{x})$, and further decompose the score using Tweedie's formula as $\mathbf{s}(t, \mathbf{x}) = -\mathbf{n_z}(t, \mathbf{x})/\gamma(t)$. The decomposition of the drift and denoiser can be found in Section A.8 and Section A.6 and follows the derivation presented in Albergo et al. (2023). This simplifies the practical implementation to learning two independent networks, one to learn the interpolant flow and another to learn the denoiser.

$$\mathbf{v}_\theta(t, \mathbf{x}) \approx \frac{\partial}{\partial t}((1-t)\mathbf{x}_0 + t\mathbf{x}_1) \tag{8}$$

$$\mathbf{n}_\theta(t, x) \approx \mathbf{z} \tag{9}$$

This is preferable to learning the drift $\mathbf{b}(t, \mathbf{x})$ directly as we can avoid evaluating $\gamma^{-1}(t)$ near $t \approx 0$ and $t \approx 1$. We can learn both of these networks by minimizing the MSE Albergo & Vanden-Eijnden (2022):

$$\mathcal{L}_{\text{flow}} = \mathcal{L}_{\text{MSE}}(\mathbf{v}_\theta(t, \mathbf{x}), \mathbf{v}(t, \mathbf{x})) \tag{10}$$

$$\mathcal{L}_{\text{score}} = \mathcal{L}_{\text{MSE}}(\mathbf{n}_\theta(t, \mathbf{x}), \mathbf{z}) \tag{11}$$

After these networks are learned, we can build a drift coefficient (as shown in Section 4.5) and use off-the-shelf solvers to obtain a pulse estimate given the camera measurements (i.e. initial condition).

While this worked in practice, we noticed many failure cases. Facial data is often corrupted by out-of-distribution and unconstrained motion noise. To make our models more robust, we attempted to condition our flow and score networks on guidance signals based on the noise in the video; however, our investigation yielded negative results (see Appendix A.3.2). We took a different approach when noticing that physiological changes are slowly varying. Medical research has discovered that physiological changes under normal conditions are often slowly varying Nichols et al. (2022) and changes in physiological state are often time-delayed. This implies that in adjacent and overlapping time windows, physiological signals should be similar. A robust flow model should ensure such temporal consistency, which we learned through a regularization scheme below.

### 4.4 Residual Correlation Loss (RCL) for temporal consistency in adjacent time windows

We learn such temporal consistency by correlating the residual vectors between the predicted and ground-truth flows of two adjacent time windows as shown in Figure 2. Assume we are given two pairs of data,

$$(\mathbf{x}_0(i), \mathbf{x}_1(i)) \text{ and } (\mathbf{x}_0' = \mathbf{x}_0(i - \delta), \mathbf{x}_1' = \mathbf{x}_1(i - \delta)) \tag{12}$$

where the latter pair is an overlapping, time-shifted version of the first pair. During training, we generate $\mathbf{x}_t$ and $\mathbf{x}_t'$ according to Equation 7, after which we predict the flow at each of these points $\mathbf{v}_\theta(t, \mathbf{x}_t)$ and $\mathbf{v}_\theta(t, \mathbf{x}_t')$ as described in Section 4.3. We note the predicted flows should be regressed to their corresponding targets $\mathbf{v}(t, \mathbf{x}_t)$ and $\mathbf{v}(t, \mathbf{x}_t')$; however, given that adjacent and overlapping time windows should have consistent physiological behavior, the error vector between the predicted and ground-truth flows (i.e. the residual) should be correlated. More precisely, we would like the residual vectors to point in the same direction.

To encourage vectors to point in the same direction, we aim to maximize the normalized dot product between the vectors. This is achieved by minimizing the proposed Residual Correlation Loss, which is equivalent to minimizing one-minus the Pearson Correlation Coefficient Cohen et al. (2009). Let $\mathbf{p} = \mathbf{v}(t, \mathbf{x}_t) - \mathbf{v}_\theta(t, \mathbf{x}_t)$ and $\mathbf{q} = \mathbf{v}(t, \mathbf{x}_t') - \mathbf{v}_\theta(t, \mathbf{x}_t')$. Then, the RCL loss is defined as

$$\mathcal{L}_{\text{RCL}}(\mathbf{p}, \mathbf{q}) = 1 - \frac{T \cdot \mathbf{p}^\top \mathbf{q} - \mu_p \mu_q}{\sqrt{(T \cdot \mathbf{p}^\top \mathbf{p} - \mu_p^2)(T \cdot \mathbf{q}^\top \mathbf{q} - \mu_q^2)}} \tag{13}$$

where $\mu_p$ and $\mu_q$ are the means of the signals, and $T$ is the length of the signal. We will show in Section 5 that minimizing this loss leads to improved iPPG recovery.

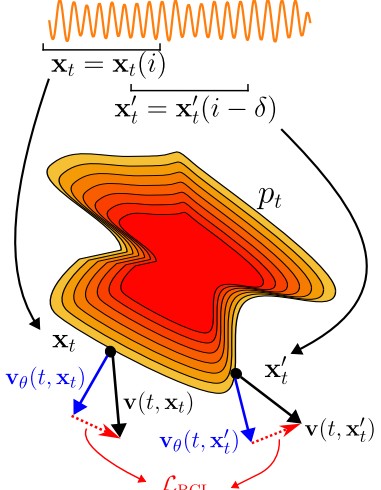

Figure 2: We sample a time-window and its time-shifted version, and predict the flow for both. For two adjacent and overlapping time-windows, the residual vector between predicted and ground-truth flows should point in the same direction, which is promoted by by minimizing the Residual Correlation Loss.

The correlation loss leads to a regularized network that promotes temporal consistency. This loss can simply be added at the training stage of our network, without any changes to the model architecture. Therefore, during training, we can add the RCL loss to the flow and score losses from Equation 10 and Equation 11. The final loss is given by

$$\mathcal{L} = \underbrace{\mathcal{L}_{\text{MSE}}(\mathbf{v}_\theta(t, \mathbf{x}), \mathbf{v}(t, \mathbf{x}))}_{\text{flow}} + \underbrace{\mathcal{L}_{\text{MSE}}(\mathbf{n}_\theta(t, \mathbf{x}), \mathbf{z})}_{\text{denoiser}} + \lambda_{\text{RCL}} \mathcal{L}_{\text{RCL}}(\mathbf{p}, \mathbf{q}) \tag{14}$$

Table 2: Heart rate estimation results on MMSE-HR, PURE and UBFC-rPPG datasets. Best results in each column are **bold**; second-best are underlined. Our method is the only one that addresses uncertainty quantification (UQ)

| Type | Method | MMSE-HR | | | PURE | | | UBFC-rPPG | | |
|---|---|---|---|---|---|---|---|---|---|---|
| | | MAE↓ | RMSE↓ | ρ↑ | MAE↓ | RMSE↓ | ρ↑ | MAE↓ | RMSE↓ | ρ↑ |
| Model-Based Unsupervised | ICA Poh et al. (2010) | 5.44 | 12.00 | - | - | - | - | 7.50 | 14.41 | 0.62 |
| | CHROM De Haan & Jeanne (2013) | 3.74 | 8.11 | 0.55 | 2.07 | 9.92 | - | 2.37 | 4.91 | 0.89 |
| | POS Wang et al. (2016) | 3.90 | 9.61 | - | 5.44 | 12.00 | - | 4.05 | 8.75 | 0.78 |
| | AutoSparsePPGNowara et al. (2020) | 4.55 | 14.42 | - | - | - | - | - | - | - |
| Model-Based Unsupervised | HR-CNN Spetlik et al. (2018) | - | - | - | 1.84 | 2.37 | - | - | - | - |
| | SynRhythm Niu et al. (2018) | - | - | - | - | - | - | 5.59 | 6.82 | 0.72 |
| | CAN Chen & McDuff (2018) | 4.06 | 9.51 | - | - | - | - | - | - | - |
| | CVD Niu et al. (2020) | - | 6.04 | 0.84 | 1.29 | 2.01 | 0.98 | 2.19 | 3.12 | 0.99 |
| | PulseGAN Song et al. (2021) | - | - | - | - | - | - | 1.19 | 2.19 | 0.98 |
| | InverseCAN Nowara et al. (2021) | 2.27 | 4.90 | - | - | - | - | - | - | - |
| | DualGAN Lu et al. (2021) | - | - | - | 0.82 | 1.31 | 0.99 | **0.44** | **0.67** | **0.99** |
| | Physformer Yu et al. (2022) | 2.84 | 5.36 | - | - | - | - | - | - | - |
| | Federated Liu et al. (2022b) | 2.99 | - | 0.79 | - | - | - | 2.00 | 4.38 | 0.93 |
| | EfficientPhys-C Liu et al. (2023) | 2.91 | 5.43 | 0.86 | - | - | - | - | - | - |
| | ContrastPhys-100 (PAMI'24) Sun & Li (2024) | **1.11** | **3.83** | 0.96 | 0.48 | **0.98** | 0.99 | 0.50 | 0.84 | 0.99 |
| Data-Driven Unsupervised | Gideon Gideon & Stent (2021) | 3.98 | 9.65 | 0.85 | 2.3 | 2.9 | 0.99 | 3.60 | 4.60 | 0.95 |
| | Yue Yue et al. (2023a) | - | - | - | 1.23 | 2.01 | 0.99 | - | - | - |
| | ContrastPhys-0 Sun & Li (2024) | 1.82 | 6.69 | 0.96 | 1.00 | 1.40 | 0.99 | - | - | - |
| **Ours** Data-Driver, Supervised | RIS-iPPG | 1.97 | 3.73 | 0.97 | **0.38** | 1.28 | 0.99 | 0.47 | 0.80 | 0.98 |

## 4.5 Test-time Sampling

Our proposed method, as compared to all previous iPPG methods, is able to sample solution space at test time and generate multiple realizations of the solution. As compared to other posterior sampling methods, the benefit of using an SDE-based method is that we can visualize solutions at all time points, an example of which is shown in Appendix Figure 10. Given that our iPPG interpolant satisfies both the continuity equation and the Fokker-Planck equation from Theorem 3.1, we write a reverse SDE that, when solved, produces an estimate of the pulse signal given the measurements i.e. initial condition. First, define the sampling SDE as

$$d\mathbf{x}_t = \mathbf{b}_F(t, \mathbf{x}_t)dt + \sigma_t dW_t, \ \sigma_t = \sqrt{2\epsilon(t)} \tag{15}$$

where $W_t$ is a Wiener process. The drift $\mathbf{b}(t, \mathbf{x})$ coefficient then becomes:

$$\mathbf{b}_F(t, \mathbf{x}_t) = \left[\mathbf{v}_\theta(t, \mathbf{x}_t) - \gamma(t) \cdot \big(\frac{d}{dt}\gamma(t)\big) \cdot \mathbf{s}_\theta(t, \mathbf{x}_t)\right] + \epsilon(t)\mathbf{s}_\theta(t, \mathbf{x}_t) \tag{16}$$

See Appendix A.8 for the derivation of the drift. While any standard solver can be used to solve Equation 15, we chose to use the implementations of Li et al. (2020); Kidger et al. (2021), and as recommended by Albergo et al. (2023), we set $\epsilon(t)$ to a constant for all $t$. In the next section, we present the results of our approach, and demonstrate the efficacy of both our uncertainty quantification as well as the RCL loss for iPPG recovery.

# 5 Implementation Details and Experimental Results

## 5.1 Datasets

We evaluate our algorithm using three datasets, which are described below

- **MMSE-HR** Zhang et al. (2016); Ertugrul et al. (2019): The MMSE-HR dataset recorded facial video at 1040×1392 pixels and 25 FPS while capturing synchronized blood pressure waveforms using a Biopac NIBP100D recording at 1000Hz (which we downsampled to 25 Hz to align with the video). Seventeen male and twenty-three female subjects were asked to perform a variety of tasks that induced motion as well as a change in heart rate, which resulted in 102 videos. We train and test on 10 second time windows, and evaluate using the leave-one-subject-out evaluation protocol of Nowara et al. (2021).

- **PURE** Stricker et al. (2014): Recorded at 30 FPS and a resolution of 640×480 pixels, the PURE dataset contains 10 subjects each of whom perform six task to induce facial motion. Corresponding pulse oximetry data were captured at 60Hz, which was downsampled to 30 Hz to align with the video data. The models were trained on 10-second time windows of pulse oximeter data, and were evaluated on 30-second windows according to the splits of Špetlík et al. (2018).

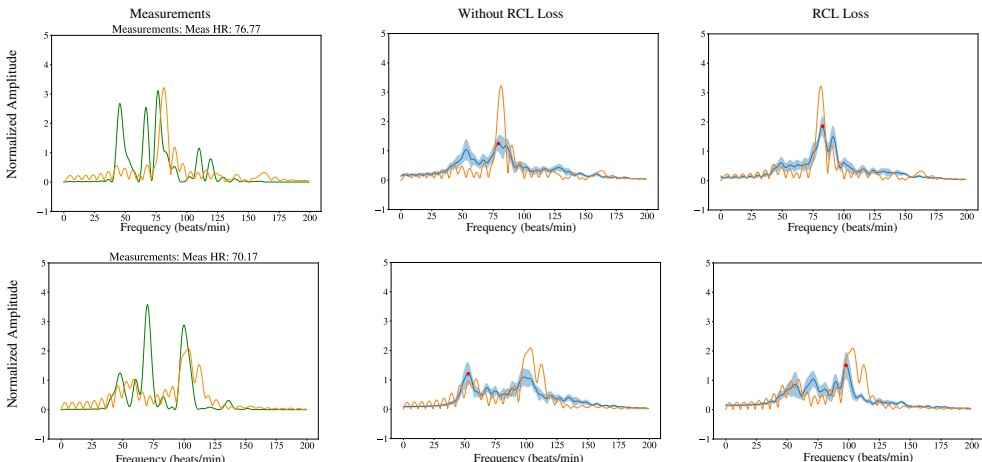

Figure 3: Qualitative results with and without the RCL loss. We plot the camera pixel measurements (green), ground-truth PPG (orange), the mean of 100 realizations of sampling (bolded blue), and 95% confidence interval of the power in each bin (light blue). Our algorithm with the RCL loss is better at predicting the true heart rate. In row 2, we see that the RCL loss also captures the modes of the measurements more effectively (i.e. the correct peak around ∼100 bpm as well as the peak of the measurements ∼75 bpm).

- **UBFC-rPPG** Bobbia et al. (2019). The UBFC-rPPG dataset contains 43 subjects, each recording one video captured at $640 \times 480$ px and 30 FPS while playing a game to induce pulse rate changes. Simultaneous pulse waves were captured using a pulse oximeter recording at 30Hz. We train on 10-second windows with a frame stride of 2.4 seconds, and evaluate according to the protocol in Lu et al. (2021): we evaluate on 10-second time windows for each video, and average all heart rates in a video for a single heart rate estimate.

## 5.2 Evaluation Metrics

**Pulse Rate Estimation Metrics**: We follow the protocol of previous work Shenoy et al. (2023) and measure the predicted pulse rate versus the ground-truth pulse rate in the windows of interest. We compute the pulse rates by first multiplying the signal by a Hanning window, followed by taking an $L = 10 \times$ signal length FFT, after which we compute the power by squaring the magnitude of the FFT. We sum the power spectra across all facial regions, and all samples from the SDE solutions, after which we chose the frequency bin with the greatest power as the pulse rate. We then compute the mean absolute error (MAE) and root mean squared error (RMSE) between the predicted and ground-truth pulse rates, as well as the Pearson Correlation Coefficient between predicted and ground-truth pulse rates as in Liu et al. (2022a):

**Spectrum Estimation Accuracy Metrics:** We measure the accuracy of spectrum estimation by comparing the predicted spectrum against that of the ground-truth spectrum. We measure the spectrum MAE and RMSE, as well as other standard regression metrics such as the coefficient of determine ($R^2$) and the Pearson Correlation Coefficient (PCC).

**Predictive Uncertainty Quantification Metrics**: The use of stochastic sampling allows us to measure predictive uncertainty. Using the implementation of Chung et al. (2021), we compare our predicted spectral magnitude against the ground-truth, and calculate the Negative Log Likelihood (NLL), the Sharpness, Continuous Ranked Probability Score, the Check Score, and the Interval Score of our predictive uncertainty. In addition, we plot the calibration curves of our method over the UBFC-rPPG and PURE test sets, as well as over protected attributes of the MMSE-HR dataset. A full description of these metrics are reserved for Appendix A.2 due to space constraints.

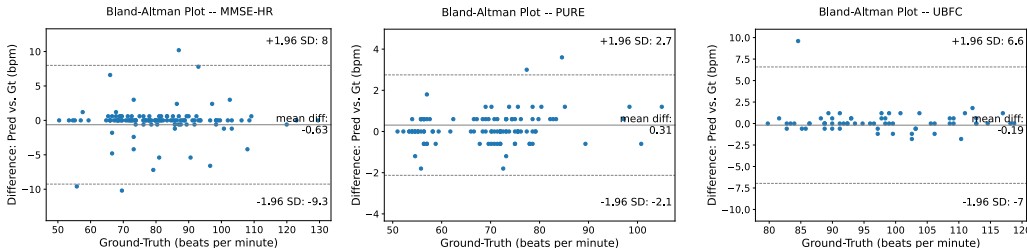

Figure 4: Bland-Altman Plots for the predicted heart rate against the ground-truth for all time windows on the test sets. We plot the ground-truth heart rate against the predicted heart rate, as well as the 95% confidence intervals. We see that the mean difference is close to zero for all three datasets, with reasonable confidence intervals.

### 5.3  Implementation Details

All computation, including preprocessing, was implemented on an A5000 NVIDIA GPU. To generate the pre-processed time-series, we first pass the input video through the OpenFace Amos et al. (2016) face detector, followed by the landmark detection using LDEQ Micaelli et al. (2023). These landmarks are interpolated across the face to delineate the right and left forehead, right and left cheek, and chin. In each region, we perform per-channel averaging of all pixels, and as in Shenoy et al. (2023), we take the ratio of the red channel to the green channel to obtain our signal estimate as in Figure 1.

Identical U-Nets, adapted from guided diffusion Dhariwal & Nichol (2021), learn both the flow and score in our framework. We train these networks using the Adam optimizer Kingma & Ba (2014) with an initial learning rate of $1e$-3. The number of training epochs varied for each dataset. Please refer to the Appendix Section A.1 for more information.

### 5.4  Results

**Pulse Rate Estimation**: Before we present UQ metrics, we must determine whether our method is accurate. We present the results of RIS-iPPG with the RCL loss in Table 2. The results for the MMSE-HR dataset are aggregated over subject-independent (i.e. 40-fold) cross-validation, while the results for PURE and UBFC-rPPG are evaluated on their test sets in accordance with previous literature. On all three datasets, we achieve very competitive performance with previous benchmarks, and achieve a new state-of-the-art on the PURE dataset. For the benchmarks in which we do not perform best, we still achieve < 1bpm error on UBFC-rPPG and < 2 bpm on MMSE-HR.

Qualitative results of the power spectrum are shown in Figure 3, where the orange signals are the ground-truth in all cases, the green signals are the measurements, and the bolded blue signals are the means from 100 realizations of our algorithm. The light blue shading represents the 95% confidence interval of the power over each frequency bin of our spectrum, and the red dot shows the peak of the spectrum. In both cases, we notice that the model with the RCL loss produces a more accurate pulse rate estimation and spectrum. The second row shows a particularly interesting example: with the RCL loss, we see greater uncertainty at the three modes of the distribution (the ground-truth frequency, the peak frequency in the measurements, and the frequency at ∼55bpm) as compared to the prediction without the RCL loss, which not only incorrectly predicts the pulse rate, but also smooths the spectrum and fails to capture the uncertainty around the peak measurement frequency. More examples on MMSE-HR, UBFC-rPPG, and PURE are included in Appendix A.10.5.

**Statistical Analysis**: In addition to a comparison against the state-of-the-art, we perform a Bland-Altman analysis to understand the performance of our algorithm. We plot the ground-truth heart rate in a window against the prediction from RIS-iPPG, and plot the 95% confidence interval for the heart rate predictions. The results from the MMSE-HR dataset show a mean difference of approximately -0.63 bpm, indicating that our method overestimates the actual heart rate by an average of over half a beat. Our heart standard

Table 3: Varying the stride $\delta$ of Equation 12 and weight $\lambda_{\text{RCL}}$ of Equation 14 for the RCL loss on the MMSE-HR Dataset. We notice best performance, on average, when using $\delta = 9$ and $\lambda_{\text{RCL}} = 0.1$

| | | Window Shift $\delta$ (seconds) | | | | | | | | | | Average |
|---|---|---|---|---|---|---|---|---|---|---|---|---|
| | | 1 | 2 | 3 | 4 | 5 | 6 | 7 | 8 | 9 | 10 | |
| | 0.0 | 3.72 | 3.72 | 3.72 | 3.72 | 3.72 | 3.72 | 3.72 | 3.72 | 3.72 | 3.72 | 3.72 |
| | 0.1 | 2.67 | 2.54 | 2.47 | 3.44 | 2.79 | 2.54 | 3.51 | 2.52 | 1.39 | 2.89 | **2.67** |
| | 0.2 | 2.64 | 3.77 | 2.42 | 2.04 | 2.77 | 4.07 | 2.82 | 1.94 | 2.52 | 2.59 | 2.74 |
| | 0.3 | 2.54 | 4.67 | 3.87 | 2.39 | 4.74 | 2.92 | 3.09 | 2.42 | 2.34 | 2.24 | 3.12 |
| RCL weight $\lambda_{\text{RCL}}$ | 0.4 | 2.59 | 2.42 | 5.97 | 3.39 | 4.19 | 2.67 | 2.69 | 3.42 | 2.12 | 2.87 | 3.23 |
| | 0.5 | 3.02 | 2.67 | 2.51 | 2.57 | 1.92 | 3.19 | 2.74 | 2.85 | 2.84 | 2.67 | 2.73 |
| | 0.6 | 3.02 | 3.01 | 2.89 | 3.07 | 2.12 | 2.77 | 2.72 | 2.47 | 3.72 | 2.34 | 2.83 |
| | 0.7 | 2.77 | 4.07 | 2.19 | 2.69 | 2.92 | 2.44 | 2.44 | 3.09 | 1.82 | 3.57 | 2.8 |
| | 0.8 | 2.67 | 2.87 | 2.84 | 2.09 | 4.57 | 3.69 | 3.22 | 2.37 | 2.14 | 2.49 | 2.89 |
| | 0.9 | 1.77 | 2.57 | 2.25 | 3.32 | 1.49 | 3.34 | 2.94 | 3.82 | 2.72 | 2.89 | 2.71 |
| | 1.0 | 2.69 | 4.17 | 3.22 | 2.34 | 2.79 | 6.54 | 3.27 | 1.62 | 1.59 | 2.19 | 3.04 |
| Average | | 2.73 | 3.34 | 3.12 | 2.82 | 3.09 | 3.44 | 3.01 | 2.74 | **2.44** | 2.79 | |

Table 4: Comparing the spectrum estimation performance and uncertainty quantification metrics with and without the RCL loss

| | Waveform Accuracy | | | | Uncertainty Metrics | | | |
|---|---|---|---|---|---|---|---|---|
| Method | MAE↓ | RMSE↓ | PCC↑ | NLL↓ | Sharpness↑ | CRPS ↓ | Check Score↓ | Interval Score↓ |
| PURE w/out RCL | 0.057 | 0.224 | 0.681 | **-3.825** | 0.201 | 0.041 | 0.021 | 0.217 |
| PURE w/RCL | **0.056** | **0.218** | **0.700** | -3.820 | **0.202** | **0.040** | **0.020** | **0.208** |
| UBFC-rPPG w/out RCL | 0.073 | 0.237 | 0.651 | -3.475 | **0.177** | 0.042 | 0.021 | 0.203 |
| UBFC-rPPG w/RCL | **0.049** | **0.185** | **0.795** | **-3.516** | 0.168 | **0.034** | **0.017** | **0.162** |

deviation is relatively large; however, this is metric is dominated by the few large outliers produced when the input measurements are too noisy. We see similar trends on both the UBFC-rPPG dataset and the PURE dataset. Models on both datasets achieve a mean difference close to zero, while maintaining single-digit standard deviations.

**Effect of overlap and weight**: We further explore the inclusion of the RCL loss by performing a grid-search over the weight parameter $\lambda_{\text{RCL}} \in [0.0, 1.0]$ and the time-shift $\delta \in [1, 10]$ seconds, with the results as shown in Table 3. Note that we train our model using 10-second windows; therefore, a stride of $\delta = 10$ corresponds to the "no-overlap" scenario. On the MMSE-HR dataset, we achieve best performance with a stride of $\delta = 9$ seconds and a weight of $\lambda = 0.1$. We also note that we can get significant improvements over traditional stochastic interpolants by including the RCL loss. This experiment was conducted on a small validation set of the MMSE-HR dataset; after selecting $\delta = 9$ seconds and a weight of $\lambda = 0.1$, we perform 40-fold cross-validation and report the final results in Table 2. Ablations on the UBFC-rPPG dataset are included in Appendix Table 9.

**Spectrum Estimation Performance**: In Table 4 we measure the spectrum estimation accuracy average *over all frequency bins* (i.e. as compared to a single heart rate) calculated for the pulse rate. As shown in Table 4, our model with the RCL loss does better than the model without the RCL loss on both the PURE and UBFC-rPPG datasets. Our RMSE, indicating the magnitude of the outliers, is lower for the model with the RCL loss, and the PCC between the predicted and ground-truth spectrum magnitude is higher. These results indicate that the RCL loss is beneficial to RIS-iPPG.

## 5.5 Predictive Uncertainty Quantification

To the best of our knowledge, RIS-iPPG is the first iPPG method to develop an stochastic sampling method for iPPG estimation, which allows us to analyze predictive uncertainty and establish new baselines. We enumerate the UQ metrics averaged over the PURE and UBFC-rPPG test sets, respectively, as shown in Table 4. The metrics were measured by computing the Fourier spectrum magnitude across all bins of the predicted signal against that of the ground-truth. We notice that over most metrics, our model with residual correlation regularization performs better than the model without it, and over the metrics on which we do not as well, our results are still comparable. The NLL values indicate that the ground-truth spectral magnitude has high likelihood given our model's estimated distribution, while the CPRS indicates that our model's predicted CDF is similar to that of the ground-truth. The check score (also known as the pinball

Table 5: Proper Scoring Rule metrics on Protected Attributes of MMSE-HR dataset

| | HR Accuracy | | | | Uncertainty Metrics | | |
|---|---|---|---|---|---|---|---|
| Protected Attributed | HR MAE↓ | HR RMSE↓ | NLL↓ | Sharpness↑ | CRPS ↓ | Check Score↓ | Interval Score↓ |
| Light Skin Tone | **2.17** | **3.74** | **-3.430** | **0.203** | 0.042 | 0.021 | 0.209 |
| Dark Skin Tone | 4.79 | 6.84 | -2.828 | 0.191 | **0.035** | **0.018** | **0.188** |
| Men | **2.07** | 3.83 | **-3.435** | 0.196 | **0.032** | **0.016** | **0.167** |
| Women | 2.23 | **3.69** | -3.415 | **0.203** | 0.039 | 0.020 | 0.195 |

loss), which measures quantile prediction performance, is lower for the RCL-regularized model, while the interval score, which balances the sharpness of the distribution versus the calibration, is lower for the model with the RCL loss.

Next, we measure uncertainty calibration Kuleshov et al. (2018), which intuitively means that when a model assigns a probability $p$ to an event, then that event should happen $100 * p$ percent empirically. We plot the calibration curves in Figure 5, which plots the predicted proportion of samples in an interval against the actual observed proportion: perfect calibration corresponds to the diagonal. For the PURE dataset, we observe that we achieve good calibration, but our model under-predicts the observed proportion, leading to a miscalibration error of 0.06. On the UBFC-rPPG dataset, we also observe good calibration performance; our model is slightly overconfident at lower observed proportions, while it is slightly overconfident at higher observed proportions. On our average, however, the models on both datasets do well.

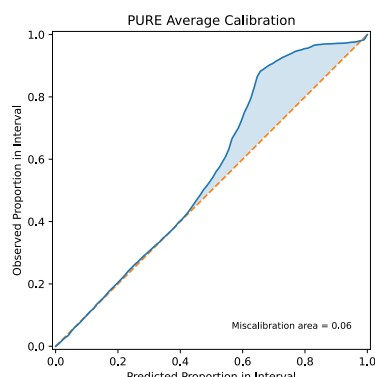

### 5.5.1 Uncertainty Quantification on Protected Attributes

In addition to calibration metrics over the entire dataset, we measure calibration metrics over protected attributes i.e. skin color and gender. We establish the first baselines for uncertainty quantification and calibration on protected attributes on the MMSE-HR dataset in Table 5. Before we establish the UQ metrics, we enumerate the pulse rate estimation metrics in Table 5. Our results on people with darker skin tones is consistent with that of previous work: pulse rate estimation is superior for those people who are better represented in the dataset. We notice better UQ performance for men, while the results between light and dark skin tones seem to be mixed; we notice that three UQ metrics indicate better results for darker skinned people that lighter skinned people. We plot calibration curves for all four subgroups in Appendix Figure 9; we notice that our miscalibration error for all four groups is 0.10 or better, showing the effectiveness of our algorithm.

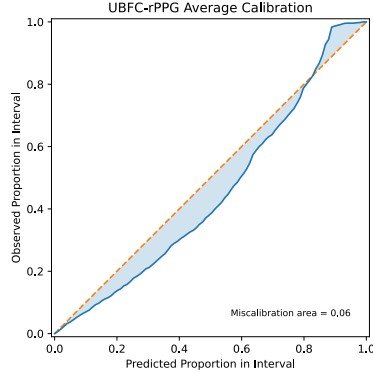

Figure 5: Calibration curves for PURE and UBFC-rPPG datasets.

## 6 Conclusion

To be truly accepted by clinicians and medical personnel, machine learning algorithms for healthcare should be "repeatedly successful in prognosticating patient's condition in [the doctor's] personal experience" Tonekaboni et al. (2019). While previous iPPG algorithms output point estimates of the pulse signal, we introduce the first posterior sampling method for iPPG that repeatedly samples likely pulse signal estimates given camera measurements, permitting an uncertainty analysis that can help doctors make better decisions. We achieve this by modeling a stochastic process between camera measurements and pulse signals, and learn the flow and score of this process to build the drift coefficient of an SDE. We improved results by temporally regularizing the flow, and show that this helps us capture the modes of the signal distribution. While we achieve strong results on intra-dataset evaluation, future work should address domain shifts between training and testing datasets, as well as addressing performance gaps on protected subpopulations.

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

Table 6: Hyperparameters used to train our model

| Parameter | MMSE-HR | PURE | UBFC-rPPG |
|---|---|---|---|
| Passband Frequency (bpm) | 42 | 42 | 42 |
| Cutoff Frequency (bpm) | 150 | 150 | 150 |
| Num Taps | 5 | 5 | 5 |
| Frame Stride (sec) | 0.4 | 0.4 | 0.4 |
| Frame Stride Test (sec) | 10 | 10 | 10 |
| FPS | 25 | 30 | 30 |
| Signal Length | 250 | 300 | 300 |
| Num Res Block | 1 | 1 | 1 |
| Attention Resolution | [2, 4] | [2, 4] | [2, 4] |
| Learning Rate | 1e-3 | 1e-3 | 1e-3 |
| Weight Decay | 0 | 0 | 0 |
| Dropout | 0 | 0 | 0 |
| Epochs | 10 | 15 | 15 |

Zijie Yue, Miaojing Shi, and Shuai Ding. Facial video-based remote physiological measurement via self-supervised learning. *IEEE Transactions on Pattern Analysis and Machine Intelligence*, 45(11), 2023a. doi: 10.1109/TPAMI.2023.3298650.

Zijie Yue, Miaojing Shi, and Shuai Ding. Facial video-based remote physiological measurement via self-supervised learning. *IEEE Transactions on Pattern Analysis and Machine Intelligence*, 2023b.

Xi Zhang, Yuan Pu, Yuki Kawamura, Andrew Loza, Yoshua Bengio, Dennis L. Shung, and Alexander Tong. Trajectory flow matching with applications to clinical time series modeling, 2025. URL `https://arxiv.org/abs/2410.21154`.

Zheng Zhang, Jeff M Girard, Yue Wu, Xing Zhang, Peng Liu, Umur Ciftci, Shaun Canavan, Michael Reale, Andy Horowitz, Huiyuan Yang, et al. Multimodal spontaneous emotion corpus for human behavior analysis. In *Proceedings of the IEEE conference on computer vision and pattern recognition*, pp. 3438–3446, 2016.

## A Appendix

### A.1 Implementation Details: Hyperparameters

As mentioned in the main paper, we implement our code in PyTorch using the PyTorch Lightning Falcon & The PyTorch Lightning team (2019) library. Our models are identical learnable UNets from Dhariwal & Nichol (2021). The hyperparameters used to train our models are in Table 6.

### A.2 Evaluation Metrics

We provide a more complete discussion of the evaluation metrics below. Let $y_{i,j}$ be the ground-truth, $\mu_{i,j}$ be the mean, and $\sigma_{i,j}$ of samples generated from test sample $i$ in frequency bin $j$.

- **Mean Absolute Error (MAE):** A measure of the difference between a predicted quantity $\hat{R}_i$ and the ground-truth quantity $R_i$. This quantity is less sensitive to outliers than the RMSE. We use this metric to measure the pulse rate estimation error as well as the spectrum estimation error. Lower values indicate better performance, and the lowest possible value is 0.

$$\text{MAE} = \frac{1}{N} \sum_{i=1}^{N} |R_i - \hat{R}_i| \qquad (17)$$

- **Root Mean Squared Error (RMSE):** A measure of the difference between a predicted quantity $\hat{R}_i$ and the ground-truth quantity $R_i$, which weights outliers heavily. We use this metric to measure the pulse rate estimation error as well as the spectrum estimation error. Lower values indicate better performance, and the lowest possible value is 0.

$$\text{RMSE} = \sqrt{\frac{1}{N}\sum_{i=1}^{N}(R_i - \hat{R}_i)^2} \tag{18}$$

- **Pearson Correlation Coefficient (PCC):** This quantity measure the linear relationship and strength of the linear relationship between two variables. We use this metric to measure the pulse rate estimation error as well as the spectrum estimation error. Higher values indicate better performance, and the highest possible score is 1.

$$\rho = \frac{\sum_{i=1}^{N}(R_i - \mu_{R_i})(\hat{R}_i - \mu_{\hat{R}_i})}{\sqrt{\sum_{i=1}^{N}(R_i - \mu_{R_i})^2(\hat{R}_i - \mu_{\hat{R}_i})^2}} \tag{19}$$

where $\hat{R}_i$ is the predicted pulse rate, $R_i$ is the ground-truth pulse rate, $N$ is the number of time windows, and $\mu_{R_i}$ and $\mu_{\hat{R}_i}$ are the means of the predicted and ground-truth pulse rates, respectively.

- **Negative Log Likelihood (NLL)** Tran et al. (2020): NLL provides an overall measure of both the predictive accuracy and quality of the predictive uncertainty quantification. Given the mean and variance of a set of predictions, construct a Gaussian distribution. Then measure the likelihood of the ground-truth given this distribution. Lower values are better, and can be negative.

$$\text{NLL}(\mathbf{y}, \mu, \sigma) = -\frac{1}{N \cdot L}\sum_{i=1}^{N}\sum_{j=1}^{L}\ln P\big(y_{i,j}|\mathcal{N}(\mu_{i,j}, \sigma_{i,j})\big) \tag{20}$$

- **Continuous Ranked Probability Score** Gneiting & Raftery (2007): This quantity measures how good the predicted distribution is compared to the ground-truth value, and is often used in meteorolgy. Lower values are better and the lowest possible value is zero. Under the assumption of the Gaussian, the CRPS is defined as:

$$\text{CRPS}(\mathbf{y}, \mu, \sigma) = \frac{1}{N \cdot L}\sum_{i=1}^{N}\sum_{j=1}^{L}\sigma_{i,j}\left[\frac{1}{\sqrt{\pi}} - 2\psi\Big(\frac{y_{i,j} - \mu_{i,j}}{\sigma_{i,j}}\Big) - \frac{y_{i,j} - \mu_{i,j}}{\sigma_{i,j}}\Big(2\Phi\Big(\frac{y_{i,j} - \mu_{i,j}}{\sigma_{i,j}}\Big) - 1\Big)\right] \tag{21}$$

where $\psi$ and $\Phi$ denote the PDF and CDF of a standard gaussian.

- **Sharpness** Tran et al. (2020): A quantity that measures the concentration of the predicted distribution, independent of the ground-truth distribution. Given the CDF of a sample $i$ in bin $j$ the sharpness is defined as

$$\text{Sharpness} = \frac{1}{N \cdot L}\sum_{i=1}^{N}\sum_{j=1}^{L}\text{var}(F_{i,j}) \tag{22}$$

Higher values indicate better performance.

- **Check Score/Pinball Loss** Steinwart & Christmann (2011): This metric measure the prediction of quantiles. This is a non-symmetric loss that penalizes larger quantiles more. Lower values are better

$$\text{Check Score/Pinball Loss} = \frac{1}{N \cdot L}\sum_{i=1}^{N}\sum_{j=1}^{Q}L(y_i, \hat{y}_i) = \begin{cases} (y_- \hat{y}_i) \cdot \tau_j & \text{if } y \geq \hat{y} \\ (y_- \hat{y}_i)(1 - \tau_j) & \text{if } y < \hat{y} \end{cases} \tag{23}$$

where $y_i$ is the true value, $\hat{y}_i$ is the predicted quantile and $\tau_j$ is the target quantile between 0 and 1. Lower values are better.

- **Interval Score** Gneiting & Raftery (2007): Given a lower bound $L$ and upper bound $U$ that is intended to cover the true value $y_{i,j}$ with probability $1 - \alpha$, the interval score is defined as

$$S_\alpha(L, U, y_{i,j}) = (U - L) + \frac{2}{\alpha}(L - y) \cdot \mathbf{1}(y < L) + \frac{2}{\alpha}(y - U) \cdot \mathbf{1}(y > U) \tag{24}$$

This measures whether the true value falls within the interval, and whether the width of the interval is narrow. Smaller values are better.

### A.3 Preliminary Investigation

### A.3.1 Inadequacy of the $A = F^{-1}$ forward model

In the main manuscript, we implement PMC-iPPG to show the inadequacy of the $A = F^{-1}$ forward model of previous work. The work of Nowara et al. (2020) makes the strong assumption that the camera pixel signal is simply the pulse signal with additive gaussian noise, which makes $A = F^{-1}$ suitable; they achieve good results because of careful signal extraction/noise removal during the time-series extraction phase, not because of the signal recovery algorithm. The work of Shenoy et al. (2023) also uses $A = F^{-1}$; however, by "unrolling" proximal gradient descent with deep denoising operators, they effectively correct an inexact forward operator. Our experiments using PMC-iPPG, which only learns the signal prior, shows that $A = F^{-1}$ is suboptimal as described in Nowara et al. (2020); Shenoy et al. (2023).

Our next step was to address this deficiency by learning the forward model Arridge et al. (2023); Lunz et al. (2020). However, we found this problem to be ill-posed for a variety of reasons as described in Appendix A.3.1 and Figure 6.

- **Stochastisity:** The mapping between the blood volume pulse signal and the camera pixel signal is stochastic and time-dependent due to unconstrained motion and its associated changes in specular and diffuse reflections. This is reflected in Figure 6, which shows the significant changes in the signal over four second time intervals.

- **One-to-many mapping:** The pixel signal from different regions of the face correspond to one blood volume pulse signal, which is captured at the finger. A deterministic function can not map a single ground-truth pulse to different facial regions. An example of this is shown in Figure 6 when comparing the pixel signals at the left and right cheek.

All these lead us to the decision to use stochastic interpolants, which *implicitly* learns the optimal transport between the distribution of camera pixel signals and BVP signals. Modeling the signal recovery process as such allows for greater flexibility, especially when accounting for the physics of the problem as well as long-tailed events such as sharp motions. By matching distributions, RIS-iPPG captures non-linear interactions that define the forward process while also allowing for uncertainty quantification, one of our primary goals.

### A.3.2 Conditioning on a guidance signal

Many previous works solve traditional imaging inverse problems via guidance. We experimented with guidance in RIS-iPPG. We reasoned that our guidance signal should tell us something about the noise when extracting signal measurements from video; therefore, we use the raw signals from the color channels of the video frames, which we assume capture motion noise via sharp changes in intensity. Then, we used the RIS framework to map our extracted signal from Section 5.3 to the ground-truth. The learned flow and score networks used the raw color channel signals as guidance when predicting flow and score. We then evaluated our learned models with guidance as in the main paper.

We compare the heart rate estimation performance using each color channel as a guidance signal, as well as using no guidance in Table 7. Clearly, the guidance signal made performance worse. However, we can not conclude that guidance signals, in general, hurt iPPG performance as we did not perform a thorough analysis of the entire design space of guidance signals. Nevertheless, we argue that a model regularized using

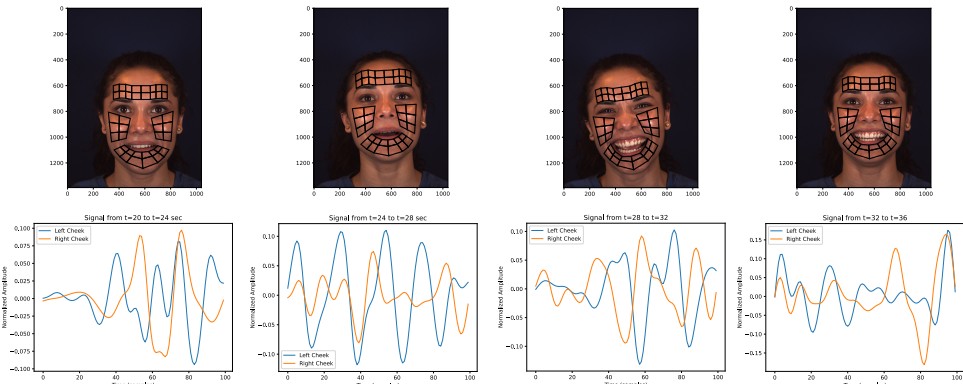

Figure 6: Change in facial position, and region detection over a short time period

the RCL loss is better as it is independent of the noise profile and focuses only on the temporal correlations of the pulse.

Table 7: Testing various guidance signals on the MMSE-HR dataset

| Guidance Signal | MAE (bpm) ↓ | RMSE (bpm) ↓ |
|---|---|---|
| Blue | 5.79 | 12.71 |
| Red | 4.17 | 11.78 |
| Green | 4.05 | 11.18 |
| No Guidance | 3.72 | 6.55 |

Table 8: Comparing Conditional VAEs and RIS-iPPG on the MMSE-HR dataset

| Method | MAE (bpm) ↓ | RMSE (bpm) ↓ |
|---|---|---|
| Conditional VAEs | 6.17 | 9.96 |
| Bayesian Neural Networks | 3.12 | 5.63 |
| RIS-iPPG | 1.97 | 3.73 |

### A.4 Comparison against other posterior sampling methods

Below we address a comparison of RIS-iPPG against more standard posterior sampling methods.

**Complexity of the Task**: While our signals are one-dimensional, the **inverse mapping** between the camera pixel signal and pulse signal is highly complex. In Figure 6, we show the how the camera pixel signal changes during sharp changes in motion every four seconds, as well as the resulting signals from two different regions of the face. These signals contain high-dimensional, non-Gaussian noise sources such as motion, specular reflections, quantization noise, etc. in the same frequency bands as the pulse. Simple models like Gaussian Processes (GP) or BNNs often assume Gaussian noise or unimodal posteriors, an assumption violated by our data distributions. RIS-iPPG can more effectively model this data.

**Empirical Analysis**: We conduct experiments to measure the performance of Conditional VAEs (cVAE) and Bayesian Neural Networks (BNN) against flow-based models. We evaluate these methods for pulse rate estimation performance in Table 8. For Conditional VAEs we adapt the work Sohn et al. (2015) and use the camera pixel signal as our conditioning element. For the BNN, we follow the guide of Jospin et al. (2022) and set the prior as recommended in the paper. Note that in Table 8, we see significant increases in pulse rate estimation error using BNNs and cVAE compared to RIS-iPPG.

**Performance of cVAE**: We notice that pulse rate estimation performance for VAEs is significantly worse than that of the flow-based models. VAEs are known to produce blurry samples due to the ELBO objective;

Table 9: Varying the RCL loss weight $\lambda_{\mathrm{RCL}}$ and the stride $\delta$ on the UBFC-rPPG Dataset

| | | Window Shift $\delta$ | | | | | | | | | | Average |
|---|---|---|---|---|---|---|---|---|---|---|---|---|
| | | 1 | 2 | 3 | 4 | 5 | 6 | 7 | 8 | 9 | 10 | |
| | 0.0 | 3.72 | 3.72 | 3.72 | 3.72 | 3.72 | 3.72 | 3.72 | 3.72 | 3.72 | 3.72 | 3.72 |
| | 0.1 | 6.71 | 3.41 | 1.19 | 4.55 | 2.79 | 3.59 | 3.05 | 7.61 | 2.39 | 3.23 | 3.85 |
| | 0.2 | 3.77 | 3.71 | 5.63 | 2.69 | 2.77 | 2.87 | 3.59 | 3.47 | 3.71 | 0.53 | 3.27 |
| | 0.3 | 1.61 | 4.01 | 2.03 | 1.79 | 4.74 | 2.03 | 1.67 | 5.09 | 5.09 | 2.39 | 3.04 |
| RCL weight $\lambda_{\mathrm{rcl}}$ | 0.4 | 2.59 | 3.35 | 2.79 | 1.07 | 4.19 | 1.67 | 2.61 | 2.69 | 2.09 | 1.07 | 2.47 |
| | 0.5 | 3.53 | 3.23 | 3.89 | 1.13 | 1.92 | 0.77 | 3.47 | 6.41 | 1.55 | 1.13 | 2.70 |
| | 0.6 | 3.91 | 2.95 | 2.45 | 0.59 | 2.12 | 3.85 | 3.65 | 2.69 | 0.77 | 0.48 | **2.44** |
| | 0.7 | 5.27 | 1.73 | 2.57 | 1.73 | 2.92 | 2.09 | 2.44 | 4.01 | 2.45 | 3.71 | 2.89 |
| | 0.8 | 3.53 | 1.97 | 0.83 | 2.81 | 4.57 | 1.97 | 2.51 | 2.21 | 3.35 | 3.77 | 2.75 |
| | 0.9 | 2.75 | 4.79 | 3.77 | 2.21 | 1.49 | 1.85 | 1.85 | 2.45 | 1.67 | 2.51 | 2.53 |
| | 1.0 | 2.63 | 4.31 | 5.27 | 2.21 | 2.79 | 3.71 | 4.91 | 2.21 | 1.61 | 2.45 | 3.21 |
| Average | | 3.63 | 3.37 | 3.10 | 2.31 | 3.09 | 2.55 | 3.04 | 3.86 | 2.58 | **2.27** | |

for a task like ours in which we are capturing waveform morphology, VAEs will smoothe features such as the systolic peak resulting in inaccurate pulse rate estimation. Additionally, previous work has noted that VAEs tend to ignore latent variables if the decoder is overly powerful in sequence modeling tasks Chen et al. (2016); Serban et al. (2017); Bowman et al. (2016). It is also well-known that VAEs suffer from posterior collapse Dai et al. (2020); Lucas et al. (2019) . Finally, VAEs often show poorer quality of samples as well Xu et al. (2024).

**Performance of BNNs**: Bayesian neural networks also have their own challenges. One of the biggest challenges is carefully defining the prior distribution over the weights of the BNN Jospin et al. (2022). Previous work has shown that the performance of BNNs are greatly affected by the choice of prior Silvestro & Andermann (2020), and often the choice of distribution over the weights is non-intuitive for the selected task. It is unclear how the choice of prior over the weights translate to a prior over signal morphology. Furthermore, inference methods in BNNs often have drawbacks: MCMC methods are exact but do not scale well to large models while variational methods are inexact and only focus on a single mode of the posterior Jospin et al. (2022). RIS-iPPG, without explicity setting a prior, effectively samples waveforms while capturing multiple modes of the posterior.

**Performance of GPs**: Gaussian Processes (GPs) models suffer from computational complexity concerns, scaling cubicly with the size of the dataset Eriksson et al. (2018). Additionally, the choice of kernel in GPs often affects the generalization ability of the model Rasmussen (1997); Sun et al. (2018); as in BNNs, prior knowledge about the kernel with reference to the task is critical for successful application of BNNs. Often, these kernels fail to capture the complex and non-stationary nature of motion artifacts.

**Benefit of RIS-iPPG**: Flow-based diffusion models solve many of the problems of VAEs, BNNs, and GPs. We can model complex data distributions, capture distinct waveform morphologies, model highly non-Gaussian and non-stationary noise, and capture multimodal posteriors. Given the recent research showing state-of-the-art performance and widespread adoption of flow-based diffusion models, it is critical that we address research in iPPG with respect to such models.

## A.5 Inference Speed of RIS-iPPG versus other posterior sampling methods

We measure the runtime performance at inference for three sampling methods in Table 10. At inference, BNNs are best in terms of speed; cVAEs and RIS-iPPG are about the same. However, the models in RIS-iPPG need to be evaluated for every timsetp; depending on the granularity with which the SDE steps are discretized, the cost of RIS-iPPG can be much higher. Note that we do not claim RIS-iPPG to be superior in terms of runtime; however, fast inference is important for real-world adoption and future work should aim to improve inference speed.

Table 10: Comparing inference speed of different methods. All were tested on a single NVIDIA A5000 GPU

| Method | Inference Time (ms) |
|---|---|
| Conditional VAE | 845.06 |
| Bayesian Neural Networks | 473.91 |
| RIS-iPPG | 850.97 |

## A.6    Deriving the Denoiser

In Section 4.3, we defined the score as $\mathbf{s}(t, \mathbf{x}) = -\mathbf{n_z}(t, \mathbf{x})/\gamma(t)$. We show how this is derived below.

Define our stochastic interpolant as before:

$$\mathbf{x}_t = \underbrace{I(t, \mathbf{x}_0, \mathbf{x}_1)}_{\mathbf{x}'} + \gamma(t)\mathbf{z}, \ \text{where } \mathbf{z} \sim \mathcal{N}(0, Id), t \in [0, 1] \tag{25}$$

The distribution of $\mathbf{x}_t$ is $p(\mathbf{x}_t) \sim \mathcal{N}(\mathbf{x}_t | \mathbf{x}', \gamma^2(t))$. Find the gradient of the log of this distribution with respect to $\mathbf{x}_t$ is then given by

$$\begin{aligned}
\nabla_\mathbf{x} \log p(\mathbf{x}_t) &= \nabla_\mathbf{x} \log \frac{1}{(\sqrt{2\pi\gamma^2(t)})^d} \exp\left(\frac{-||\mathbf{x}_t - \mathbf{x}'||^2}{2\gamma^2(t)}\right) \\
&= \nabla_\mathbf{x}\left(\frac{-||\mathbf{x}_t - \mathbf{x}'||^2}{2\gamma^2(t)} - \log\left(\sqrt{2\pi\gamma^2(t)}\right)^d\right) \\
&= -\frac{\mathbf{x}_t - \mathbf{x}}{\gamma^2(t)} = -\frac{\mathbf{z}}{\gamma(t)}
\end{aligned} \tag{26}$$

This is the claim from Section 4.3.

## A.7    Intuition Behind the RCL loss

We regularize our models by minimizing the RCL loss, or one-minus the normalized correlation coefficient of the error residuals between time-shifted versions of the training signal. The intuition behind this is based on the slowly-varying nature of physiological signals: in short time-windows barring an acute stressor, human physiology remains nearly constant. Therefore, in our model, the error between the predicted flows in overlapping time windows should be nearly identical. We show this through the following theorem:

**Theorem A.1.** *The RCL loss of Equation 13 is minimized when the error residuals point in the same direction.*

*Proof.* As in Section 4.4, define two pairs of training data as

$$(\mathbf{x}_0(i), \mathbf{x}_1(i)) \text{ and } (\mathbf{x}_0' = \mathbf{x}_0(i - \delta), \mathbf{x}_1' = \mathbf{x}_1(i - \delta)) \tag{27}$$

As described in Section 4.3 we construct two flow targets $\mathbf{v}(t, \mathbf{x}_t)$ and $\mathbf{v}(t, \mathbf{x}_t')$ which are predicted by our flow model. With $i$ as the time index, let $\mathbf{p} = \mathbf{v}(t, \mathbf{x}_t(i)) - \mathbf{v}_\theta(t, \mathbf{x}_t(i))$ and $\mathbf{q} = \mathbf{v}(t, \mathbf{x}_t(i - \delta)) - \mathbf{v}_\theta(t, \mathbf{x}_t(i - \delta))$: we seek to align these vectors by aligning these vector by minimizing the RCL loss

$$\begin{aligned}
\min_\theta \mathcal{L}_{\mathrm{RCL}}(\mathbf{p}, \mathbf{q}) &= \min_\theta \left[1 - \frac{T \cdot \mathbf{p}^\top \mathbf{q} - \mu_p \mu_q}{\sqrt{(T \cdot \mathbf{p}^\top \mathbf{p} - \mu_p^2)(T \cdot \mathbf{q}^\top \mathbf{q} - \mu_q^2)}}\right] \\
&= \max_\theta \frac{T \cdot \mathbf{p}^\top \mathbf{q} - \mu_p \mu_q}{\sqrt{(T \cdot \mathbf{p}^\top \mathbf{p} - \mu_p^2)(T \cdot \mathbf{q}^\top \mathbf{q} - \mu_q^2)}}
\end{aligned} \tag{28}$$

We can simplify this optimization problem by first recognizing that $T$ is a scaling factor that is independent of the flow. We can always zero-mean and $L_2$ normalize the signals $\mathbf{p}$ and $\mathbf{q}$ before computing the maximization above. Assume that the normalized error residuals for $\mathbf{p}$ and $\mathbf{q}$ are $\mathbf{y}(i)$ and $\mathbf{y}(i - \delta)$ since we assume a time-shift in the signals results in a time-shift in the residuals. This is achieved through using convolutional operators, which is the basis upon which the flow prediction model is built. Then

$$
\begin{aligned}
\max_\theta \mathbf{y}(i)^\top \mathbf{y}(i - \delta) &= \max_\theta ||\mathbf{y}(i)|| \cdot ||\mathbf{y}(i - \delta)|| \cdot \cos(\phi) \\
&= \max_\theta \cos(\phi)
\end{aligned}
\tag{29}
$$

where $\phi$ is the angle between the vectors $\mathbf{y}(i)$ and $\mathbf{y}(i - \delta)$. This value is maximized when $\phi = 0$, or the vectors point in the same direction. $\qquad\square$

$\hfill\square$

## A.8  Deriving the Sampling SDE

In Theorem 3.1, we state that the Fokker-Planck equations satisfy the continuity theorem. We show that this is the case (replicated and simplified from Albergo et al. (2023):

**Theorem A.2.** (Theorem 2.6 of Albergo et al. (2023)) The probability distribution of the interpolant $x_t$ defined in Equation equation 7 is absolutely continuous with respect to the Lebesgue measure at times $t \in [0, 1]$ and solves the transport equation

$$
\frac{\partial}{\partial t} p_t + \nabla \cdot (\mathbf{b} p) = 0
\tag{30}
$$

In addition, the forward and backward Fokker-Planck equations are satisfied

$$
\frac{\partial}{\partial t} p_t + \nabla \cdot (\mathbf{b}_F p) = \epsilon(t)\Delta p, \mathbf{b}_F = \mathbf{b}(t, \mathbf{x}) + \epsilon(t)\mathbf{s}(t, \mathbf{x})
\tag{31}
$$

$$
\frac{\partial}{\partial t} p_t + \nabla \cdot (\mathbf{b}_B p) = -\epsilon(t)\Delta p, \mathbf{b}_B = \mathbf{b}(t, \mathbf{x}) - \epsilon(t)\mathbf{s}(t, \mathbf{x})
\tag{32}
$$

where $\epsilon(t)$ is some noise schedule.

*Proof.* We would like to show that the drift coefficient satisfies the Continuity Equation in Equation equation 30. Consider the diffusion term from Equation 31. From SDE theory, we know that we can write the Laplacian of the distribution $p$ as:

$$
\epsilon(t)\Delta p = \epsilon(t)\nabla \cdot (p\nabla \log p) = \epsilon(t)\nabla \cdot (s(t, \mathbf{x})p)
\tag{33}
$$

Now, we would like to show that the drift coefficient satisfies the continuity equation. Substituting the above result into Equation equation 31, we have

$$
\begin{aligned}
\frac{\partial}{\partial t} p_t + \nabla \cdot (\mathbf{b}_F p) &= \epsilon(t)\Delta p \\
\frac{\partial}{\partial t} p_t + \nabla \cdot ((\mathbf{b}(t, \mathbf{x}) + \epsilon(t)\mathbf{s}(t, \mathbf{x}))p) &= \epsilon(t)\nabla \cdot (ps(t, \mathbf{x})) \\
\frac{\partial}{\partial t} p_t + \nabla \cdot ((\mathbf{b}(t, \mathbf{x})p) + \epsilon(t) \ \nabla \cdot (\mathbf{s}(t, \mathbf{x})p) &= \epsilon(t)\nabla \cdot (\mathbf{s}(t, \mathbf{x})p) \\
\frac{\partial}{\partial t} p_t + \nabla \cdot ((\mathbf{b}(t, \mathbf{x})p) &= 0
\end{aligned}
\tag{34}
$$

The derivation is similar for Equation 32. □

We now seek to derive the equation for the drift coefficient from Equation 16. The sampling SDE is defined as

$$dx_t = \mathbf{b}_F(t, \mathbf{x}_t)dt + \sigma_t dW_t, \ \sigma_t = \sqrt{2\epsilon(t)} \tag{35}$$

and the drift coefficient $\mathbf{b}_F(t, \mathbf{x}_t)$ is defined as

$$
\begin{aligned}
\mathbf{b}_F &= \mathbf{b}(t, \mathbf{x}) + \epsilon(t)\mathbf{s}(t, \mathbf{x}) \\
\mathbf{b}_F &= \mathbb{E}[\frac{\partial}{\partial t}\big(I(\mathbf{x}_0, \mathbf{x}_1, t) + \gamma(t)\mathbf{z}\big)] + \epsilon(t)\mathbf{s}(t, \mathbf{x}) \\
\mathbf{b}_F &= \mathbb{E}[\frac{\partial}{\partial t}I(\mathbf{x}_0, \mathbf{x}_1, t) + (\frac{\partial}{\partial t}\gamma(t))\frac{-\mathbf{n_z}(t, \mathbf{x})}{\gamma(t)}] + \epsilon(t)\mathbf{s}(t, \mathbf{x}) \\
\mathbf{b}_F &= \mathbf{v}_\theta(t, \mathbf{x}) - \big(\frac{\partial}{\partial t}\gamma(t)\big)\gamma(t)\mathbf{n_z}(t, \mathbf{x})\big) + \epsilon(t)\mathbf{s}(t, \mathbf{x})
\end{aligned}
\tag{36}
$$

The first equality restates the drift equation from Equation 31. In the second equality, we replace substitute the definition of our iPPG interpolant for $\mathbf{b}(t, \mathbf{x})$. In the third equality, we take the derivative with respect to each term, and replace $\mathbf{z}$ with our denoiser $\mathbf{s}(t, \mathbf{x}) = \frac{-\mathbf{n_z}(t, \mathbf{x})}{\gamma(t)}$. We simplify terms in the last equation.

□

## A.9  Stability, Data Requirements, and Approximation Quality

The work of Albergo et al. (2023) discusses this in detail in Sections 2.4 and 2.5, with proofs of likelihood control, density estimation, and cross-entropy calculations in the appendix. Demonstrating these properties in practice is important. We do so as follows:

- **Stability and Convergence:** To show our stability and convergence, in Appendix A.8 we plot the training and validation loss curves for the flow, score, and RCL components in Figure 7. We see that both training and validation curves decrease monotonically, confirming that we are successfully approximating our intended drift coefficient. We also provide a proof of stability in Theorem A.3.

- **Data Requirements:** The datasets we used recorded at least one minute of video per subject at at least 25 FPS, with many subjects recording multiple videos under stimuli. This provides adequate data to sample the manifold of physiological signals under varying levels of noise.

- **Approximation Quality:** Our Gauge R&R analysis, shown in Appendix A.10.6 and Figure 14, quantifies the repeatability and precision of our system. The variance due to repeatability is negligible (1.7%) compared to the variance between different signal frequencies ("Part"). This indicates high-fidelity approximation.

- **Robustness and Generalization via RCL:**: Regarding generalization, the Residual Correlation Loss (RCL) acts as an inductive bias (or smoothness prior). By constraining the solution manifold to signals that are consistent across time shifts, we theoretically reduce the hypothesis space to physically plausible signals. This improves robustness against high-frequency outliers (e.g., sudden motion artifacts) that violate this correlation structure. We have also added training/validation curves in Figure 7 to show that our model learns the flow and score correctly.

- **Error Composition (Approximation vs. Discretization):** We model the total sampling error as the sum of the *network approximation error* and the *solver discretization error*. As detailed in Equation 37, the error bound is given by:

$$\text{Total Error} \le \mathbb{E}\left[\int_0^1 \|b_F(t, x_t) - b_\theta(t, x_t)\|dt\right] + \mathcal{O}(\Delta t) \tag{37}$$

The first term represents the training quality (how well our neural network $b_\theta$ approximates the true vector field $b_F$). The second term, $\mathcal{O}(\Delta t)$, represents the error introduced by the SDE solver steps. This confirms that the error is additive and controllable via training convergence (Figure 7) and step-size selection.

We also show the stability of our system through a proof below.

**Theorem A.3.** Let $\hat{\mathbf{x}}_1$ be a camera measurement and $\tilde{\mathbf{x}}_1 = \hat{\mathbf{x}}_1 + \delta$ be a perturbed measurement, where $\|\delta\| < \infty$. Let $\hat{\mathbf{x}}_0$ and $\tilde{\mathbf{x}}_0$ be generated by the same Wiener process $W_t$. If the learned drift coefficient $\mathbf{b}_{F,\theta}(t, \mathbf{x})$ is Lipschitz continuous with constant $L$, then the reconstruction error is bounded by:

$$\|\hat{\mathbf{x}}_0 - \tilde{\mathbf{x}}_0\| \le e^L \|\delta\| \tag{38}$$

The system is bounded-input, bounded-output (BIBO) stable.

*Proof.* Define the reverse time SDE for both processes as:

$$\begin{aligned} d\hat{\mathbf{x}}_t &= \mathbf{b}_{F,\theta}(t, \hat{\mathbf{x}}_t)\, dt + \sigma_t\, dW_t \\ d\tilde{\mathbf{x}}_t &= \mathbf{b}_{F,\theta}(t, \tilde{\mathbf{x}}_t)\, dt + \sigma_t\, dW_t \end{aligned} \tag{39}$$

Subtracting these equations, the noise terms cancel out, and we get the ODE for the difference:

$$\frac{d}{dt}(\hat{\mathbf{x}}_t - \tilde{\mathbf{x}}_t) = \mathbf{b}_{F,\theta}(t, \hat{\mathbf{x}}_t) - \mathbf{b}_{F,\theta}(t, \tilde{\mathbf{x}}_t) \tag{40}$$

Now, let the squared error between our two solutions be $u(t) = \|\hat{\mathbf{x}}_t - \tilde{\mathbf{x}}_t\|^2$. By the Chain rule, we have:

$$\begin{aligned} \frac{d}{dt}u(t) &= \frac{d}{dt}\langle \hat{\mathbf{x}}_t - \tilde{\mathbf{x}}_t, \hat{\mathbf{x}}_t - \tilde{\mathbf{x}}_t \rangle \\ &= 2\left\langle \hat{\mathbf{x}}_t - \tilde{\mathbf{x}}_t, \frac{d}{dt}(\hat{\mathbf{x}}_t - \tilde{\mathbf{x}}_t) \right\rangle \\ &= 2\langle \hat{\mathbf{x}}_t - \tilde{\mathbf{x}}_t, \mathbf{b}_{F,\theta}(t, \hat{\mathbf{x}}_t) - \mathbf{b}_{F,\theta}(t, \tilde{\mathbf{x}}_t) \rangle \\ &\le 2\|\hat{\mathbf{x}}_t - \tilde{\mathbf{x}}_t\| \cdot L\|\hat{\mathbf{x}}_t - \tilde{\mathbf{x}}_t\| \\ &= 2Lu(t) \end{aligned} \tag{41}$$

In the inequality step, we used the Cauchy-Schwarz inequality and the Lipschitz condition. Now, applying the differential form of Gronwall's inequality backwards from $t = 1$ to $t = 0$, we get:

$$\begin{aligned} u(0) &\le u(1)e^{2L(1-0)} \\ \|\hat{\mathbf{x}}_0 - \tilde{\mathbf{x}}_0\|^2 &\le e^{2L}\|\hat{\mathbf{x}}_1 - \tilde{\mathbf{x}}_1\|^2 \\ \|\hat{\mathbf{x}}_0 - \tilde{\mathbf{x}}_0\| &\le e^L\|\hat{\mathbf{x}}_1 - \tilde{\mathbf{x}}_1\| \end{aligned} \tag{42}$$

This concludes the proof showing BIBO Stability. $\qquad\square$

## A.10 Additional Quantitative/Qualitative results

### A.10.1 RCL loss between predicted flows versus the error residuals

We investigate whether impose temporal regularization on the predicted flows from adjacent time windows as compared to the error residuals from adjacent time windows results in equivalent or better performance. We hypothesize that the RCL on predicted flows would be suboptimal: this would encourage the two time-shifted versions of the signal to align (i.e. resulting in a time-shift $\delta = 0$) despite the fact that we know *a priori* that they should be consistent, but not aligned. Therefore, we decided to align the error residuals instead of the predicted flows themselves.

To verify our intuition, we trained our model with RCL regularization between predicted flows instead of error residuals. In Figure 8, we plot the validation loss of the RCL with flows versus the validation loss with error residuals. It is apparent that the validation error decreases typically for the RCL loss with error residuals; using predicted flows, however, the loss oscillates and the network does not learn. Therefore, we use the RCL regularization with error residuals.

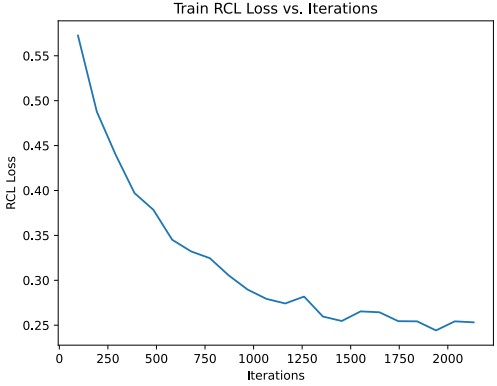

(a) RCL training loss vs training iterations

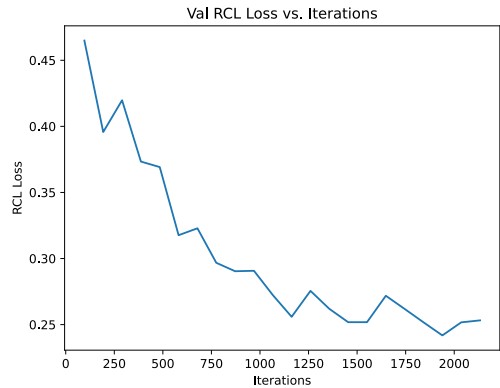

(b) RCL validation loss vs training iterations

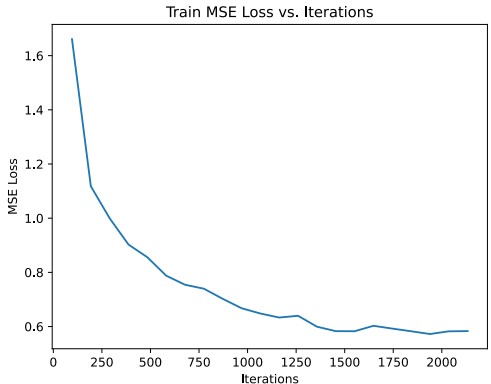

(c) Flow+Score+RCL training loss vs training iterations

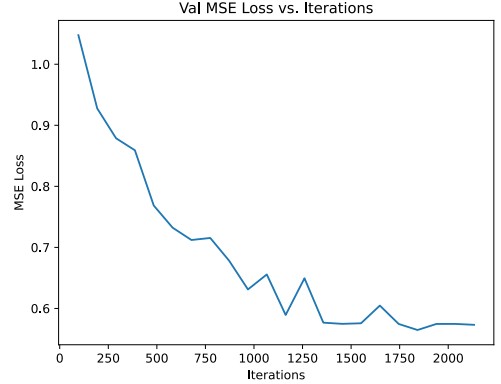

(d) Flow+Score+RCL validation loss vs training iterations

Figure 7: The training and validation loss curves for the RCL loss and the entire loss (flow loss + score loss + RCL loss of Equation 14)

Table 11: Cross-dataset model evaluation. In some scenarios (e.g. MMSE-HR → UBFC-rPPG), we achieve good performance, but in other cross-dataset experiments, our method fails.

| Train | Test | MAE (bpm) ↓ | RMSE (bpm) ↓ |
|---|---|---|---|
| MMSE-HR | PURE | 15.52 | 27.26 |
| | UBFC-RPPG | 1.73 | 4.67 |
| PURE | MMSE-HR | 7.93 | 15.19 |
| | UBFC-rPPG | 21.59 | 33.49 |
| UBFC-rPPG | MMSE-HR | 6.59 | 12.26 |
| | PURE | 0.26 | 0.53 |

### A.10.2 Inclusion of RCL Loss: Effect of overlap and weight

Similarly to Table 3, we perform an ablation study on the stride $\delta$ and the the weight $\lambda_{\mathrm{RCL}}$ during model training. Ablations on the UBFC-rPPG dataset are included in Table 9.

### A.10.3 Cross-dataset evaluation

We evaluate the model's ability to generalize to unseen data. In the experiments shown in Table 11, the columns "Train" indicates the dataset on which our flow and score models are trained, while "Test" is the

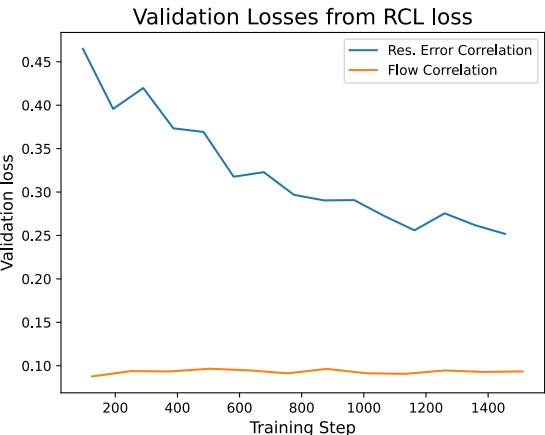

Figure 8: Validation loss when using the RCL loss with predicted flows versus error residuals. The netork learns when using error residuals, but does not learn when using predicted flows.

dataset on which these models are evaluated. Once again, we report the heart rate estimation metrics from before. We note that in some scenarios, for example training on MMSE-HR and testing on UBFC-rPPG, we perform well with an average error of 1.73 beats per minute. This suggests that the domain of the MMSE-HR dataset (video recording conditions, motion, etc.) is statistically similar to the test set of UBFC-rPPG. However, our experimentation show that this is not always the case; for example training with the PURE dataset but testing on either the MMSE-HR or UBFC-rPPG datasets shows poor performance. A visual inspection of the PURE dataset, as compared to the MMSE-HR and PURE datasets, shows significantly different lighting, as well as limited and controlled motion. Both the MMSE-HR and UBFC-rPPG datasets contain significantly more unconstrained motion—with tasks specifically designed to induce such motion— indicating a significant domain shift in the data.

We surmise that this degradation is due to the learned implicit forward model. We implicitly learn a stochastic function $A(\cdot)$ and noise in Equation 6, conditioned on two distributions. When performing cross dataset evaluation, we replace one distribution (source) with another (target), resulting in large errors. One possible solution could be to project the target domain samples into the source domain and re-run the SDE. Addressing the domain shift issues is important, which we leave for follow-up work.

### A.10.4   Calibration Curves on Protected Attributes

Given the relative accuracy of RIS-iPPG, we now explore UQ metrics on both protected subgroups. We notice that for most metrics, UQ is better for light skin tones as compared to dark skin tones (and for the other metrics the light and dark skin tone metrics are nearly identical. The NLL for light skin tones, however, is significantly better than that of the dark skin tones, indicating that the model fails to learn the distribution of pulse in darker skin tones. The UQ metrics for men are generally better than those for women, though similarly to the pulse rate estimation results, the metrics are quite similar.

In addition to the quantitative metrics, we plot the calibration curves in Figure 9 on protected attributes. On the dark skin tones, our method is over confident when the observed (i.e. ground-truth) proportion is low, but when observing a higher proportion our model is underconfident. For light skin tones, men, and women, our model is consistently underconfident at predicting the spectral magnitude in each frequency bin. In all cases, however, our calibration error is at or lower 0.10, with the largest error at 0.10.

### A.10.5   Additional Qualitative Results

We plot additional qualitative results on all datasets in Figures 11, 12, 13. The orange signals are the ground-truth in all cases, the green signals are the measurements, and the bolded blue signals are the means from

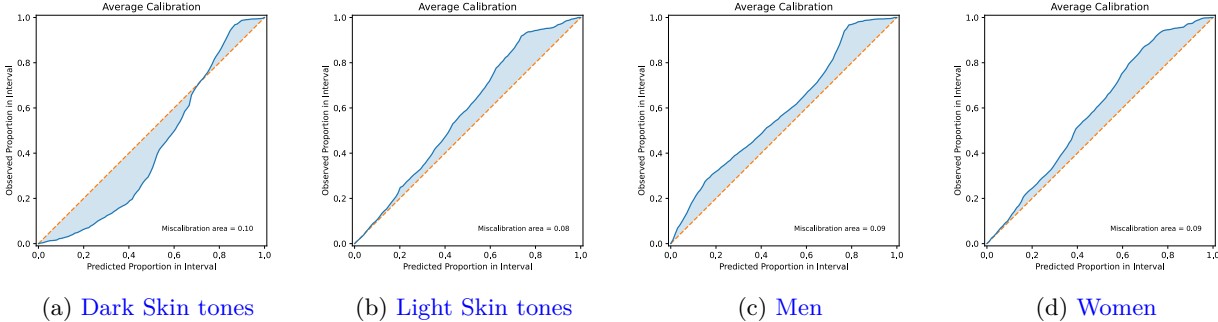

(a) Dark Skin tones      (b) Light Skin tones      (c) Men      (d) Women

Figure 9: Average calibration curves for the four protected attributes on the MMSE-HR dataset. In all four scenarios our method performs well; however, our method is relatively worse on dark skin tones.

100 realizations of our algorithm. The light blue shading represents the 95% confidence interval of the power over each frequency bin of our spectrum. In Figure 11, the first and second rows indicate that our algorithm is able to regress a measurement with an incorrect heart rate to the correct heart rate while attenuating the power of unrelated frequencies. Furthermore, we capture two modes of our distribution: in the first row of examples, the RCL loss helps us capture the mode at the measurements as well as the mode at the ground-truth heart rate. The second row shows similar results, with greater uncertainty around the measurements when the RCL loss is included. In the third row, we see that the pulse rate prediction is nearly identical with and without the RCL loss; however, the absence of the RCL loss encourages significant uncertainty at many frequencies above the predicted heart rate, while the RCL loss encourages higher uncertainty only around the harmonic. We see similar results on the UBFC-rPPG and PURE dataset in Figure 12 and 13.

### A.10.6 Sample-Level Gauge R&R Results

While the previous metrics capture population level metrics, we revisit the quote from Tonekaboni et al. (2019) regarding the need for tools that are "repeatedly successful in prognosticating a patient's condition in [the doctor's] personal experience", and provide tools to answer the question: can we trust the results? To answer this, we borrow techniques from industrial mathematics and perform an ANOVA Gauge repeatability and reproducibility (R&R) test, which quantifies the amount of variability in the samples due to the measurement system itself, and determines whether the measurement system itself is acceptable. This test quantifies the precision of the system (as compared to accuracy, which is presented Section 5.4). In traditional industrial mathematics, a gauge (for example, a lathe) drills multiple sized holes (parts) multiple times (samples) by a multiple people (operators), after which the diameter of the hole is measured. The diameters are compared against each other to quantify the variance in the process (lathe) and the measurement system themselves, after which metrics such as repeatability, reproducibility, part-to-part variation, and more are quantified to determine whether a system is acceptable.

We adopt this analysis by considering our "gauge" to be the stochastic algorithm, the "parts" to be the frequency bins below 200bpm of the magnitude of the Fourier transform of our solution, the "samples" to be the number of samples we generate from our SDE, and "operators" to be the different facial regions from which we measure signals for a single subject. Since this test operates on a single test example, we chose a sample which we know to be accurate (via heart rate absolute error), compute Gauge R&R test, and display the results in Figure 14. We first state that our estimate is *accurate*; the camera measurements themselves were close to the ground-truth, and the solution denoised this accurate spectrum. Our Gauge R&R analysis then analyzed the precision of our system, and sources of variation. The table in Figure 14 shows to which metric we can attribute the majority of our variation; clearly, the largest variation is from the part-to-part variation, while the smallest variation comes from repeatability (i.e. measuring the same frequency bin multiple times) versus the reproducibility (i.e. different facial regions agreeing on the power in the same frequency bin). Given that the largest variation is in the part-to-part model, we conclude that the precision of our system is sufficient.

### A.11 Reconstruction error vs HR error

There are some scenarios in which the HR error and the reconstruction error diverge. We show an example in Figure 15 in which the reconstruction error is better without the RCL loss, but the heart rate error is lower with the RCL loss. The model with the RCL loss produces higher error and standard deviation, even though on average the heart rate prediction is better.

We show a scenario in Figure 16 in which our normalized error is high even though our reconstruction is quite good. We plot our signal and reconstruction in the first row. In the bottom row we plot the absolute error between the mean signal and ground-truth, the standard deviation of the power measured across each frequency bin for all samples, the normalized error across each frequency bin, and the bins which have a normalized error greater and less than as in Sun et al. (2024). We see significant normalized error because our signal is very confident in reconstruction (low standard deviation) relative to the absolute error, which causes the normalized error to explode (which happens similarly in the computation of Equation **??**). One of the reasons this happens is because of the ground-truth: the ground-truth signal is captured at the *finger* while we reconstruct the signal from the *face*. While these signals are very similar, they are not the same, resulting in error. In fact, we do not want to reconstruct the finger signal exactly—we want to reconstruct the signal from the face. Ideally we would capture ground-truth PPG signal from the face, but given the data collection constraints, collecting finger PPG is the best option.

Figure 10: Inference via solving an SDE. We plot the signal measurements as well as the SDE solution at $\Delta t = 0.1$ time steps in the range $t \in [0, 1]$.

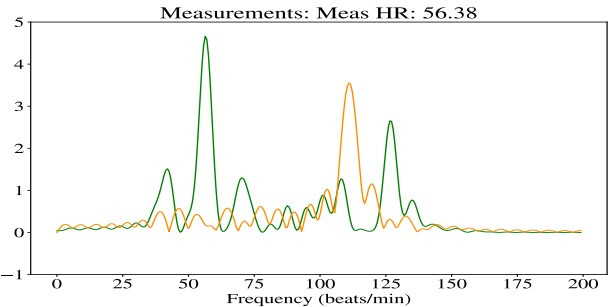

(a) The original signal measurements (green) and ground-truth (orange).

(b) $t = 0.1$

(c) $t = 0.2$

(d) $t = 0.3$

(e) $t = 0.4$

(f) $t = 0.5$

(g) $t = 0.6$

(h) $t = 0.7$

(i) $t = 0.8$

(j) $t = 0.9$

(k) $t = 1.0$

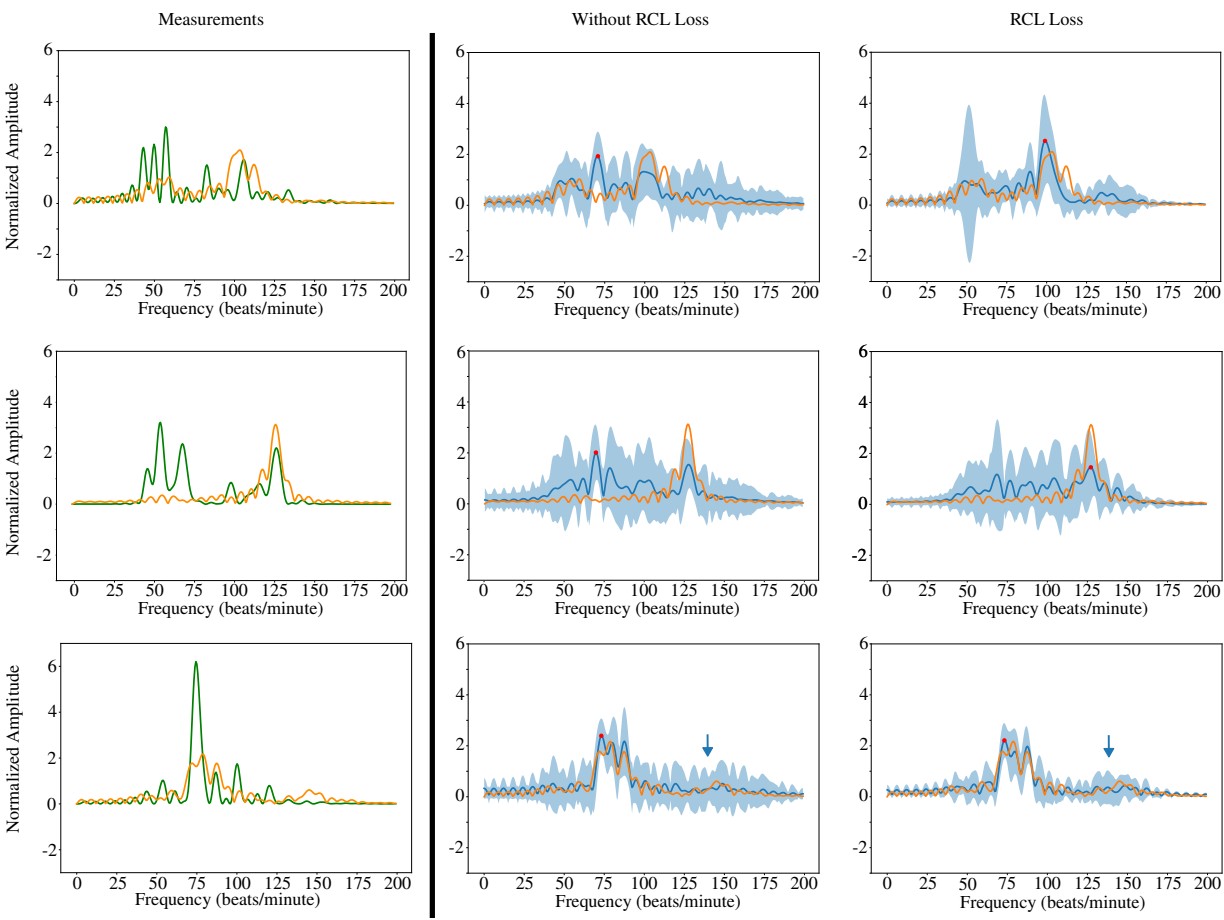

Figure 11: Qualitative results with and without the RCL loss. We plot the camera pixel measurements (green), ground-truth PPG (orange), the mean of 100 realizations of sampling (bolded blue), and 95% confidence interval of the power in each bin (light blue). Our algorithm with the RCL loss is able to predict the modes of the distribution (first two rows), while also limiting uncertainty except in the frequency bins around the harmonic (light blue arrow, third row).

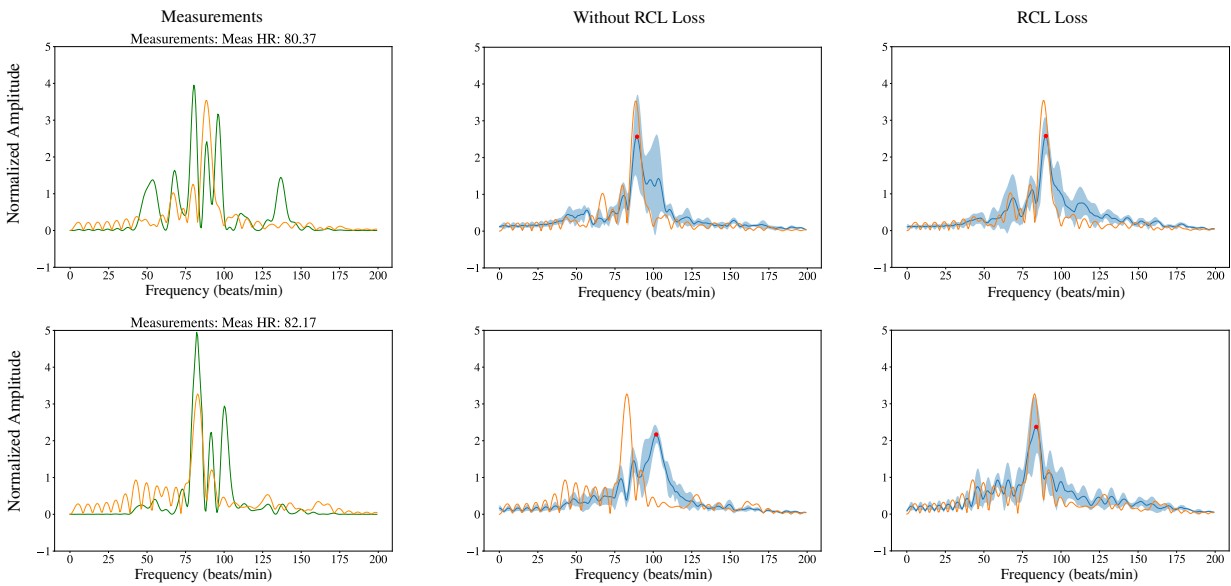

Figure 12: Results on the UBFC-rPPG dataset. The orange signals are the ground-truth, while the green signals are the measurements. The heavy blue signals are the means of 100 measurements. Models with regularization show significantly less variance in prediction.

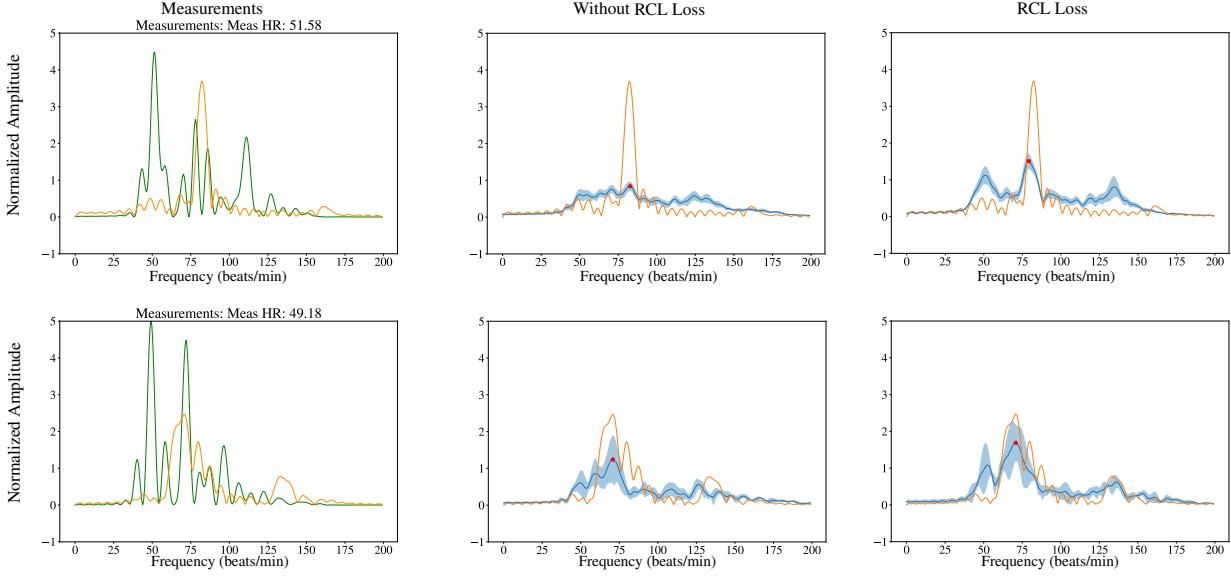

Figure 13: Results on the PURE dataset. The orange signals are the ground-truth, while the green signals are the measurements. The heavy blue signals are the means of 100 measurements. We notice similar performance with and without the RCL loss

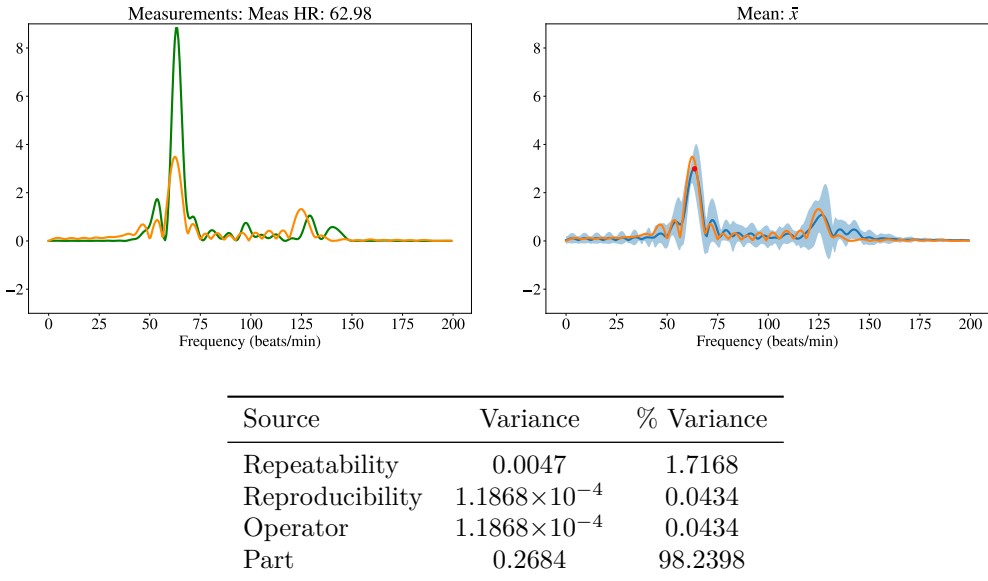

| Source | Variance | % Variance |
|---|---|---|
| Repeatability | 0.0047 | 1.7168 |
| Reproducibility | $1.1868\times10^{-4}$ | 0.0434 |
| Operator | $1.1868\times10^{-4}$ | 0.0434 |
| Part | 0.2684 | 98.2398 |

Figure 14: The measurements (green), ground-truth(orange), and mean signal and confidence intervals (heavy blue and light blue). The Table shows the Gauge R&R test for precision.

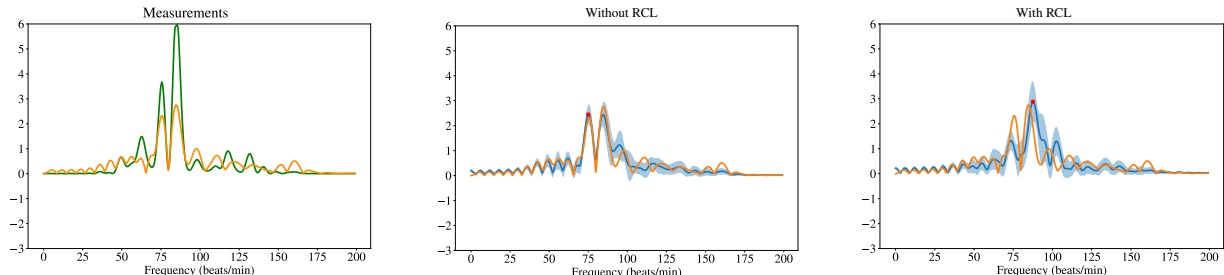

Figure 15: The reconstruction error is better without the RCL loss, but the heart rate estimation error is worse. This scenario is not necessarily uncommon. Without the RCL loss the network is overconfident in a wrong heart rate prediction.

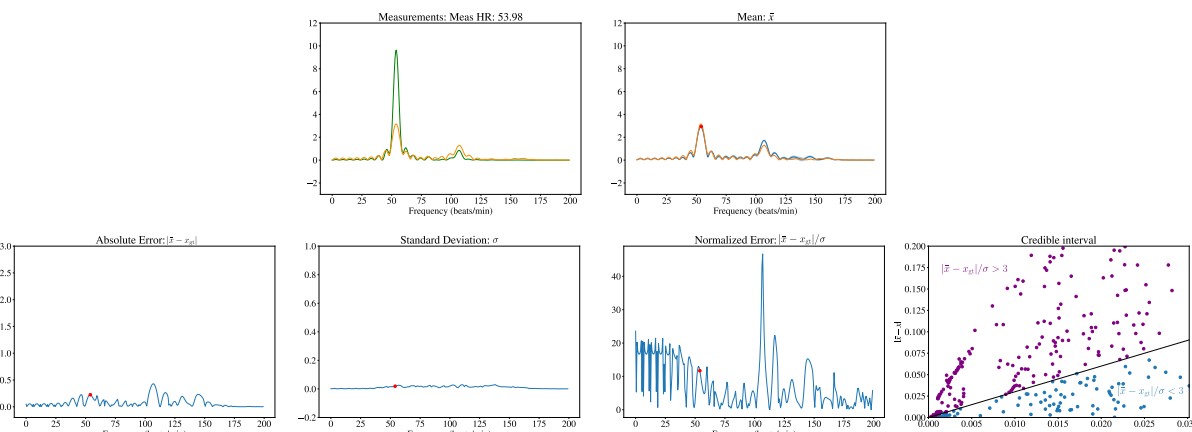

Figure 16: We plot the Measurements and reconstruction in the top row, and also plot the absolute error of the mean signal against the ground-truth and the standard deviation of the power in each frequency bin across all samples. We use these quantities to predict the normalized error, and plot the frequency bins in which the normalized error is greater and less than 3. Given that the ground-truth PPG signal is just an estimate of the pulse from face, we can see large normalized error during inference even though our reconstruction is nearly correct.

# Revision Letter

We thank the reviewers for their comments. We have made significant changes to the manuscript by adding new experiments and figures. We have also updated the text significantly. We address the reviewer comments below.

---

# Reviewer uiSP

**Reviewer Point P 1.1** —

- *The physiological and imaging processes are collapsed into a black-box forward model $A(x_0)$, without modeling or estimating its structure. This oversimplification makes the inverse recovery problem severely underconstrained and undermines the argument that the problem formulation closely reflects the physical generation process.*

- *The comparison with Plug-and-Play Monte Carlo (PMC) methods is unconvincing. The forward model is approximated as $A = F^{-1}$, based on tenuous links to prior work. The poor performance of PMC-iPPG could stem from this unrealistic assumption rather than limitations of the sampling approach itself. Negative results are presented without deeper analysis of what failed (e.g., model mismatch, training instability).*

- *The assumption $A = F^{-1}$ is weakly motivated and likely inaccurate. Explain where this approximation comes from and what its limitations are.*

- *Rather than introducing $A(\cdot)$ as a black box, provide some physical intuition or approximation, e.g., color channel mixing, temporal filtering, based on known properties of the iPPG pipeline.*

**Response**:
  We thank the reviewer for their insightful critique. Our goal when proposing PMC-iPPG was to explicitly show the drawbacks of using a simple forward model such as $A = F^{-1}$ as proposed in [1, 2]. The work of [1] makes the strong assumption that the camera pixel signal is simply the pulse signal with additive gaussian noise, which makes $A = F^{-1}$ suitable; they achieve good results because of careful signal extraction/noise removal during the time-series extraction phase, not because of the signal recovery algorithm. The work of [2] also uses $A = F^{-1}$ ; however, by "unrolling" proximal gradient descent with deep denoising operators, they effectively correct an inexact forward operator. Our experiments using PMC-iPPG, which only learns the signal prior, shows that $A = F^{-1}$ is suboptimal as described in [1, 2].
  Our next step was to address this deficiency by learning the forward model [3, 4]. However, we found this problem to be ill-posed for a variety of reasons as described in Appendix A.3.1 and Figure 6.

- **Stochastiscity:** The mapping between the blood volume pulse signal and the camera pixel signal is stochastic and time-dependent due to unconstrained motion and its associated changes in specular and diffuse reflections. This is reflected in Figure 6, which shows the significant changes in the signal over four second time intervals.

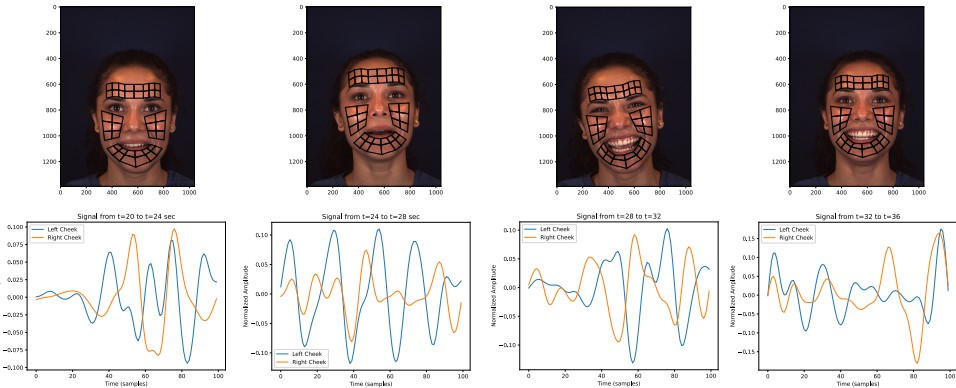

Figure 6: Change in facial position, and region detection over a short time period

- **One-to-many mapping:** The pixel signal from different regions of the face correspond to one blood volume pulse signal, which is captured at the finger. A deterministic function can not map a single ground-truth pulse to different facial regions. An example of this is shown in Figure 6 when comparing the pixel signals at the left and right cheek.

All these lead us to the decision to use stochastic interpolants, which *implicitly* learns the optimal transport between the distribution of camera pixel signals and BVP signals. Modeling the signal recovery process as such allows for greater flexibility, especially when accounting for the physics of the problem as well as long-tailed events such as sharp motions. By matching distributions, RIS-iPPG captures non-linear interactions that define the forward process while also allowing for uncertainty quantification, one of our primary goals.

**Action:** We have re-written Section 4.2 of the main paper, added Appendix A.3.1, and included Figure 6 to explain our rationale.

**Reviewer Point P 1.2** — *The authors emphasize that "no previous algorithm samples the solution space... at test time", but this is misleading. Several Bayesian deep learning and ensemble-based approaches do provide uncertainty estimates in medical AI contexts.*

**Response**:

Thanks for pointing this out. We clarify that we are the first iPPG method to perform posterior sampling for pulse signal recovery. While there are many other types of posterior sampling methods that can be used, *no previous literature has actually done so for iPPG*, and it is important that researchers address this task [5].

We acknowledge that Bayesian methods are well-established in the broader Medical AI literature, especially for diagnostic classification. However, distinct from these tasks, RIS-iPPG samples high-dimensional signal trajectories (waveforms) rather than single scalar values. We tried using Bayesian Neural Networks and Conditional VAEs for pulse rate estimation as shown in Table VIII but achieved poor result. A full discussion of these methods is reserved for Reviewer Point 1.3.

**Action:** We have updated the manuscript to reflect the distinctions for iPPG, added Appendix A.4 and Table VIII, and clarified that our method samples waveform.

**Reviewer Point P 1.3** —

- *The paper introduces a complex stochastic interpolant-based model to map between distributions of camera and BVP signals, but it doesn't justify why this approach is necessary or preferable over simpler conditional generative models (e.g., conditional VAEs, Bayesian neural nets). The task is relatively low-dimensional and structured, raising questions about whether such a high-capacity method is warranted.*

- *Justify the use of stochastic interpolants over simpler alternatives: The task is low-dimensional and could potentially be solved with VAEs, Gaussian Processes, or Bayesian regression. Explain why this method is uniquely suited.*

- *The choice to model pulse recovery as an inverse problem solved via SDEs and stochastic interpolation is intriguing, but also possibly excessive. The fundamental signal recovery task is low-dimensional and highly structured, often one-dimensional (heart rate signal) with relatively predictable dynamics. Why is a high-capacity generative model like a diffusion process necessary? The paper lacks a justification of why this level of complexity is needed over more straightforward probabilistic models (e.g. variational methods, Gaussian processes).*

**Response**:

Table VIII: Comparing Conditional VAEs and RIS-iPPG on the MMSE-HR dataset

| Method | MAE (bpm) ↓ | RMSE (bpm) ↓ |
|---|---|---|
| Conditional VAE | 6.17 | 9.96 |
| Bayesian Neural Networks | 3.12 | 5.63 |
| RIS-iPPG | 1.97 | 3.73 |

Thanks for this valuable critique. We address each point below.

**Empirical Analysis**: We conduct experiments to measure the performance of Conditional VAEs (cVAE) and Bayesian Neural Networks (BNN) against flow-based models. We evaluate these methods for pulse rate estimation performance in Table VIII. For Conditional VAEs we adapt the work [6] and use the camera pixel signal as our conditioning element. For the BNN, we follow the guide of [7] and set the prior as recommended in the paper. Note that in Table VIII, we see significant increases in pulse rate estimation error using BNNs and cVAE compared to RIS-iPPG.

**Complexity of the Task**: While our signals are one-dimensional, the **inverse mapping** between the camera pixel signal and pulse signal is highly complex. In Figure 6, we show the how the camera pixel signal changes during sharp changes in motion every four seconds, as well as the resulting signals from two different regions of the face. These signals contain high-dimensional, non-Gaussian noise sources such as motion, specular reflections, quantization noise, etc. in the same frequency bands as the pulse. Simple models like Gaussian Processes (GP) or BNNs often assume Gaussian noise or unimodal posteriors, an assumption violated by our data distributions. RIS-iPPG can more effectively model this data.

**Performance of cVAE**: We notice that pulse rate estimation performance for VAEs is significantly worse than that of the flow-based models. VAEs are known to produce blurry samples due to the ELBO objective; for a task like ours in which we are capturing waveform morphology, VAEs will smoothe features such as the systolic peak resulting in inaccurate pulse rate estimation. Additionally, previous

work has noted that VAEs tend to ignore latent variables if the decoder is overly powerful in sequence modeling tasks [8, 9, 10]. It is also well-known that VAEs suffer from posterior collapse [11, 12]. Finally, VAEs often show poorer quality of samples as well [13].

**Performance of BNNs**: Bayesian neural networks also have their own challenges. One of the biggest challenges is carefully defining the prior distribution over the weights of the BNN [7]. Previous work has shown that the performance of BNNs are greatly affected by the choice of prior [14], and often the choice of distribution over the weights is non-intuitive for the selected task. It is unclear how the choice of prior over the weights translate to a prior over signal morphology. Furthermore, inference methods in BNNs often have drawbacks: MCMC methods are exact but do not scale well to large models while variational methods are inexact and only focus on a single mode of the posterior [7]. RIS-iPPG, without explicity setting a prior, effectively samples waveforms while capturing multiple modes of the posterior.

**Performance of GPs**: Gaussian Processes (GPs) models suffer from computational complexity concerns, scaling cubically with the size of the dataset [15]. Additionally, the choice of kernel in GPs often affects the generalization ability of the model [16, 17]; as in BNNs, prior knowledge about the kernel with reference to the task is critical for successful application of BNNs. Often, these kernels fail to capture the complex and non-stationary nature of motion artifacts.

**Benefit of RIS-iPPG**: Flow-based diffusion models solve many of the problems of VAEs, BNNs, and GPs. We can model complex data distributions, capture distinct waveform morphologies, model highly non-Gaussian and non-stationary noise, and capture multimodal posteriors. Given the recent research showing state-of-the-art performance and widespread adoption of flow-based diffusion models, it is critical that we address research in iPPG with respect to such models.

**Action:** We have added this analysis to Appendix A.4, and included Table VIII in Appendix A.4.

## Reviewer Point P 1.4 —

- *The claim that test-time sampling solves the trust problem in clinical adoption is oversimplified. Sampling from a posterior doesn't inherently provide trustworthy or calibrated uncertainty. The authors do not evaluate uncertainty calibration, nor do they explain how clinicians would interpret or use the uncertainty estimates in practice.*

- *Moreover, just sampling from a posterior (via diffusion or otherwise) is not inherently more clinically trustworthy unless uncertainty calibration and interpretability are clearly demonstrated. The authors equate sampling with trustworthiness, which oversimplifies the issue.*

**Response**:
Thank you for this feedback; we have significantly improved the paper following this comment. We have extensively re-written the analysis of RIS-iPPG, and have included **five UQ metrics** for predictive uncertainty quantification on which evaluate our algorithm. We have also included **calibration curves**, both on the original test sets of PURE and UBFC-rPPG, but also on protected attributes (gender, skin color) on the MMSE-HR dataset. We have established the first baselines for uncertainty quantification, upon which we hope others build.

**How do this address trust?**: We argue that the above additions help improve trustworthiness in the following ways:

Table IV: Comparing the spectrum estimation performance and uncertainty quantification metrics with and without the RCL loss

| Method | Waveform Accuracy | | | | | Uncertainty Metrics | | | |
| | MAE↓ | RMSE↓ | PCC↑ | NLL↓ | Sharpness↑ | CRPS↓ | Check Score↓ | Interval Score↓ |
|---|---|---|---|---|---|---|---|---|
| PURE w/out RCL | 0.057 | 0.224 | 0.681 | **-3.825** | 0.201 | 0.041 | 0.021 | 0.217 |
| PURE w/RCL | **0.056** | **0.218** | **0.700** | -3.820 | **0.202** | **0.040** | **0.020** | **0.208** |
| UBFC-rPPG w/out RCL | 0.073 | 0.237 | 0.651 | -3.475 | **0.177** | 0.042 | 0.021 | 0.203 |
| UBFC-rPPG w/RCL | **0.049** | **0.185** | **0.795** | **-3.516** | 0.168 | **0.034** | **0.017** | **0.162** |

Table V: Proper Scoring Rule metrics on Protected Attributes of MMSE-HR dataset

| Protected Attributed | HR Accuracy | | | Uncertainty Metrics | | | |
| | HR MAE↓ | HR RMSE↓ | NLL↓ | Sharpness↑ | CRPS↓ | Check Score↓ | Interval Score↓ |
|---|---|---|---|---|---|---|---|
| Light Skin Tone | **2.17** | **3.74** | **-3.430** | **0.203** | 0.042 | 0.021 | 0.209 |
| Dark Skin Tone | 4.79 | 6.84 | -2.828 | 0.191 | **0.035** | **0.018** | **0.188** |
| Men | **2.07** | 3.83 | **-3.435** | 0.196 | **0.032** | **0.016** | **0.167** |
| Women | 2.23 | **3.69** | -3.415 | **0.203** | 0.039 | 0.020 | 0.195 |

- **Calibration as a Proxy for Trustworthiness**: A clinician who uses the algorithm expects that a model's confidence would match its real-world accuracy. Calibration curves in Figures 5 and 9, in which the our model predictions are close to the diagonal, show that confidence is well correlated with model accuracy (and not confidently wrong, for example).

- **Capturing Multimodal posteriors** We see evidence that our model predicts multiple posteriors instead of averaging the spectrum (i.e. Appendix A.9.5, Figure 13), capturing not only the true peak, but also other peaks and the harmonic

- **Quantifying Bias for Skin Tone Equity**: We have added Table V, which shows the quantified uncertainty for protected attributes. This is critical for clinicians as they determine for whom the model is most reliable.

Below we summarize the changes:

- Added Tables IV and V enumerating UQ metrics over the test sets of UBFC-rPPG and PURE, and the protected attributes of MMSE-HR.

- Added Figures 5 and 9, which are 6 calibration curves evaluated on different subsets of the data.

- Re-wrote much of Section 5 to describe the new UQ metrics and analysis of uncertainty quantification.

**Action:** We have added Table IV, Table V, Figure 5, Figure 9, and re-wrote much of Section 5 to address this comment.

**Reviewer Point P 1.5** — *Learning accurate scores and flow fields is central to the method but is only briefly described. Issues such as stability, data requirements, or approximation quality are not discussed.*

**Response**:

We agree that these are important; the work of [18] discusses this in detail in Sections 2.4 and 2.5, with proofs of likelihood control, density estimation, and cross-entropy calculations in the appendix. We agree that demonstrating these properties in practice is important. We do so as follows:

- **Stability and Convergence:** To show our stability and convergence, in Appendix A.8 we plot the training and validation loss curves for the flow, score, and RCL components in Figure 7. We see that both training and validation curves decrease monotonically, confirming that we are successfully approximating our intended drift coefficient.

- **Data Requirements:** The datasets we used recorded at least one minute of video per subject at at least 25 FPS, with many subjects recording multiple videos under stimuli. This provides adequate data to sample the manifold of physiological signals under varying levels of noise.

- **Approximation Quality:** Our Gauge R&R analysis, shown in Appendix A.9.6, quantifies the repeatability and precision of our system. The variance due to repeatability is negligible (1.7%) compared to the variance between different signal frequencies ("Part"). This indicates high-fidelity approximation.

**Action:** We have added Figure 7 to Appendix A.8, and included a Gauge R&R analysis to Appendix A.9.6 to address these comments.

## Reviewer Point P 1.6 —

- *Section3 - "Our goal is to link two arbitrary distributions...": While technically this is true in theory, in practice the method assumes paired samples (e.g., $(x_0, x_1)$) and the ability to model conditional relationships. The term "arbitrary" might mislead readers into thinking the method can connect any two unstructured, unpaired distributions, which is not practical. Rephrase to something like: "Our goal is to link two related data distributions, such as camera pixel signals and physiological signals, by learning a continuous stochastic interpolation between them.*

- *Section 3 - "...interpolates from one distribution's data point to the another distribution's data point." This phrase is awkward and technically imprecise. It sounds like you're connecting two specific data points, not distributions. Clarify with: "...interpolates from a sample from one distribution to a corresponding sample from another."*

- *Several key equations, including Eq. 1, 7, and 12, are introduced abruptly or without sufficient motivation.*

**Response**:
Thank you for the suggestions. We have directly made the changes as requested in the first two bullet points. We have also re-written the text describing Equations 1, 7, and 12 to be more clear.

- **Equation 1:** "The function $I(t, \mathbf{x}_0, \mathbf{x}_1)$ smoothly interpolates between the points $\mathbf{x}_0$ and $\mathbf{x}_1$ as a function of $t$ such that $\mathbf{x}_{t=0} = \mathbf{x}_0$ and $\mathbf{x}_{t=1} = \mathbf{x}_1$, while $\gamma(t)\mathbf{z}$ is an added noise term. To bridge between these points, we would like to know the instantaneous change of $\mathbf{x}_t$, which if learned for all $t$, would enable us to transport $\mathbf{x}_1$ to $\mathbf{x}_0$ or vice versa via an SDE."

- **Equation 7:** "We seek to define a a stochastic process that smoothly interpolates between samples from two distributions, namely the camera intensity signals $\mathbf{x}_1 \sim p_1$ and its ground-truth pulse signal $\mathbf{x}_0 \sim p_0$.

$$\mathbf{x}_t = (1-t)\mathbf{x}_0 + t\mathbf{x}_1 + \sqrt{2t(1-t)}\mathbf{z} \tag{1}$$

This choice of $I(t, \mathbf{x}_0, \mathbf{x}_1) = (1-t)\mathbf{x}_0 + t\mathbf{x}_1$ and $\gamma(t) = \sqrt{2t(1-t)}$ as in Equation 7 ensures that at $\mathbf{x}_{t=0} = \mathbf{x}_0$ and $\mathbf{x}_{t=1} = \mathbf{x}_1$. Using this interpolant, our goal is to build a drift coefficient that satisfies Theorem 1. We are looking for the instantaneous change of Equation 1: therefore, we are seeking $\mathbf{b}(t, \mathbf{x}) = \mathbb{E}[\frac{\partial}{\partial t}\big((1-t)\mathbf{x}_0 + t\mathbf{x}_1 + \sqrt{2t(1-t)}\mathbf{z}\big)]$"

- **Equation 12:** "Our goal is to learn the flow, the score, and the correlation; the MSE losses for the score and flow [18] are added to our proposed correlation loss. The flow and score can be learned via simple $L_2$ minimization. The final loss is given by ..."

**Action:** We have made the requested change.

**Reviewer Point P 1.7** — *Clarify what $x_0$ actually represents: Is it a temporal waveform? A spatial BVP map? A latent signal? Be precise about its dimensionality and nature.*

**Response**:

We have clarified this. The following text is added: Given a video signal of time $T$, we extract time-domain pixel intensity signals from five regions of the face to generate $\mathbf{x}_1 \in \mathbb{R}^{T \times 5}$ as in Figure 1; the ground-truth BVP signal is repeated five times to generate $\mathbf{x}_0 \in \mathbb{R}^{T \times 5}$. Broadcasting the signal in such a way allows for mapping the heterogeneous signal $\mathbf{x}_1$ to the homogeneous signal $\mathbf{x}_0$, effectively using spatial redundancy to improve the signal-to-noise ratio.

**Action:** We have added this text to the manuscript.

**Reviewer Point P 1.8** —

- *Break down Equation 12 step-by-step: It's a key part of the method, but is difficult to follow without prior exposure. Help the reader understand how the drift term is derived and what each component does.*

- *The decomposition of the drift term and use of Tweedie's formula are technical and important, but underexplained.*

**Response**:

We have added a derivation of the drift coefficient as well as the score to the Appendix. In Appendix A.5 we show how to derive the denoiser, which is more stable than to learn than the score. In Appendix A.7, we showed that the drift coefficient is composed of a flow and a score, which are learned through two separate loss terms.

**Action:** We have added a derivation of the drift coefficient to Appendix A.5 and A.7.

**Reviewer Point P 1.9** — *The idea of correlation of residuals is intuitive, but the connection between residual alignment and better physiological signal recovery should be backed by either theoretical insight or empirical correlation (e.g., does lower RCL correspond to better waveform accuracy?).*

**Response**:

    Our driving principle behind the RCL loss is that physiological signals such as pulse, under normal conditions, are slowly varying. The evidence for this claim is based on clinical experimentation, which is detailed in [19]. To constrain our network in such a way, we must ensure that residuals point in the same direction. We have included a discussion of this in Appendix A.6.

    To understand how our model benefits from the RCL loss, we plot the results of our algorithm with and without the RCL loss in Figure 3, as well as the quantitative metrics in Tables 3 and 4. Table 3 shows an HR estimation performance improvement when including the RCL loss. Waveform RMSE decreased from 0.224 to 0.218 and from 0.237 to 0.185 on the PURE and UBFC-rPPG datasets respectively when including the RCL loss, as shown in Table IV. We include these new results to strengthen our claims. Also note that in the submission manuscript, we had included Figures 3, 12, and 13 qualitatively showing how signal estimation improves with the RCL loss; we keep these figures.

**Action:** We have added text to Section 4.4 providing the intuition as to why signals overlapping time window should be similar. Section 5.4 quantitatively shows how the RCL improves performance. We included a further discussion in Appendix A.6. In submission manuscript, we had included Figures 3, 12, and 13 qualitatively showing how signal estimation is improves with the RCL loss; we keep these figures.

**Reviewer Point P 1.10** — *The paper does not establish that current lack of test-time sampling is the main bottleneck to clinical adoption. Real-world clinical challenges (e.g. robustness to skin tone variation, ethical concerns, lack of regulation) are arguably more pressing than uncertainty estimates. I would like the authors to comment on this.*

**Response**:

    We agree with the reviewer that challenges such as skin tone variation are important. However, we argue that uncertainty quantification and model evaluation on skin tone variation are closely related, especially for a clinician who has to make a decision whether to trust the model. To address the point, we include Table V in the main paper, which measures uncertainty on protected attributes such as gender and skin tone. We also include calibration curves for these protected attributes in Figure 9. The results in Table V show a performance gap between skin tones but the calibration curves in Figure 9 show decent performance. We hope that these new baselines will initiate more research in both skin tones variation and uncertainty quantification for iPPG.

**Action:** We have added text to the introduction that better explains why uncertainty quantification is important. We have also added Table 5 to the text to show the efficacy of our algorithm on skin tone and gender.

**Reviewer Point P 1.11** — *Section 4.4: How are overlapping windows selected? How much overlap is used? Is the method robust to window size and stride?*

**Response**:

    We had performed a grid search (using the validation set) over the the values of $\delta$ and $\lambda_{\text{RCL}}$ to find the best options. On the MMSE-HR dataset, we found that $\lambda_{\text{RCL}} = 0.1$ and $\delta = 9$ seconds worked best. Note that there are a variety of combinations of $\lambda_{\text{RCL}}$ and $\delta$ that show good performance; however, setting $\lambda_{\text{RCL}} = 0$ is suboptimal.

---

**Action:** We have added this ablation study to Section A.5.1

# Reviewer VHhU

**Reviewer Point P 2.1** — The work is reasonably clearly written but it is challenging to understand the data structure. Section 4.1 starts with some irrelevant sentences about the pulmonary system, but does not specify what form the data comes in $(x_1, x_0)$ pairs I assume? Multiple time points? Dimensionality?

**Response**: Thank you for pointing this out. We believe that the details about the pulmonary system are important as they introduce the physics behind the measurement process in Equation 6. To more precisely describe our algorithmic inputs, we add the following sentences to Section 4.2: We begin by generating $(\mathbf{x}_0, \mathbf{x}_1)$ pairs from the data. Given a video signal of time $T$, we extract time-domain pixel intensity signals from five regions of the face to generate $\mathbf{x}_1 \in \mathbb{R}^{T \times 5}$ as in Figure 1; the corresponding ground-truth BVP signal measured from the finger is repeated five times to generate $\mathbf{x}_0 \in \mathbb{R}^{T \times 5}$, resulting in paired data $(\mathbf{x}_0, \mathbf{x}_1)$.

**Action:** We add the above text to Section 4.2.

**Reviewer Point P 2.2** — The authors mention that inverting the mapping A is challenging. Why not just learn the iPPG → BVP mapping directly then?

**Response**: Learning the iPPG → BVP map is exactly what RIS-iPPG is doing, but based on reviewer comments, we did not write it clearly enough. Please also see our response to Reviewer Point P1.1. The goal of the preliminary investigation in Section 4.2 is precisely define the limitations of previous methods [1, 2]. We show that the approximation to the $A = F^{-1}$ is imprecise and often implicitly corrected. This implicit correction is highly non-linear, however; therefore, the formulation of previous work [2, 1] is at best imprecise and likely misleading. To see if we can correct previous works' formulation, we attempt to learn a forward model, the experiments of which are in Appendix A.3.1. This attempt turned out to be unsuccessful for a variety of reasons, which we describe in Appendix A.3.1.

That leads us to the original conclusion: a method to learn the mapping directly is likely a more effective solution. Thus, we developed RIS-iPPG.

**Action:** We updated Section 4.2 and added new experiments to Appendix A.3.1.

**Reviewer Point P 2.3** — The temporal regularization feels like a bit of a hack compared to incorporating the temporal component into the model. Would it be too computationally expensive to do?

**Response**: Thank you for the question. We considered incorporating explicit temporal components (such as RNNs, LSTMs) into the model architecture, but decided against it for two reasons:

- **Inference Efficiency (Solving the SDE)**: Our method relies on solving an SDE iteratively at test time. This requires multiple function evaluations of the network. If we were to incorporate heavy temporal layers (self-attention over long sequences, which are not necessary) into the

backbone, the computational cost the SDE solver would increase significantly. By keeping the backbone "stateless" and enforcing temporal consistency via the RCL loss instead, we maintain a low inference cost.

- **Training Stability**: Learning accurate flow and score vector fields for diffusion is already a complex optimization task. Introducing recurrent dynamics can often lead to training instabilities (e.g., vanishing/exploding gradients) or significantly increased memory usage during backpropagation. The RCL is a training-time only constraint.

In summary, the RCL allows us to achieve the desired physiological consistency (as shown in the improved metrics in Table 4) without the heavy computational penalty of a temporal architecture.

**Action:** None

**Reviewer Point P 2.4** — Why is the regularization on the residual and not the BVP signal itself?

**Response**: This is a great question. If we applied the regularization to the network outputs itself, it would encourage the network to ignore the phase shift i.e. $v(t) \approx v(t - \delta)$. However, we know *a priori* that the signals **should not** be aligned, only that they should be consistent. By aligning the error residuals, we are saying that the structure of the error should be consistent, which is to be expected for a BVP signal over a short time period.

As a sanity check, we explored the use of the RCL loss on the flow predictions, and ran experiments as discussed in Appendix Section A.9.1. As expected, the model struggled to converge, and resulted in weak performance.

**Action:** We have added an experiment to address this in Appendix Section A.8.1

**Reviewer Point P 2.5** — Why would the dimensions of the iPPG and BVP data be the same? Is the dimension just 1 in fact?

**Response**: Stochastic interpolants are convex combinations of signals in high dimensional space; for this to be a valid interpolant, all signals must lie in the space high dimensional space. We chose the spatial dimension to be 5 to capture 5 regions of the face. We do this under the assumption that certain regions of the face may be corrupted by noise, but not all regions simultaneously . Given a time window of length $T$, we extract pixel intensity signals from 5 regions of the face to create a sample $\mathbf{x}_1 \in \mathbb{R}^{T \times 5}$. The pulse signal is collected from finger PPG; this signal is originally $\mathbf{x}_0 \in \mathbb{R}^{T \times 1}$. During the loss computation, Pytorch "broadcasts" the ground-truth signal to match the length of the predicted signal (i.e. of dimension $\mathbb{R}^{T \times 5}$); for clarity, we mention that the signal is repeated 5 times for loss computation.

**Action:** None

**Reviewer Point P 2.6** — It would be good to include a more interpretable metric like ECE as well.

**Response**: While ECE is usually defined for classification tasks, we have added a variety of new metrics suitable for regression including the Continuous Ranked Probability Score, Sharpness, Check

Score/Pinball Loss, and Interval Score in addition to the Negative Log Likelihood in the main paper Table IV and Table V. Furthermore, we have plotted the calibration curves, which instead of summarizing a mean, shows the calibration error at each predicted proportion as well as the miscalibration error [20]. We achieve a miscalibration error of 0.06 on both the UBFC-rPPG and PURE datasets as shown in Figure 5.

**Action:** We have added new metrics related to regression in Table 5, as well as included calibration curves in Figures 5 and 9.

**Reviewer Point P 2.7** — Equation 13: rho is usually used for Spearman correlation. These don't really need defining either, they are v common metrics.

**Response**: We agree with the reviewer that $\rho$ is often used for Spearman. However, we followed the convention used previous iPPG papers such as [21] to avoid confusion and promote consistency. We have elected to keep the definition of these metrics in Section 5.2.

**Action:** No action taken.

**Reviewer Point P 2.8** — Define "Given the N signal estimates, we perform an uncertainty analysis of pulse signal recovery from a facial video." i.e. just Monte Carlo estimation?

**Response**: Correct. To be precise, we approximate the posterior distribution of the pulse signal $p(x_0|x_1)$ by generating N independent trajectories via the reverse-time SDE. This acts as a Monte Carlo estimator of the posterior, allowing us to compute empirical confidence intervals and calibration metrics. We agree that using the explicit term helps clarify the methodology. We have updated the text to reflect this.

**Action:** We have clarified this in the introduction

---

# Reviewer ZKFz

**Reviewer Point P 3.1** — *Existing theoretical analysis primarily serves to motivate and explain the proposed formulation, but does not address properties such as generalization, robustness, or stability. For example, how would the error from training compose with the error from the SDE solver? It would be nice to include those outcome-focused theoretical analyses.*

**Response**: We thank the reviewer for suggesting an outcome-focused theoretical analysis. We agree that addressing properties such as generalization, robustness, and stability is critical for clinical adoption. We have added a new section to **Appendix A.9** detailing these proofs. We summarize our theoretical results below:

1. **Error Composition (Approximation vs. Discretization):** To address the reviewer's question on how errors compose, we model the total sampling error as the sum of the *network approximation error* and the *solver discretization error*. As detailed in Equation 2, the error bound is given by:

$$\text{Total Error} \leq \mathbb{E}\left[\int_0^1 \|b_F(t, x_t) - b_\theta(t, x_t)\|dt\right] + \mathcal{O}(\Delta t) \tag{2}$$

The first term represents the training quality (how well our neural network $b_\theta$ approximates the true vector field $b_F$). The second term, $\mathcal{O}(\Delta t)$, represents the error introduced by the SDE solver steps. This confirms that the error is additive and controllable via training convergence (Figure 7) and step-size selection.

2. **Theoretical Stability (BIBO):** We have proven that our system is **Bounded-Input, Bounded-Output (BIBO) stable**. In **Theorem 1** (added to Appendix A.9), we show that if the learned drift coefficient $b_{F,\theta}$ is Lipschitz continuous with constant $L$ (a property generally satisfied by U-Net architectures with bounded activations), the reconstruction error between a clean measurement $\mathbf{x}_1$ and a perturbed measurement $\tilde{\mathbf{x}} = \mathbf{x}_1 + \delta$ is bounded by:

$$\|\hat{\mathbf{x}}_0 - \tilde{\mathbf{x}}_0\| \leq e^L \|\delta\| \tag{3}$$

This proves that bounded input noise $\|\delta\| < \infty$ results in bounded reconstruction error.

3. **Empirical Validation of Stability:** We validate this theoretical stability via a **Gauge R&R analysis** (Appendix A.9.6). We demonstrate that the variance due to the measurement system (repeatability) is negligible ($1.7\%$) compared to the signal variation. This empirically confirms the stability predicted by our theorem.

4. **Robustness and Generalization via RCL:** Finally, regarding generalization, the Residual Correlation Loss (RCL) acts as an inductive bias (or smoothness prior). By constraining the solution manifold to signals that are consistent across time shifts, we theoretically reduce the hypothesis space to physically plausible signals. This improves robustness against high-frequency outliers (e.g., sudden motion artifacts) that violate this correlation structure. We have also added training/validation curves in Figure 7 to show that our model learns the flow and score correctly.

**Action:** We have added significant changes to Appendix A.9. We have added a proof on BIBO stability to Appendix A.9, added the error composition analysis (Eq. 37) to the Appendix A.9, and include a discussion of Robustness and Generalization in Appendix A.9. We have also included the Gauge R&R empirical stability analysis in Appendix A.10.6.

**Reviewer Point P 3.2** — *Test-time SDE-based sampling has the advantage to access solutions to all time points. Is it iterative w.r.t. , i.e. you can't get later if you don't get earlier during the simulation / the solver you're using. If so, the sampling seems to take some time and it would be interesting to see the inference time comparing different methods to see if it's time-efficient as well (No need to beat them, comparable is sufficient).*

**Response**: Thanks for your comment. Yes, solving an SDE is expensive as it is sequential and iterative, require multiple evaluations of our flow and score networks. We can not obtain the solution at a later time without first evaluating the solution at earlier times; if inference speed is the dominating constraint when chosing a posterior sampling method, then RIS-iPPG may not be the best choice.

We further explore this by measuring the inference times for a single forward pass of different models in Table 10, Appendix A.5. We notice that RIS-iPPG has the highest inference time; however, RIS-iPPG has many benefits, as enumerated in Appendix A.4. We believe inference speed is important; future work should explore accelerated sampling techniques, an active research area for the diffusion community.

We add the Table X and the following text to the manuscript Section A.5:

Table X: Comparing inference speed of different methods. All were tested on a single NVIDIA A5000 GPU

| Method | Inference Time (ms) |
|---|---|
| Conditional VAE | 845.06 |
| Bayesian Neural Networks | 473.91 |
| RIS-iPPG | 850.97 |

We measure the runtime performance at inference for three sampling methods in Table X. At inference, BNNs are best in terms of speed; cVAEs and RIS-iPPG are about the same. However, the models in RIS-iPPG need to be evaluated for every timstep, and must do so sequentially; depending on the granularity with which the SDE steps are discretized, the cost of RIS-iPPG can be much higher. Note that we do not claim RIS-iPPG to be superior in terms of runtime; however, fast inference is important for real-world adoption and future work should aim to improve inference speed.

**Action:** We have added the above table and text to Appendix Section A.5.

**Reviewer Point P 3.3** — *Typo: "OUr" in table 2 caption.*

**Response**: Thanks for pointing this out. We have fixed it.

**Action:** We have fixed this.

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

Figure 10: Inference via solving an SDE. We plot the signal measurements as well as the SDE solution at $\Delta t = 0.1$ time steps in the range $t \in [0, 1]$.

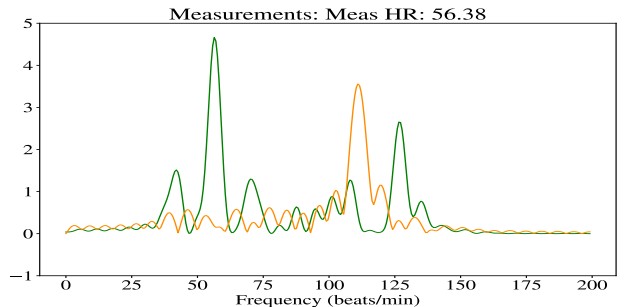

(a) The original signal measurements (green) and ground-truth (orange).

(b) $t = 0.1$

(c) $t = 0.2$

(d) $t = 0.3$

(e) $t = 0.4$

(f) $t = 0.5$

(g) $t = 0.6$

(h) $t = 0.7$

(i) $t = 0.8$

(j) $t = 0.9$

(k) $t = 1.0$

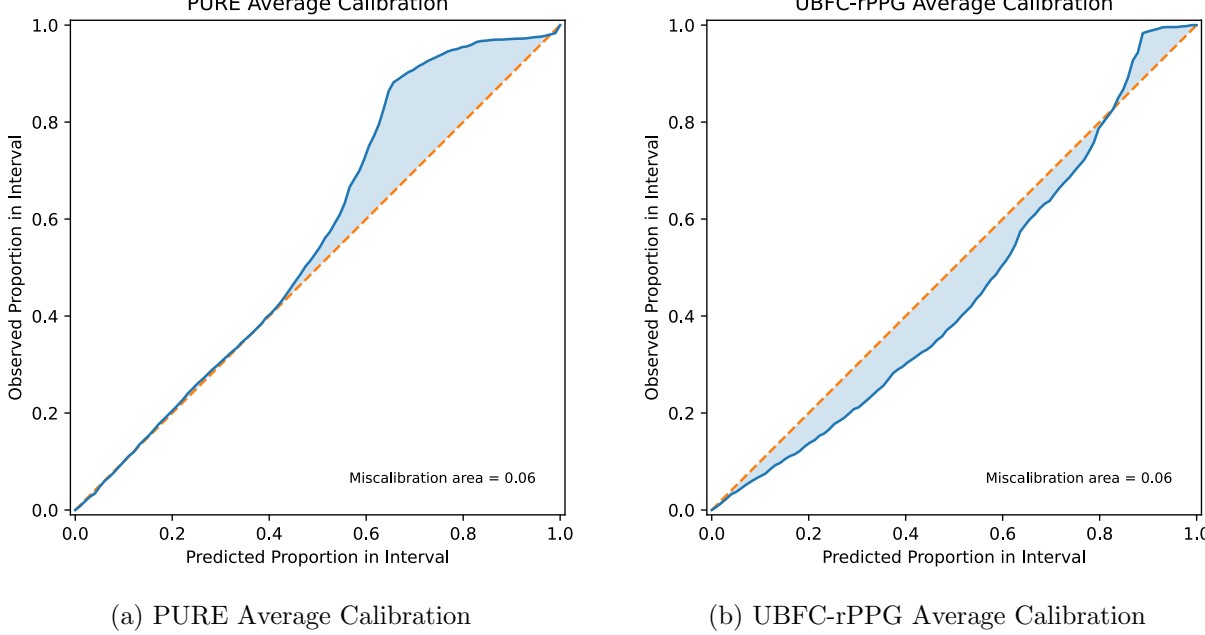

(a) PURE Average Calibration      (b) UBFC-rPPG Average Calibration

Figure 5: Calibration curves for PURE and UBFC-rPPG datasets. For PURE, the model underpredicts the observed proportion. For UBFC-rPPG, the model is slightly overconfident at lower observed proportions and slightly underconfident at higher ones.

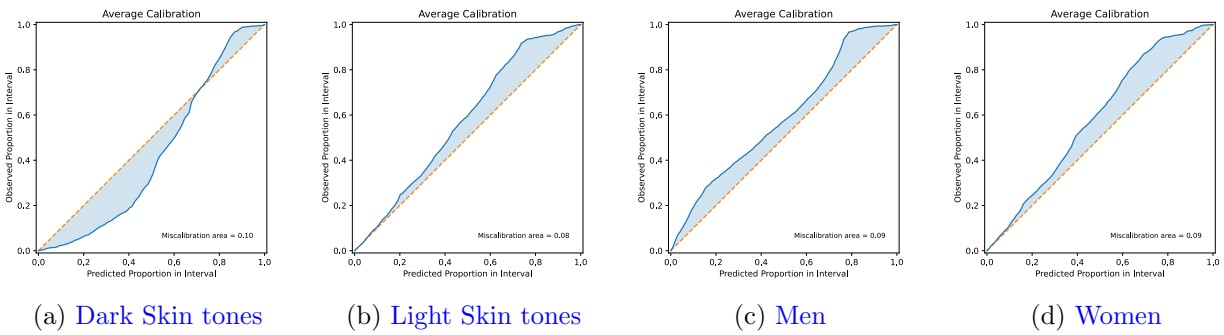

(a) Dark Skin tones    (b) Light Skin tones    (c) Men    (d) Women

Figure 9: Average calibration curves for the four protected attributes on the MMSE-HR dataset. In all four scenarios our method performs well; however, our method is relatively worse on dark skin tones.

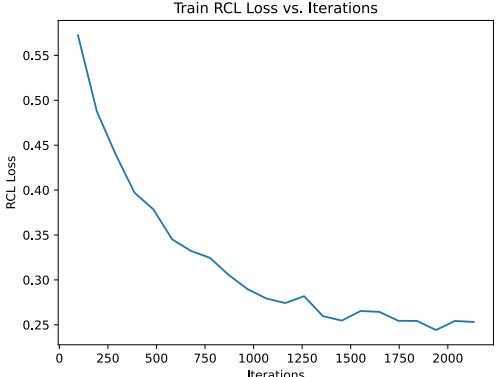

(a) RCL training loss vs training iterations

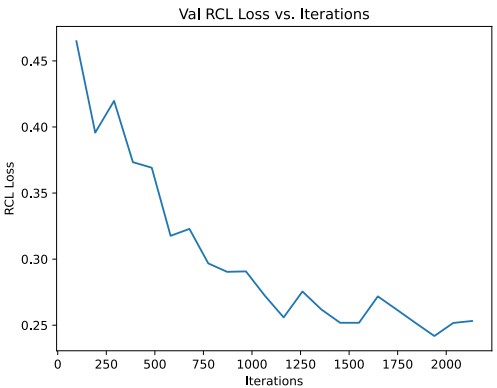

(b) RCL validation loss vs training iterations

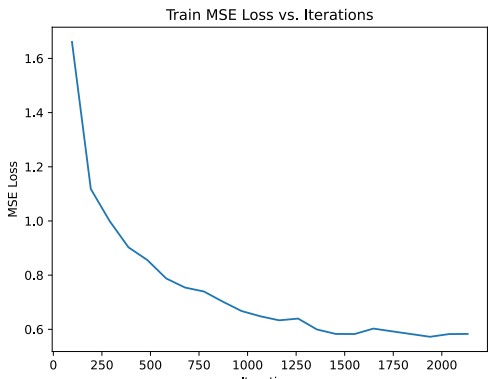

(c) Flow+Score+RCL training loss vs training iterations

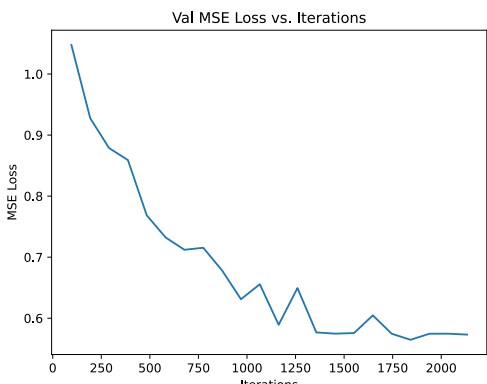

(d) Flow+Score+RCL validation loss vs training iterations

Figure 7: The training and validation loss curves for the RCL loss and the entire loss (flow loss + score loss + RCL loss of equation 12)

