# OpenReview forum: "Uncertainty-quantified Pulse Signal Recovery from Facial Video using Regularized Stochastic Interpolants"
_TMLR — Rejected by TMLR_

### Review · Reviewer_uiSP · 2025-08-07

**Summary Of Contributions:**

This paper introduces RIS-iPPG, a probabilistic method for recovering blood volume pulse (BVP) signals from facial video. Instead of producing a single estimate, it samples multiple solutions at test time to quantify uncertainty. The approach treats signal recovery as an inverse problem and uses a stochastic differential equation to model signal evolution. A regularization term promotes temporal consistency based on the smooth nature of physiological signals. The approach is tested on three datasets and shows that it can produce accurate signal estimates while also offering insights into how confident the model is in its predictions.

**Additional Comments:**

- The paper does not establish that current lack of test-time sampling is the main bottleneck to clinical adoption. Real-world clinical challenges (e.g. robustness to skin tone variation, ethical concerns, lack of regulation) are arguably more pressing than uncertainty estimates. I would like the authors to comment on this.

- The authors emphasize that “no previous algorithm samples the solution space… at test time”, but this is misleading. Several Bayesian deep learning and ensemble-based approaches do provide uncertainty estimates in medical AI contexts. Moreover, just sampling from a posterior (via diffusion or otherwise) is not inherently more clinically trustworthy unless uncertainty calibration and interpretability are clearly demonstrated. The authors equate sampling with trustworthiness, which oversimplifies the issue.

- The choice to model pulse recovery as an inverse problem solved via SDEs and stochastic interpolation is intriguing, but also possibly excessive. The fundamental signal recovery task is low-dimensional and highly structured, often one-dimensional (heart rate signal) with relatively predictable dynamics. Why is a high-capacity generative model like a diffusion process necessary? The paper lacks a justification of why this level of complexity is needed over more straightforward probabilistic models (e.g. variational methods, Gaussian processes).

- Section 4.4: How are overlapping windows selected? How much overlap is used? Is the method robust to window size and stride?

 Minor:

- Several citations are awkwardly placed. For example: “Advances in self-supervised [references] algorithms learn using facial video data only…” should be “Advances in self-supervised algorithms [references] learn using facial video data only…”. Or “…led to clinical validation studies of iPPG algorithms [reference] in neonatal care units” should be “…led to clinical validation studies of iPPG algorithms in neonatal care units [reference].”

- Typo on p.5: “induced by pump a called the heart”  should be reworded as: “induced by a pump called the heart.”

**Audience:**

Yes

**Audience Explanation:**

This paper touches on multiple timely research threads: contactless physiological monitoring, uncertainty-aware inference, and diffusion-inspired generative models. These are all active areas of interest for the TMLR audience. The integration of flow-based models and stochastic differential equations for time-series signal estimation is novel within the iPPG domain and could stimulate new directions in medical machine learning. However, the technical execution and conceptual framing need significant refinement to make the paper impactful and credible.

**Broader Impact Concerns:**

None.

**Claims And Evidence:**

No

**Claims Explanation:**

While the paper presents a technically interesting framework, framing iPPG recovery as an inverse problem and using stochastic interpolants for uncertainty-aware signal recovery, many of its core claims are either insufficiently supported or conceptually overstated.

- The physiological and imaging processes are collapsed into a black-box forward model A(x_0), without modeling or estimating its structure. This oversimplification makes the inverse recovery problem severely underconstrained and undermines the argument that the problem formulation closely reflects the physical generation process.

- The comparison with Plug-and-Play Monte Carlo (PMC) methods is unconvincing. The forward model is approximated as A = F^(-1), based on tenuous links to prior work. The poor performance of PMC-iPPG could stem from this unrealistic assumption rather than limitations of the sampling approach itself. Negative results are presented without deeper analysis of what failed (e.g., model mismatch, training instability).

- The paper introduces a complex stochastic interpolant-based model to map between distributions of camera and BVP signals, but it doesn't justify why this approach is necessary or preferable over simpler conditional generative models (e.g., conditional VAEs, Bayesian neural nets). The task is relatively low-dimensional and structured, raising questions about whether such a high-capacity method is warranted.

- The claim that test-time sampling solves the trust problem in clinical adoption is oversimplified. Sampling from a posterior doesn't inherently provide trustworthy or calibrated uncertainty. The authors do not evaluate uncertainty calibration, nor do they explain how clinicians would interpret or use the uncertainty estimates in practice.

- Several key equations, including Eq. 1, 7, and 12, are introduced abruptly or without sufficient motivation. The decomposition of the drift term and use of Tweedie's formula are technical and important, but underexplained.

- Learning accurate scores and flow fields is central to the method but is only briefly described. Issues such as stability, data requirements, or approximation quality are not discussed.

**Requested Changes:**

- Section3 - “Our goal is to link two arbitrary distributions…”: While technically this is true in theory, in practice the method assumes paired samples (e.g., (x_0,x_1)) and the ability to model conditional relationships. The term "arbitrary" might mislead readers into thinking the method can connect any two unstructured, unpaired distributions, which is not practical. Rephrase to something like: “Our goal is to link two related data distributions, such as camera pixel signals and physiological signals, by learning a continuous stochastic interpolation between them.

- Section 3 - “...interpolates from one distribution’s data point to the another distribution’s data point.” This phrase is awkward and technically imprecise. It sounds like you’re connecting two specific data points, not distributions. Clarify with: “...interpolates from a sample from one distribution to a corresponding sample from another."

- Clarify what x_0  actually represents: Is it a temporal waveform? A spatial BVP map? A latent signal? Be precise about its dimensionality and nature. Rather than introducing A(.) as a black box, provide some physical intuition or approximation, e.g., color channel mixing, temporal filtering, based on known properties of the iPPG pipeline.

- The assumption A=F^(-1) is weakly motivated and likely inaccurate. Explain where this approximation comes from and what its limitations are.

- Justify the use of stochastic interpolants over simpler alternatives: The task is low-dimensional and could potentially be solved with VAEs, Gaussian Processes, or Bayesian regression. Explain why this method is uniquely suited.

- Break down Equation 12 step-by-step: It’s a key part of the method, but is difficult to follow without prior exposure. Help the reader understand how the drift term is derived and what each component does.

- The idea of correlation of residuals is intuitive, but the connection between residual alignment and better physiological signal recovery should be backed by either theoretical insight or empirical correlation (e.g., does lower RCL correspond to better waveform accuracy?).

---

> ### Author Response · Authors · 2025-12-16
> **Reviewer Point P.1.1**
>
> * _The physiological and imaging processes are collapsed into a black-box forward model $A(x_0)$, without modeling or estimating its structure. This oversimplification makes the inverse recovery problem severely underconstrained and undermines the argument that the problem formulation closely reflects the physical generation process._
> * _The comparison with Plug-and-Play Monte Carlo (PMC) methods is unconvincing. The forward model is approximated as $A = F^{−1}$, based on tenuous links to prior work. The poor performance of PMC-iPPG could stem from this unrealistic assumption rather than limitations of the sampling approach itself. Negative results are presented without deeper analysis of what failed (e.g., model mismatch, training instability)._
> * _The assumption $A = F^{−1}$ is weakly motivated and likely inaccurate. Explain where this approximation comes from and what its limitations are._
> * _Rather than introducing $A(\cdot)$ as a black box, provide some physical intuition or approximation, e.g., color channel mixing, temporal filtering, based on known properties of the iPPG pipeline._
>
>
> We thank the reviewer for their insightful critique. Our goal when proposing PMC-iPPG was to explicitly show the drawbacks of using a simple forward model such as $A = F^{-1}$ as proposed in [1][2]. The work of [2] makes the strong assumption that the camera
> pixel signal is simply the pulse signal with additive gaussian noise, which makes $A=F^{-1}$ suitable; they achieve good results because of careful signal extraction/noise removal during the time-series extraction phase, not because of the signal recovery algorithm. The work of [1] also uses $A = F^{-1}$ ; however, by “unrolling” proximal gradient descent with deep denoising operators, they effectively correct an inexact forward operator. Our experiments using PMC-iPPG, which only learns the signal prior, shows that $A = F^{-1}$ as described in [1][2] is suboptimal .
>
> Our next step was to address this deficiency by learning the forward model[3][4]. However, we found this problem to be ill-posed for a variety of reasons as described in Appendix A.3.1 and Figure 6.
>
> * **Stochastiscity**: The mapping between the blood volume pulse signal and the camera pixel signal is stochastic and time-dependent due to unconstrained motion and its associated changes in specular and diffuse reflections. This is reflected in Figure 6, which shows the significant changes in the signal over four second time intervals.
>
> * **One-to-Many Mapping**: The pixel signal from different regions of the face correspond to one blood volume pulse signal, which is captured at the finger. A deterministic function can not map a single ground-truth pulse to different facial regions. An example of this is shown in Figure 6 when comparing the pixel signals at the left and right cheek.
>
> All these lead us to the decision to use stochastic interpolants, which _implicitly_ learns the optimal transport between the distribution of camera pixel signals and BVP signals. Modeling the signal recovery process as such allows for greater flexibility, especially when accounting for the physics of the problem as well as long-tailed events such as sharp motions. By matching distributions, RIS-iPPG captures non-linear interactions that define the forward processing while also allowing for uncertainty quantification, one of our primary goals.
>
> **Action**: We have re-written Section 4.2 of the main paper, added Appendix A.3.1, and included Figure 6 to explain our rationale.
>
> [1] Vineet R Shenoy, Tim K Marks, Hassan Mansour, and Suhas Lohit. Unrolled ippg: Video heart rate estimation via unrolling proximal gradient descent. In 2023 IEEE International Conference on Image Processing (ICIP), pp. 2715–2719. IEEE, 2023.
>
> [2] Ewa M Nowara, Tim K Marks, Hassan Mansour, and Ashok Veeraraghavan. Near-infrared imaging photoplethysmography during driving. IEEE Transactions on Intelligent Transportation Systems, 23(4):3589–3600, 2020.
>
> [3] Simon Arridge, Andreas Hauptmann, and Yury Korolev. Inverse problems with learned forward operators.arXiv preprint arXiv:2311.12528, 2023
>
> [4] Sebastian Lunz, Andreas Hauptmann, Tanja Tarvainen, Carola-Bibiane Schönlieb, and Simon Robert Arridge. On learned operator correction in inverse problems. SIAM journal on imaging sciences, 14:92127, 2020. URL https://api.semanticscholar.org/CorpusID:224899525.

---

> ### Author Response · Authors · 2025-12-16
> **Reviewer Point P.1.2**
>
> * _The authors emphasize that “no previous algorithm samples the solution
> space. . . at test time”, but this is misleading. Several Bayesian deep learning and ensemble-based approaches do provide uncertainty estimates in medical AI contexts._
>
>
>
> Thanks for pointing this out. We clarify that we are the first _iPPG method_ to perform posterior sampling for pulse signal recovery. While there are many other types of posterior sampling methods that can be used, _no previous literature has actually done so for iPPG_, and it is important that researchers address this task [1].
>
> We acknowledge that Bayesian methods are well-established in the broader Medical AI literature, especially for diagnostic classification. However, distinct from these tasks, RIS-iPPG samples high-dimensional signal trajectories (waveforms) rather than single scalar values. We tried using Bayesian Neural Networks and Conditional VAEs for pulse rate estimation as shown in Table 8 but achieved poor result. A full discussion of these methods is reserved for Reviewer Point 1.3.
>
> **Action**: We have updated the manuscript to reflect the distinctions for iPPG, added Appendix A.4 and Table 8, and clarified that our method samples waveform.
>
> [1] S. Tonekaboni, S. Joshi, M. D. McCradden, and A. Goldenberg, “What clinicians want: contextualizing explainable machine learning for clinical end use,” in Machine learning for healthcare conference. PMLR, 2019, pp. 359–380.

---

> ### Author Response · Authors · 2025-12-16
> **Reviewer Point P.1.3: Part 1**
>
> * _The paper introduces a complex stochastic interpolant-based model to map between distributions of camera and BVP signals, but it doesn’t justify why this approach is necessary or preferable over simpler conditional generative models (e.g., conditional VAEs, Bayesian neural nets). The task is relatively low-dimensional and structured, raising questions about whether such a high-capacity method is warranted._
>
> * _Justify the use of stochastic interpolants over simpler alternatives: The task is low-dimensional and could potentially be solved with VAEs, Gaussian Processes, or Bayesian regression. Explain why this method is uniquely suited._
>
> * _The choice to model pulse recovery as an inverse problem solved via SDEs and stochastic
> interpolation is intriguing, but also possibly excessive. The fundamental signal recovery task is low-dimensional and highly structured, often one-dimensional (heart rate signal) with relatively predictable dynamics. Why is a high-capacity generative model like a diffusion process necessary? The paper lacks a justification of why this level of complexity is needed over more straightforward probabilistic models (e.g. variational methods, Gaussian processes)._
>
>
> Thanks for this valuable critique. We address each point below.
>
>
>  **Complexity of the Task**: While our signals are one-dimensional, the **inverse mapping** between the camera pixel signal and pulse signal is highly complex. In Figure 6, we show how the camera pixel signal changes during sharp changes in motion every four seconds, as well as the resulting signals from two different regions of the face. These signals contain high-dimensional, non-Gaussian noise sources such as motion, specular reflections, quantization noise, etc. in the same frequency bands as the pulse. Simple models like Gaussian Processes (GP) or BNNs often assume Gaussian noise or unimodal posteriors, an assumption violated by our data distributions. RIS-iPPG can more effectively model this data.
>
> **Empirical Analysis**: We conduct experiments to measure the performance of Conditional VAEs (cVAE) and Bayesian Neural Networks (BNN) against flow-based models. We evaluate these methods for pulse rate estimation performance in Table 8. For Conditional VAEs we adapt the work [1] and use the camera pixel signal as our conditioning element. For the BNN, we follow the guide of [2] and set the prior as recommended in the paper. Note that in Table 8, we see significant increases in pulse rate estimation error using BNNs and cVAE compared to RIS-iPPG.
>
>  **Performance of cVAE**: We notice that pulse rate estimation performance for VAEs is significantly worse than that of the flow-based models. VAEs are known to produce blurry samples due to the ELBO objective; for a task like ours in which we are capturing waveform morphology, VAEs will smoothe features such as the systolic peak resulting in inaccurate pulse rate estimation. Additionally, previous work has noted that VAEs tend to ignore latent variables if the decoder is overly powerful in sequence modeling tasks [3][4][5]. It is also well-known that VAEs suffer from posterior collapse [6][7]. Finally, VAEs often show poorer quality of samples as well [8].
>
> **Performance of BNNs**: Bayesian neural networks also have their own challenges. One of the biggest challenges is carefully defining the prior distribution over the weights of the BNN[2]. Previous work has shown that the performance of BNNs are greatly affected by the choice of prior [9], and often the choice of distribution over the weights is non-intuitive for the selected task. It is unclear how the choice of prior over the weights translate to a prior over signal morphology. Furthermore, inference methods in BNNs often have drawbacks: MCMC methods are exact but do not scale well to large models while variational methods are inexact and only focus on a single mode of the posterior[2]. RIS-iPPG, without explicity setting a prior, effectively samples waveforms while capturing multiple modes of the posterior.
>
> **Performance of GPs**: These models suffer from computational complexity concerns, scaling cubicly with the size of the dataset ]10]. Additionally, the choice of kernel in GPs often affects the generalization ability of the model [11] [12]; as in BNNs, prior knowledge about the kernel with reference to the task is critical for successful application of GPs. Often, these kernels fail to capture the complex and non-stationary nature of motion artifacts.
>
> #### Table 8: Comparing Conditional VAEs and RIS-iPPG on the MMSE-HR dataset
> | Method | MAE (bpm) $\downarrow$ | RMSE (bpm) $\downarrow$ |
> | :--- | :--- | :--- |
> | Conditional VAES | 6.17 | 9.96 |
> | Bayesian Neural Networks | 3.12 | 5.63 |
> | RIS-iPPG | 1.97 | 3.73 |

---

> ### Author Response · Authors · 2025-12-16
> **Reviewer Point P.1.3: Part 2**
>
> **Benefit of RIS-iPPG**: Flow-based diffusion models solve many of the problems of VAEs, BNNs, and GPs. We can model complex data distributions, capture distinct waveform morphologies, model highly non-Gaussian and non-stationary noise, and capture multimodal posteriors. Given the recent research showing state-of-the-art performance and widespread adoption of flow-based diffusion models, it is critical that we address research in iPPG with respect to such models.
>
> **Action**: We have added this analysis to Section A.4, and included Table 8 in Appendix A.4.
>
> References:
> [1] K. Sohn, H. Lee, and X. Yan, “Learning structured output representation using deep conditional generative models,” Advances in neural information processing systems, vol. 28, 2015.
>
> [2] L. V. Jospin, H. Laga, F. Boussaid, W. Buntine, and M. Bennamoun, “Hands-on bayesian neural networks—a tutorial for deep learning users,” IEEE Computational Intelligence Magazine, vol. 17, no. 2, pp. 29–48, 2022.
>
> [3] X. Chen, D. P. Kingma, T. Salimans, Y. Duan, P. Dhariwal, J. Schulman, I. Sutskever, and P. Abbeel, “Variational lossy autoencoder,” arXiv preprint arXiv:1611.02731, 2016.
>
> [4] I. Serban, A. Sordoni, R. Lowe, L. Charlin, J. Pineau, A. Courville, and Y. Bengio, “A
> hierarchical latent variable encoder-decoder model for generating dialogues,” in Proceedings of the AAAI conference on artificial intelligence, vol. 31, no. 1, 2017.
>
> [5] S. Bowman, L. Vilnis, O. Vinyals, A. Dai, R. Jozefowicz, and S. Bengio, “Generating sentences from a continuous space,” in Proceedings of the 20th SIGNLL conference on computational natural language learning, 2016, pp. 10–21.
>
> [6] B. Dai, Z. Wang, and D. Wipf, “The usual suspects? reassessing blame for vae posterior
> collapse,” in International conference on machine learning. PMLR, 2020, pp. 2313–2322.
>
> [7] J. Lucas, G. Tucker, R. B. Grosse, and M. Norouzi, “Don’t blame the elbo! a linear vae
> perspective on posterior collapse,” Advances in Neural Information Processing Systems, vol. 32,
> 2019.
>
> [8] J. Xu, H. Le, and D. Samaras, “Assessing sample quality via the latent space of generative models,” in European Conference on Computer Vision. Springer, 2024, pp. 449–464.
>
> [9] D. Silvestro and T. Andermann, “Prior choice affects ability of bayesian neural networks to
> identify unknowns,” arXiv preprint arXiv:2005.04987, 2020.
>
> [10] D. Eriksson, K. Dong, E. Lee, D. Bindel, and A. G. Wilson, “Scaling gaussian process regression with derivatives,” Advances in neural information processing systems, vol. 31, 2018.
>
> [11] C. E. Rasmussen, “Evaluation of gaussian processes and other methods for non-linear regres-
> sion,” Ph.D. dissertation, University of Toronto Toronto, Canada, 1997.
>
> [12] S. Sun, G. Zhang, C. Wang, W. Zeng, J. Li, and R. Grosse, “Differentiable compositional kernel
> learning for gaussian processes,” in International Conference on Machine Learning. PMLR,
> 2018, pp. 4828–4837.

---

> ### Author Response · Authors · 2025-12-16
> **Reviewer Point P.1.4**
>
> * _The claim that test-time sampling solves the trust problem in clinical adoption is oversimplified. Sampling from a posterior doesn't inherently provide trustworthy or calibrated uncertainty. The authors do not evaluate uncertainty calibration, nor do they explain how clinicians would interpret or use the uncertainty estimates in practice._
>
> * _Moreover, just sampling from a posterior (via diffusion or otherwise) is not inherently more clinically trustworthy unless uncertainty calibration and interpretability are clearly demonstrated. The authors equate sampling with trustworthiness, which oversimplifies the issue._
>
>
> Thank you for this feedback; we have significantly improved the paper following this comment. We have extensively re-written the analysis of RIS-iPPG, and have included five UQ metrics for predictive uncertainty quantification on which evaluate our algorithm. We have also included calibration curves, both on the original test sets of PURE and UBFC-rPPG, but also on protected attributes (gender, skin color) on the MMSE-HR dataset. We have established the first baselines for uncertainty quantification, upon which we hope others build.
>
> **How do this address trust?**: We argue that the above additions help improve trustworthiness in the following ways:
>
>  * **Calibration as a Proxy for Trustworthiness**: A clinician who uses the algorithm expects that a model's confidence would match its real-world accuracy. Calibration curves in Figures 5 and Figure 9, in which the our model predictions are close to the diagonal, show that confidence is well correlated with model accuracy (and not confidently wrong, for example).
> * **Capturing Multimodal posteriors** We see evidence that our model predicts multiple posteriors instead of averaging the spectrum (i.e. Appendix A.9.5, Figure 13), capturing not only the true peak, but also other peaks and the harmonic
> * **Quantifying Bias for Skin Tone Equity**: We have added Table 5, which shows the quantified uncertainty for protected attributes. This is critical for clinicians as they determine for whom the model is most reliable.
>
>
> We have added the following to the manuscript.
>
> * Added Tables 4 and 5 enumerating UQ metrics over the test sets of UBFC-rPPG and PURE, and the protected attributes of MMSE-HR
>
> * Added Figures 5 and 9, which are 6 calibration curves evaluated on different subsets of the data
>
> * Re-wrote much of Section 5 to describe the new UQ metrics and analysis of uncertainty quantification
>
>
> **Action**: We have added Table 4, Table  5, Figure 5, Figure 9, and re-wrote much of Section 5 to address this comment.
>
> #### Table 4: Comparing the spectrum estimation performance and uncertainty quantification metrics with and without the RCL loss
>
> | Method | MAE| RMSE | PCC | NLL | Sharpness | CRPS | Check Score | Interval Score |
> | :--- | :--- | :--- | :--- | :--- | :--- | :--- | :--- | :--- |
> | PURE w/out RCL | 0.057 | 0.681 | 0.224 | -3.825 | 0.201 | 0.041 | 0.021 | 0.217 |
> | PURE w/RCL | 0.700 | 0.056 | 0.218 | -3.820 | 0.202 | 0.040 | 0.020 | 0.208 |
> | UBFC-rPPG w/out RCL | 0.07 | 0.651 | 0.237 | -3.475 | 0.177 | 0.042 | 0.0121 | 0.162 |
> | UBFC-rPPG w/RCL | 0.049 | 0.795 | 0.185 | -3.516 | 0.168 | 0.034 | 0.017 | 0.162 |
>
>
>
> #### Table 5: Proper Scoring Rule metrics on Protected Attributes of MMSE-HR dataset
>
> | Protected Attribute | HR MAE | HR RMSE | NLL | Sharpness | CRPS | Check Score | Interval Score |
> | :--- | :--- | :--- | :--- | :--- | :--- | :--- | :--- |
> | Light Skin Tone | 2.17 | 3.74 | -3.430 | 0.203 | 0.042 | 0.021 | 0.209 |
> | Dark Skin Tone | 4.79 | 6.84 | -2.828 | 0.191 | 0.035 | 0.018 | 0.188 |
> | Men | 2.07 | 3.83 | -3.435 | 0.196 | 0.032 | 0.016 | 0.167 |
> | Women | 2.23 | 3.69 | -3.415 | 0.203 | 0.039 | 0.020 | 0.195 |

---

> ### Author Response · Authors · 2025-12-16
> **Reviewer Point P.1.5**
>
> * _Learning accurate scores and flow fields is central to the method but is only briefly described. Issues such as stability, data requirements, or approximation quality are not discussed._
>
>
> We agree that these are important; the work of [1] discusses this in detail in Sections 2.4 and 2.5, with proofs of likelihood control, density estimation, and cross-entropy calculations in the appendix. We agree that demonstrating these properties in practice is important. We do so as follows:
>
> * **Stability and Convergence:** To show our stability and convergece, in Appendix A.8 we plot the training and validation loss curves for the flow, score, and RCL components in Figure 7. We see that both training and validation curves decrease monotonically, confirming that we are successfully approximating our intended drift coefficient.
> * **Data Requirements:** The datasets we used recorded at least one minute of video per subject at at least 25 FPS, with many subjects recording multiple videos under stimuli. This provides adequate data to sample the manifold of physiological signals under varying levels of noise.
> * **Approximation Quality:** Our Gauge R\&R analysis, shown in Appendix A.9.6, quantifies the repeatability and precision of our system. The variance due to repeatability is negligible (1.7\%) compared to the variance between different signal frequencies ("Part"). This suggests correct model formulation and learning.
>
> **Action**: We have added Figure 7 to Appendix A.8, and included a Gauge R\&R analysis to Appendix A.9.6 to address these comments.
>
> [1] M. S. Albergo, N. M. Boffi, and E. Vanden-Eijnden, “Stochastic interpolants:
> A unifying framework for flows and diffusions,” 2023. [Online]. Available: https:
> //arxiv.org/abs/2303.08797

---

> ### Author Response · Authors · 2025-12-16
> **Reviewer Point P.1.6**
>
> * _Our goal is to link two arbitrary distributions. . . ”: While technically this is true
> in theory, in practice the method assumes paired samples (e.g., $(x_0, x_1)$) and the ability to
> model conditional relationships. The term ”arbitrary” might mislead readers into thinking
> the method can connect any two unstructured, unpaired distributions, which is not practical.
> Rephrase to something like: “Our goal is to link two related data distributions, such as cam-
> era pixel signals and physiological signals, by learning a continuous stochastic interpolation
> between them_
>
> * _Section 3 - “...interpolates from one distribution’s data point to the another distribution’s data
> point.” This phrase is awkward and technically imprecise. It sounds like you’re connecting
> two specific data points, not distributions. Clarify with: “...interpolates from a sample from
> one distribution to a corresponding sample from another.”_
>
> * Several key equations, including Eq. 1, 7, and 12, are introduced abruptly or without sufficient
> motivation.
>
> Thank you for the suggestions. We have directly made the changes as requested in the first two bullet points. We have also re-written the text describing Equations 1, 7, and 12 to be more clear.
>
>
> * **Equation 1:** ``The function $I(t, \mathbf{x}_0, \mathbf{x}_1)$ smoothly interpolates between the points $\mathbf{x}_0$ and $\mathbf{x}_1$ as a function of $t$ such that $\mathbf{x}_{t=0} = \mathbf{x}_0$ and $\mathbf{x}_{t=1} = \mathbf{x}_1$, while $\gamma(t)\mathbf{z}$ is an added noise term. To bridge between these points, we would like to know the instantaneous change of $\mathbf{x}_t$, which if learned for all $t$, would enable us to transport $\mathbf{x}_1$ to $\mathbf{x}_0$ or vice versa via an SDE."
> * **Equation 7:** "We seek to define a a stochastic process that smoothly interpolates between samples from two distributions, namely the camera intensity signals $\mathbf{x}_1\sim p_1$ and its ground-truth pulse signal $\mathbf{x}_0 \sim p_0$.
>
>
>
>     $$\mathbf{x}_t = (1-t)\mathbf{x}_0 + t\mathbf{x}_1 + \sqrt{2t(1-t)} \mathbf{z}$$
>
> This choice of $I(t, \mathbf{x}_0, \mathbf{x}_1) = (1-t)\mathbf{x}_0 + t\mathbf{x}_1$ and $\gamma(t) = \sqrt{2t(1-t)}$ as in Equation 7 ensures that at $\mathbf{x}_{t=0} = \mathbf{x}_0$ and $\mathbf{x}_{t=1} = \mathbf{x}_1$. Using this interpolant, our goal is to build a drift coefficient that satisfies Theorem 1. We are looking for the instantaneous change of Equation~\ref{eq:interpolant}: therefore, we are seeking
> $\mathbf{b}(t,\mathbf{x}) = \mathbb{E}[\frac{\partial}{\partial t }\big((1-t)\mathbf{x}_0 + t\mathbf{x}_1 + \sqrt{2t(1-t)} \mathbf{z}\big)]$"
>
> * **Equation 12:** "Our goal is to learn the flow, the score, and the correlation; the MSE losses for the score and flow [1] are added to our proposed correlation loss. The flow and score can be learned via simple $L_2$ minimization. The final loss is given by ..."
>
>
> **Action**: We have made the requested change.
>
> References:
> [1] M. S. Albergo, N. M. Boffi, and E. Vanden-Eijnden, “Stochastic interpolants:
> A unifying framework for flows and diffusions,” 2023. [Online]. Available: https:
> //arxiv.org/abs/2303.08797

---

> ### Author Response · Authors · 2025-12-16
> **Reviewer Point P.1.8**
>
> * _Break down Equation 12 step-by-step: It’s a key part of the method, but is difficult to follow without prior exposure. Help the reader understand how the drift term is derived and what each component does._
>
> * _The decomposition of the drift term and use of Tweedie's formula are technical and important, but underexplained._
>
> We have added a derivation of the drift coefficient as well as the score to the Appendix. In Appendix A.5 we show how to derive the denoiser, which is more stable than to learn than the score. In Appendix A.7, we showed that the drift coefficient is composed of a flow and a score, which are learned through two separate loss terms.
>
> **Action**: We have added a derivation of the drift coefficient as well as the score to the Appendix A.5 and A.7.

---

> ### Author Response · Authors · 2025-12-16
> **Reviewer Point P.1.9**
>
> * _The idea of correlation of residuals is intuitive, but the connection between residual alignment and better physiological signal recovery should be backed by either theoretical insight or empirical correlation (e.g., does lower RCL correspond to better waveform accuracy?)._
>
>
> Thank you for your comment. Our driving principle behind the RCL loss is that physiological signals such as pulse, under normal conditions, are slowly varying. The evidence for this claim is based on clinical experimentation, which is detailed in[1]. To constrain our network in such a way, we must ensure that residuals point in the same direction. We have included a discussion of this in Appendix A.6.
>
> To understand how our model benefits from the RCL loss, we plot the results of our algorithm with and without the RCL loss in Figure 3, as well as the quantitative metrics in Tables 3 and 4. Table 3 shows an HR estimation performance improvement when including the RCL loss. Waveform RMSE decreased from 0.224 to 0.218 and from 0.237 to 0.185 on the PURE and UBFC-rPPG datasets respectively when including the RCL loss, as shown in Table 4. We include these new results to strengthen our claims.  Also note that in the submission manuscript, we had included Figures 3, 12, and 13 qualitatively showing how signal estimation improves with the RCL loss; we keep these figures.
>
> #### Table 4: Comparing the spectrum estimation performance and uncertainty quantification metrics with and without the RCL loss
>
> | Method | MAE| RMSE | PCC | NLL | Sharpness | CRPS | Check Score | Interval Score |
> | :--- | :--- | :--- | :--- | :--- | :--- | :--- | :--- | :--- |
> | PURE w/out RCL | 0.057 | 0.681 | 0.224 | -3.825 | 0.201 | 0.041 | 0.021 | 0.217 |
> | PURE w/RCL | 0.700 | 0.056 | 0.218 | -3.820 | 0.202 | 0.040 | 0.020 | 0.208 |
> | UBFC-rPPG w/out RCL | 0.07 | 0.651 | 0.237 | -3.475 | 0.177 | 0.042 | 0.0121 | 0.162 |
> | UBFC-rPPG w/RCL | 0.049 | 0.795 | 0.185 | -3.516 | 0.168 | 0.034 | 0.017 | 0.162 |
>
> **Action**: We have added text to Section 4.4 providing the intuition as to why signals overlapping time window should be similar. Section 5.4 quantitatively shows how the RCL improves performance. We included a further discussion in Appendix A.6. In submission manuscript, we had included Figures 3, 12, and 13 qualitatively showing how signal estimation is improves with the RCL loss; we keep these figures.
>
>
> [1] W. W. Nichols, M. O’Rourke, E. R. Edelman, and C. Vlachopoulos, McDonald’s blood flow in
> arteries: theoretical, experimental and clinical principles. CRC press, 2022.

---

> ### Author Response · Authors · 2025-12-16
> **Reviewer Point P.1.10**
>
> * _The paper does not establish that current lack of test-time sampling is the main bottleneck to clinical adoption. Real-world clinical challenges (e.g. robustness to skin tone variation, ethical concerns, lack of regulation) are arguably more pressing than uncertainty estimates. I would like the authors to comment on this._
>
>
> We agree with the reviewer that challenges such as skin tone variation are important. However, we argue that uncertainty quantification and model evaluation on skin tone variation are closely related, especially for a clinician who has to make a decision whether to trust the model. To address the point, we include Table 5 in the main paper, which measures uncertainty on protected attributes such as gender and skin tone. We also include calibration curves for these protected attributes in Figure 9. The results in Table 5 show a performance gap between skin tones but the calibration curves in Figure 9 show decent performance. We hope that these new baselines will initiate more research in both skin tone variation and uncertainty quantification for iPPG.
>
>
> #### Table 5: Proper Scoring Rule metrics on Protected Attributes of MMSE-HR dataset
>
> | Protected Attribute | HR MAE | HR RMSE | NLL | Sharpness | CRPS | Check Score | Interval Score |
> | :--- | :--- | :--- | :--- | :--- | :--- | :--- | :--- |
> | Light Skin Tone | 2.17 | 3.74 | -3.430 | 0.203 | 0.042 | 0.021 | 0.209 |
> | Dark Skin Tone | 4.79 | 6.84 | -2.828 | 0.191 | 0.035 | 0.018 | 0.188 |
> | Men | 2.07 | 3.83 | -3.435 | 0.196 | 0.032 | 0.016 | 0.167 |
> | Women | 2.23 | 3.69 | -3.415 | 0.203 | 0.039 | 0.020 | 0.195 |
>
>
> **Action**: We have added text to the introduction that better explains why uncertainty quantification is important. We have also added Table 5 to the text to show the efficacy of our algorithm on skin tone and gender.
>
>
> [1] S. Tonekaboni, S. Joshi, M. D. McCradden, and A. Goldenberg, “What clinicians want: contex-
> tualizing explainable machine learning for clinical end use,” in Machine learning for healthcare
> conference. PMLR, 2019, pp. 359–380

---

> ### Author Response · Authors · 2025-12-17
> **Reviewer Point P.1.11**
>
> * _Section 4.4: How are overlapping windows selected? How much overlap is used? Is the method robust to window size and stride?_
>
>
> We had performed a grid search (using the validation set) over the the values of $\delta$ and $\lambda_{\text{RCL}}$ to find the best options. On the MMSE-HR dataset, we found that $\lambda_{\text{RCL}} = 0.1$ and $\delta = 9$ seconds worked best. Note that there are a variety of combinations of $\lambda_{\text{RCL}} $ and $\delta$ that show good performance; however, setting $\lambda_{\text{RCL}} = 0$ is suboptimal. Our results on the UBFC-rPPG dataset are in Table 9.
>
> #### Table 3: Varying the stride $\delta$ of Equation 12 and weight $\lambda_{\text{RCL}}$ of Equation 14 for the RCL loss on the MMSE-HR Dataset. We notice best performance, on average, when using $\delta=9$ and $\lambda_{\text{RCL}} = 0.1$
> | RCL weight $\lambda_{\text{RCL}}$ | Window Shift $\delta$ (seconds) | 1 | 2 | 3 | 4 | 5 | 6 | 7 | 8 | 9 | 10 | Average |
> | :---: | :---: | :---: | :---: | :---: | :---: | :---: | :---: | :---: | :---: | :---: | :---: | :---: |
> | 0.0 | | 3.72 | 3.72 | 3.72 | 3.72 | 3.72 | 3.72 | 3.72 | 3.72 | 3.72 | 3.72 | 3.72 |
> | 0.1 | | 2.67 | 2.54 | 2.47 | 3.44 | 2.79 | 2.54 | 3.51 | 2.52 | **1.39** | 2.89 | **2.67** |
> | 0.2 | | 2.64 | 3.77 | 2.42 | 2.04 | 2.77 | 4.07 | 2.82 | 1.94 | 2.52 | 2.59 | 2.74 |
> | 0.3 | | 2.54 | 4.67 | 3.87 | 2.39 | 4.74 | 2.92 | 3.09 | 2.42 | 2.34 | 2.24 | 3.12 |
> | 0.4 | | 2.59 | 2.42 | 5.97 | 3.39 | 4.19 | 2.67 | 2.69 | 3.42 | 2.12 | 2.87 | 3.23 |
> | 0.5 | | 3.02 | 2.67 | 2.51 | 2.57 | 1.92 | 3.19 | 2.74 | 2.85 | 2.84 | 2.67 | 2.73 |
> | 0.6 | | 3.02 | 3.01 | 2.89 | 3.07 | 2.12 | 2.77 | 2.72 | 2.47 | 3.72 | 2.34 | 2.83 |
> | 0.7 | | 2.77 | 4.07 | 2.19 | 2.69 | 2.92 | 2.44 | 2.44 | 3.09 | 1.82 | 3.57 | 2.80 |
> | 0.8 | | 2.67 | 2.87 | 2.84 | 2.09 | 4.57 | 3.69 | 3.22 | 2.37 | 2.14 | 2.49 | 2.89 |
> | 0.9 | | 1.77 | 2.57 | 2.25 | 3.32 | 1.49 | 3.34 | 2.94 | 3.82 | 2.72 | 2.89 | 2.71 |
> | 1.0 | | 2.69 | 4.17 | 3.22 | 2.34 | 2.79 | 6.54 | 3.27 | 1.62 | 1.59 | 2.19 | 3.04 |
> | Average | | 2.73 | 3.34 | 3.12 | 2.82 | 3.09 | 3.44 | 3.01 | 2.74 | **2.44** | 2.79 | |
>
> **Action**: We have included Tables 3 and 9 addressing this comment.

---

> ### Author Response · Authors · 2025-12-17
> **TeX not displaying correctly**
>
> The TeX is not displaying correctly in this comment. We will try to fix it. In the interim, please refer to the manuscript directly for the changes.

---

> ### Author Response · Authors · 2025-12-17
> **Reviewer Point P.1.7**
>
> _We forgot to include Reviewer Point P.1.7. and now the responses are out of order. We apologize for the confusion_
>
> * _Clarify what $x_0$ actually represents: Is it a temporal waveform?
> A spatial BVP map? A latent signal? Be precise about its dimensionality and nature._
>
> We have clarified this. The following text is added: Given a video signal of time $T$, we extract time-domain pixel intensity signals from five regions of the face to generate $\mathbf{x}_1  \in \mathbb{R}^{T\times5}$ as in Figure 1; the ground-truth BVP signal is repeated five times to generate $\mathbf{x}_0  \in \mathbb{R}^{T\times5}$. Broadcasting the signal in such a way allows for mapping the heterogeneous signal $\mathbf{x}_1$ to the homogeneous signal $\mathbf{x}_0$, effectively using spatial redundancy to improve the signal-to-noise ratio.
>
> **Action**: We added this text to the manuscript

---

> ### Author Response · Authors · 2026-01-15
> **Adding a revision letter**
>
> Since our LaTeX is not displaying correctly, we decided to add a typed revision letter, which is now appended to the manuscript. This letter starts at Page 36 of the updated PDF.

---

> ### Author Response · Authors · 2026-02-16
> **Revisions now visible**
>
> We had mistakenly set the responses to private. We have now made them public and welcome your comments. We have also appended to the manuscript a revision letter identical to the comments above for ease of reading.
>
> The Authors

---

### Review · Reviewer_VHhU · 2025-09-24

**Summary Of Contributions:**

The authors use a stochastic interpolant model that maps from (typically unobserved) blood volume pulse (BVP) to Imaging Photoplethysmography (iPPG) data (i.e. imaging of the skin). They invert this to obtain posterior samples of the BVP. They optionally add temporal regularization. The performance across three datasets is inline with the current SOTA, but additionally allows uncertainty quantification which is potentially valuable for clinical application.

**Audience:**

Yes

**Audience Explanation:**

Interesting application.

**Claims And Evidence:**

Yes

**Claims Explanation:**

The work is reasonably clearly written but it is challenging to understand the data structure. Section 4.1 starts with some irrelevant sentences about the pulmonary system, but does not specify what form the data comes in (x1,x0) pairs I assume? Multiple time points? Dimensionality?

The authors mention that inverting the mapping A is challenging. Why not just learn the iPPG -> BVP mapping directly then?

The temporal regularization feels like a bit of a hack compared to incorporating the temporal component into the model. Would it be too computationally expensive to do? Why is the regularization on the residual and not the BVP signal itself?

Why would the dimensions of the iPPG and BVP data be the same? Is the dimension just 1 in fact?

The results are not extremely strong but that is not a requirement for TMLR publication, and the uncertainty quant is a nice addition. It would be good to include a more interpretable metric like ECE as well.

Minor comments

Equation 13: rho is usually used for Spearman correlation. These don't really need defining either, they are v common metrics.
Define n_z
"Given the N signal estimates, we perform an uncertainty analysis of pulse signal recovery from a facial video." i.e. just Monte Carlo esimation?

**Requested Changes:**

Make the problem definition clearer. Add ECE to the uncertainty quantification metrics.

---

> ### Author Response · Authors · 2025-12-16
> **Reviewer Point P.2.1**
>
> * _The work is reasonably clearly written but it is challenging to understand the data structure. Section 4.1 starts with some irrelevant sentences about the pulmonary system, but does not specify what form the data comes in (x1, x0) pairs I assume? Multiple time
> points? Dimensionality?_
>
>
> Thank you for pointing this out. We believe that the details about the pulmonary system are important as they introduce the physics behind the measurement process in Equation 6. To more precisely describe our algorithmic inputs, we add the following sentences to Section 4.2: We begin by generating $(\mathbf{x}_0, \mathbf{x}_1)$ pairs from the data. Given a video signal of time $T$, we extract time-domain pixel intensity signals from five regions of the face to generate $\mathbf{x}_1  \in \mathbb{R}^{T\times5}$ as in Figure 1; the corresponding ground-truth BVP signal measured from the finger is repeated five times to generate $\mathbf{x}_0  \in \mathbb{R}^{T\times5}$, resulting in paired data $(\mathbf{x}_0, \mathbf{x}_1)$.
>
> **Action**: We add the above text to Section 4.2.

---

> ### Author Response · Authors · 2025-12-16
> **Reviewer Point P.2.2**
>
> * _The authors mention that inverting the mapping A is challenging. Why not just learn the iPPG $\rightarrow$ BVP mapping directly then?_
>
>
> Learning the iPPG $\rightarrow$ BVP map is _exactly_ what RIS-iPPG is doing, but based on reviewer comments, we did not write it clearly enough. The goal of the preliminary investigation in Section 4.2 is to precisely define the limitations of previous methods [1] [2]. We show that the approximation to the $A=F^{-1}$ is imprecise and often implicitly corrected. This implicit correction is highly non-linear, however; therefore, the formulation of previous work [1][2] is at best imprecise and likely misleading. To see if we can correct previous works' formulation, we attempt to learn a forward model, the experiments of which are in Appendix A.3.1. This attempt turned out to be unsuccessful for a variety of reasons, which we describe in Appendix A.3.1.
>
> That leads us to the original conclusion: a method to learn the mapping directly is likely a more effective solution. Thus, we developed RIS-iPPG.
>
>
>
> **Action**: We updated Section 4.2 and added new experiments and context to Appendix A.3.1.
>
> References:
>
> [1] Vineet R Shenoy, Tim K Marks, Hassan Mansour, and Suhas Lohit. Unrolled ippg: Video heart rate estimation via unrolling proximal gradient descent. In 2023 IEEE International Conference on Image Processing (ICIP), pp. 2715–2719. IEEE, 2023.
>
> [2] Ewa M Nowara, Tim K Marks, Hassan Mansour, and Ashok Veeraraghavan. Near-infrared imaging photoplethysmography during driving. IEEE Transactions on Intelligent Transportation Systems, 23(4):3589–3600, 2020.

---

> ### Author Response · Authors · 2025-12-16
> **Reviewer Point P.2.3**
>
> * _The temporal regularization feels like a bit of a hack compared to incorporating the temporal component into the model. Would it be too computationally expensive to do?_
>
>
> Thank you for the question. We considered incorporating explicit temporal components (such as RNNs, LSTMs) into the model architecture, but decided against it for two reasons:
>
>
> *  **Inference Efficiency (Solving the SDE)**: Our method relies on solving an SDE iteratively at test time. This requires multiple function evaluations of the network. If we were to incorporate heavy temporal layers (self-attention over long sequences, which are not necessary) into the backbone, the computational cost  the SDE solver would increase significantly. By keeping the backbone "stateless" and enforcing temporal consistency via the RCL loss instead, we maintain a low inference cost.
> * **Training Stability**: Learning accurate flow and score vector fields for diffusion is already a complex optimization task. Introducing recurrent dynamics can often lead to training instabilities (e.g., vanishing/exploding gradients) or significantly increased memory usage during backpropagation. The RCL is a training-time only constraint.
>
>
>
>
> In summary, the RCL allows us to achieve the desired physiological consistency (as shown in the improved metrics in Table 4) without the heavy computational penalty of a temporal architecture.
>
>
> **Action**: None

---

> ### Author Response · Authors · 2025-12-16
> **Reviewer Point P.2.4**
>
> * _Why is the regularization on the residual and not the BVP signal
> itself?_
>
> This is a great question. If we applied the regularization to the network outputs itself, it would encourage the network to ignore the phase shift i.e. $v(t) \approx v(t- \delta)$. However, we know _a priori_ that the signals **should not** be aligned, only that they should be consistent. By aligning the error residuals, we are saying that the structure of the error should be consistent, which is to be expected for a BVP signal over a short time period.
>
> As a sanity check, we explored the use of the RCL loss on the flow predictions directly, and ran experiments as discussed in Appendix Section A.9.1. As expected, the model struggled to converge as shown in Figure 8, and resulted in weak performance.
>
>
>
> **Action**: We have added an experiment to address this in Appendix Section A.8.1

---

> ### Author Response · Authors · 2025-12-16
> **Reviewer Point P.2.5**
>
> * _Why would the dimensions of the iPPG and BVP data be the same? Is the dimension just 1 in fact?_
>
>
> Stochastic interpolants are convex combinations of signals in high dimensional space; for this to be a valid interpolant, all signals must lie in the space high dimensional space. We chose the spatial dimension to be 5 to capture 5 regions of the face. We do this under the assumption that certain regions of the face may be corrupted by noise, but not all regions simultaneously . Given a time window of length $T$, we extract pixel intensity signals from 5 regions of the face to create a sample $\mathbf{x}_1 \in \mathbb{R}^{T\times5}$. The pulse signal is collected from finger PPG; this signal is originally $\mathbf{x}_0 \in \mathbb{R}^{T\times 1}$. During the loss computation, Pytorch "broadcasts" the ground-truth signal to match the length of the predicted signal (i.e. of dimension $\mathbb{R}^{T\times5}$); for clarity, we mention that the signal is repeated 5 times for loss computation.
>
>
>
> **Action**: None

---

> ### Author Response · Authors · 2025-12-16
> **Reviewer Point P.2.6**
>
> * _It would be good to include a more interpretable metric like ECE as well._
>
> While ECE is usually defined for classification tasks, we have added a variety of new metrics suitable for regression including the Continuous Ranked Probability Score, Sharpness, Check Score/Pinball Loss, and Interval Score in addition to the Negative Log Likelihood in the main paper Table 4 and Table 5.  Furthermore, we have plotted the calibration curves, which instead of summarizing a mean, shows the calibration error at each predicted proportion as well as the miscalibration error [1]. We achieve a miscalibration error of 0.06 on both the UBFC-rPPG and PURE datasets as shown in Figure 5. We also plot calibratoin curves on protected attributes in Figure 9.
>
> References:
>
> [1]Kuleshov, Volodymyr, Nathan Fenner, and Stefano Ermon. "Accurate uncertainties for deep learning using calibrated regression." International conference on machine learning. PMLR, 2018.
>
>
> #### Table 4: Comparing the spectrum estimation performance and uncertainty quantification metrics with and without the RCL loss
>
> | Method | MAE| RMSE | PCC | NLL | Sharpness | CRPS | Check Score | Interval Score |
> | :--- | :--- | :--- | :--- | :--- | :--- | :--- | :--- | :--- |
> | PURE w/out RCL | 0.057 | 0.681 | 0.224 | -3.825 | 0.201 | 0.041 | 0.021 | 0.217 |
> | PURE w/RCL | 0.700 | 0.056 | 0.218 | -3.820 | 0.202 | 0.040 | 0.020 | 0.208 |
> | UBFC-rPPG w/out RCL | 0.07 | 0.651 | 0.237 | -3.475 | 0.177 | 0.042 | 0.0121 | 0.162 |
> | UBFC-rPPG w/RCL | 0.049 | 0.795 | 0.185 | -3.516 | 0.168 | 0.034 | 0.017 | 0.162 |
>
>
> #### Table 5: Proper Scoring Rule metrics on Protected Attributes of MMSE-HR dataset
>
> | Protected Attribute | HR MAE | HR RMSE | NLL | Sharpness | CRPS | Check Score | Interval Score |
> | :--- | :--- | :--- | :--- | :--- | :--- | :--- | :--- |
> | Light Skin Tone | 2.17 | 3.74 | -3.430 | 0.203 | 0.042 | 0.021 | 0.209 |
> | Dark Skin Tone | 4.79 | 6.84 | -2.828 | 0.191 | 0.035 | 0.018 | 0.188 |
> | Men | 2.07 | 3.83 | -3.435 | 0.196 | 0.032 | 0.016 | 0.167 |
> | Women | 2.23 | 3.69 | -3.415 | 0.203 | 0.039 | 0.020 | 0.195 |
>
>
> **Action**: We have added new metrics related to regression in Table 5, as well as included calibration curves in Figures 5 and 9.

---

> ### Author Response · Authors · 2025-12-16
> **Reviewer Point P.2.7**
>
> * _Equation 13: $\rho$ is usually used for Spearman correlation. These
> don’t really need defining either, they are v common metrics._
>
>
> We agree with the reviewer that $\rho$ is often used for Spearman. However, we followed the convention used previous iPPG papers such as [1] to avoid confusion and promote consistency. We have elected to keep the definition of these metrics in Section 5.2.
>
>
> **Action**: None
>
> [1] X. Liu, G. Narayanswamy, A. Paruchuri, X. Zhang, J. Tang, Y. Zhang, Y. Wang, S. Sen-
> gupta, S. Patel, and D. McDuff, “rppg-toolbox: Deep remote ppg toolbox,” arXiv preprint
> arXiv:2210.00716, 2022

---

> ### Author Response · Authors · 2025-12-16
> **Reviewer Point P.2.8**
>
> * _Define ``Given the N signal estimates, we perform an uncertainty analysis of pulse signal recovery from a facial video." i.e. just Monte Carlo estimation?_
>
>
> Correct. To be precise, we approximate the posterior distribution of the pulse signal $p(x_0|x_1)$ by generating N independent trajectories via the reverse-time SDE. This acts as a Monte Carlo estimator of the posterior, allowing us to compute empirical confidence intervals and calibration metrics. We agree that using the explicit term helps clarify the methodology. We have updated the text to reflect this.
>
>
>
> **Action**: We have clarified this in the introduction

---

> ### Author Response · Authors · 2025-12-16
> **Reviewer Point P.1.7**
>
> * _Clarify what $x_0$ actually represents: Is it a temporal waveform?
> A spatial BVP map? A latent signal? Be precise about its dimensionality and nature._
>
> We have clarified this. The following text is added: Given a video signal of time $T$, we extract time-domain pixel intensity signals from five regions of the face to generate $\mathbf{x}_1  \in \mathbb{R}^{T\times5}$ as in Figure 1; the ground-truth BVP signal is repeated five times to generate $\mathbf{x}_0  \in \mathbb{R}^{T\times5}$. Broadcasting the signal in such a way allows for mapping the heterogeneous signal $\mathbf{x}_1$ to the homogeneous signal $\mathbf{x}_0$, effectively using spatial redundancy to improve the signal-to-noise ratio.
>
> **Action**: We added this text to the manuscript

---

> ### Author Response · Authors · 2026-02-16
> **Revisions now visible**
>
> We had mistakenly set the responses to private. We have now made them public and welcome your comments. We have also appended to the manuscript a revision letter identical to the comments above for ease of reading.
>
> The Authors

---

### Review · Reviewer_ZKFz · 2026-01-09

**Summary Of Contributions:**

The authors propose RIS-iPPG, a new generative sampling method that aims to solve the iPPG problem, which is also the first to provide quantified uncertainty estimates refined through RCL loss which they also propose in this paper, with sufficient empirical evidence on three datasets and good theoretical guarantees.

**Audience:**

Yes

**Audience Explanation:**

While the paper focuses on the iPPG problem, it introduces a generative, sampling-based formulation for posterior signal recovery with uncertainty quantification that is of broader interest. The approach and regularization design are applicable to other inverse or signal reconstruction problems, particularly in medical and physiological sensing. In addition, the use of diffusion-inspired stochastic interpolants makes the work relevant to TMLR readers interested in generative modeling, uncertainty estimation, and structured time-series recovery beyond the specific iPPG application.

**Claims And Evidence:**

Yes

**Claims Explanation:**

The authors compare RIS-iPPG to a broad list of baselines and SOTAs of different types on three different datasets of the iPPG problem, showing comparable and competitive results, being top or second of performance in most evaluated metrics. Additionally, uncertainty metrics are also reported in ablation of the proposed RCL loss on two datasets. Methodologies are well narrated and basic intuitions/theories behind are clearly explained.

**Requested Changes:**

1. Existing theoretical analysis primarily serves to motivate and explain the proposed formulation, but does not address properties such as generalization, robustness, or stability. For example, how would the error from training compose with the error from the SDE solver? It would be nice to include those outcome-focused theoretical analyses.
2. Test-time SDE-based sampling has the advantage to access solutions to all time points. Is it iterative w.r.t. $t$, i.e. you can't get later $t$ if you don't get earlier $t$ during the simulation / the solver you're using. If so, the sampling seems to take some time and it would be interesting to see the inference time comparing different methods to see if it's time-efficient as well (No need to beat them, comparable is sufficient).
3. Typo: "OUr" in table 2 caption.

---

> ### Author Response · Authors · 2026-01-14
> **Review Point P.3.1**
>
> We thank the reviewer for all their comments to improve our paper. We address them, starting with Point P.3.1.
> * _Existing theoretical analysis primarily serves to motivate and explain the proposed formulation, but does not address properties such as generalization, robustness, or stability. For example, how would the error from training compose with the error from the SDE solver? It would be nice to include those outcome-focused theoretical analyses._
>
> We thank the reviewer for suggesting an outcome-focused theoretical analysis. We agree that addressing properties such as generalization, robustness, and stability is critical for clinical adoption. We have added a new section to Appendix A.9 detailing these concepts. We summarize our additions below:
>
> * **Error Composition (Approximation vs. Discretization):** To address the reviewer's question on how errors compose, we model the total sampling error as the sum of the \textit{network approximation error} and the \textit{solver discretization error}. As detailed in Equation 2, the error bound is given by:
>
> $$\text{Total Error} \leq \mathbb{E}\left[\int_{0}^{1} \|b_{F}(t,x_{t}) - b_{\theta}(t,x_{t})\| dt\right] + \mathcal{O}(\Delta t)$$
>
> The first term represents the training quality (how well our neural network $b_{\theta}$ approximates the true vector field $b_F$). The second term, $\mathcal{O}(\Delta t)$, represents the error introduced by the SDE solver steps. This confirms that the error is additive and controllable via training convergence (Figure 7) and step-size selection.
>
> * **Theoretical Stability (BIBO):**  We have proven that our system is \textbf{Bounded-Input, Bounded-Output (BIBO) stable}. In \textbf{Theorem 1} (added to Appendix A.9), we show that if the learned drift coefficient $b_{F, \theta}$ is Lipschitz continuous with constant $L$ (a property generally satisfied by U-Net architectures with bounded activations), the reconstruction error between a clean measurement $\mathbf{\hat{x}}_1$ and a perturbed measurement $\mathbf{\tilde{x}}_1 = \mathbf{\hat{x}}_1 + \delta$ is bounded by:
>
>     $$\|\mathbf{\hat{x}}_0 - \mathbf{\tilde{x}}_0 \| \leq e^L \|\delta\|$$
>
>     This proves that bounded input noise $\|\delta\| < \infty$ results in bounded reconstruction error.
>
> * **Empirical Validation of Stability:** We validate this theoretical stability via a \textbf{Gauge R\&R analysis} (Appendix A.9.6). We demonstrate that the variance due to the measurement system (repeatability) is negligible ($1.7 \text{ percent}$) compared to the signal variation. This empirically confirms the stability predicted by our theorem.
>
> * **Robustness and Generalization via RCL:** Finally, regarding generalization, the Residual Correlation Loss (RCL) acts as an inductive bias (or smoothness prior). By constraining the solution manifold to signals that are consistent across time shifts, we theoretically reduce the hypothesis space to physically plausible signals. This improves robustness against high-frequency outliers (e.g., sudden motion artifacts) that violate this correlation structure. We have also added training/validation curves in Figure 7 to show that our model learns the flow and score correctly.
>
> **Action**: We have added significant changes to Appendix A.9. We have added a proof on BIBO stability to Appendix A.9, added the error composition analysis (Eq. 37) to the Appendix A.9, and include a discussion of Robustness and Generalization in Appendix A.9. We have also included the Gauge R&R empirical stability analysis in Appendix A.10.6.

---

> ### Author Response · Authors · 2026-01-14
> **Reviewer Point P.3.2**
>
> * _Test-time SDE-based sampling has the advantage to access solutions to all time points. Is it iterative w.r.t. $t$, i.e. you can't get later if you don't get earlier during the simulation / the solver you're using. If so, the sampling seems to take some time and it would be interesting to see the inference time comparing different methods to see if it's time-efficient as well (No need to beat them, comparable is sufficient)._
>
> Thanks for your comment. Yes, solving an SDE is expensive as it is sequential and iterative, require multiple evaluations of our flow and score networks. We can not obtain the solution at a later time without first evaluating the solution at earlier times; if inference speed is the dominating constraint when chosing a posterior sampling method, then RIS-iPPG may not be the best choice.
>
> We further explore this by measuring the inference times for a single forward pass of different models in Table 10, Appendix A.5. We notice that RIS-iPPG has the highest inference time; however, RIS-iPPG has many benefits, as enumerated in Appendix A.4. We believe inference speed is important; future work should explore accelerated sampling techniques, an active research area for the diffusion community.
>
> #### Table 10: Comparing inference speed of different methods. All were tested on a single NVIDIA A5000 GPU
> | Method | Inference Time (ms) |
> | :--- | :--- |
> | Conditional VAEs | 845.06 |
> | Bayesian Neural Networks | 473.91 |
> | RIS-iPPG | 850.97 |
>
> We add the Table 10 and the following text to the manuscript Section A.5:
>
> _We measure the runtime performance at inference for three sampling methods in Table 10. At inference, BNNs are best in terms of speed; cVAEs and RIS-iPPG are about the same. However, the models in RIS-iPPG need to be evaluated for every timstep, and must do so sequentially; depending on the granularity with which the SDE steps are discretized, the cost of RIS-iPPG can be much higher. Note that we do not claim RIS-iPPG to be superior in terms of runtime; however, fast inference is important for real-world adoption and future work should aim to improve inference speed._
>
> **Action**: We have added the above table and text to Appendix Section A.5.

---

> ### Author Response · Authors · 2026-01-14
> **Reviewer Point P.3.3**
>
> * _Typo: ``OUr" in table 2 caption._
>
> Thanks for pointing this out. We have fixed it.
>
> **Action**: We have fixed the typo.

---

> ### Author Response · Authors · 2026-02-16
> **Revisions now visible**
>
> We had mistakenly set the responses to private. We have now made them public and welcome your comments. We have also appended to the manuscript a revision letter identical to the comments above for ease of reading.
>
> The Authors

---

### Decision · Action_Editor_5146 · 2026-05-11

**Recommendation:** Reject

**Audience:**

Yes

**Audience Explanation:**

Yes. The paper sits at the intersection of several active research areas: generative modeling, uncertainty quantification, and biomedical signal processing, each of which has a substantial audience within TMLR. The work is of general interest to researchers working on probabilistic inference and scientific applications of machine learning.

**Claims And Evidence:**

No

**Claims Explanation:**

The authors demonstrate competitive performance and provide a reasonable empirical case for their formulation. The revised manuscript substantially strengthens the evidence. However, the uncertainty quantification results lack baselines from other probabilistic methods, making it difficult to assess the calibration quality. Additionally, the justification for the complexity of the approach over simpler alternatives remains more empirical than principled. Reviewer VHhU remains unconvinced after rebuttal, and recommended leaning reject.

**Resubmission Of Major Revision:**

The authors may consider submitting a major revision at a later time.